# A COVID-19 descriptive study of life after lockdown in Wuhan, China

Registered report  

Subject Areas:
psychology

Keywords:
loneliness, depression, well-being, COVID-19

Author for correspondence:
Thuy-vy Thi Nguyen
e-mail: thuy-vy.nguyen@durham.ac.uk

Tong Zhou[1], Thuy-vy Thi Nguyen[2], Jiayi Zhong[1] and Junsheng Liu[1]

[1]Department of Psychology, The School of Psychology and Cognitive Science, East China Normal University, Shanghai, People's Republic of China
[2]University of Durham, Durham, UK

(iD) TTN, 0000-0003-0777-4204

On 8 April 2020, the Chinese government lifted the lockdown and opened up public transportation in Wuhan, China, the epicentre of the COVID-19 pandemic. After 76 days in lockdown, Wuhan residents were allowed to travel outside of the city and go back to work. Yet, given that there is still no vaccine for the virus, this leaves many doubting whether life will indeed go back to normal. The aim of this research was to track longitudinal changes in motivation for self-isolating, life-structured, indicators of well-being and mental health after lockdown was lifted. We have recruited 462 participants in Wuhan, China, prior to lockdown lift between 3 and 7 April 2020 (Time 1), and have followed up with 292 returning participants between 18 and 22 April 2020 (Time 2), 284 between 6 and 10 May 2020 (Time 3), and 279 between 25 and 29 May 2020 (Time 4). This four-wave study used latent growth models to examine how Wuhan residents' psychological experiences change (if at all) within the first two months after lockdown was lifted. The Stage 1 manuscript associated with this submission received in-principle acceptance (IPA) on 2 June 2020. Following IPA, the accepted Stage 1 version of the manuscript was preregistered on the OSF at https://osf.io/g2t3b. This preregistration was performed prior to data analysis. Generally, our study found that: (i) a majority of people still continue to value self-isolation after lockdown was lifted; (ii) by the end of lockdown, people perceived gradual return to normality and restored structure of everyday life; (iii) the psychological well-being slightly improved after lockdown was lifted; (iv) people who used problem solving and help-seeking as coping strategies during lockdown had better well-being and mental health by the end of the lockdown; (v) those who experienced more disruptions in daily life during lockdown would display more indicators of psychological ill-being by the end of the lockdown.

# 1. Research question and background

## 1.1. Purpose

In this registered report, we conducted a longitudinal study to investigate the psychological experiences of those living in Wuhan, China, prior to lockdown lift, and how their experiences changed within two months after lockdown is lifted. We planned to administer four assessments at four different time points: one immediately prior to lockdown lift in Wuhan, and three follow-up surveys every two weeks after. As such, this data allowed us to observe change over the course of the first two months after lockdown. The time frame of when each assessment took place is displayed in figure 1.

## 1.2. Background

On 23 January, Wuhan, China—the capital city of Hubei province with a population of more than 11 million people—went under lockdown after the city was identified as the epicentre of the novel coronavirus disease. Several restrictions were enforced: (i) residents were not allowed to leave the city, (ii) were only allowed to leave their houses for food, and (iii) all public transportation was cancelled. By mid-February, the Chinese government toughened up the restrictions: no one was allowed to leave their homes without permission and residents' movements were surveillanced to track those who were buying cold medicine (*NPR* [1], *The Guardian* [2]).

On 8 April, China ended the lockdown in Wuhan, China, allowing residents to travel outside of the city limits, people to go back to work and businesses to reopen. However, progress towards 'business-as-usual' will take time as residents' movements continue to be regulated and people are advised to keep practising self-isolation and physical distancing. Given that this 'life after lockdown' will be a reality in many places around the world until a vaccine for the virus is discovered, Wuhan provides an interesting case to observe changes after lockdown, including change in motivation for self-isolating and residents' psychological well-being.

## 1.3. Significance

News outlets and media coverage of the ongoing outbreak have expressed concerns that life under lockdown is challenging to many and the process of going back to normal life after lockdown will be difficult. However, from recent analysis of data collected as part of a registered report by Weinstein & Nguyen [3], the authors observed little change in ill-being (i.e. loneliness, depressive symptoms, anxiety). This is consistent with another report from data collected at the University College London (https://www.marchnetwork.org/research) suggesting that life satisfaction has not decreased after four weeks into lockdown. Data collected from a national sample in the United States shows a rise in percentage of those reporting moderate to severe psychological distress around the end of March and beginning of April but this percentage gradually went down in April and May (https://covid19pulse.usc.edu/). In China, a study that looked at entrepreneurs and employees also reported anxiety to be within the normal range during and after lockdown (https://english.ckgsb.edu.cn/new/psychological-resilience-before-and-after-work-resumption-during-covid-19-episode-1-tracking-work-resumption/).

This evidence yielded support for the argument that while a decrease in mental health might be observed immediately after a significant event occurs, such as job loss, a natural disaster, an accident or in this case the pandemic, a single stressful life event is not likely to cause permanent impact on mental health [4,5]. However, data from two-panel studies suggested that it depends on the life event; people appear to be more able to bounce back from events such as loss of a spouse or divorce, but unemployment or disability appear to have more long-lasting impact (see review by [6]). Further, there are also great individual differences in adaptation, making it challenging to anticipate change in psychological well-being and ill-being throughout the course of the current pandemic.

Therefore, it is important to investigate how psychological well-being and ill-being change in the first two months after lockdown by centring on the experiences of those in Wuhan, due to its unique situation as the centre of the coronavirus disease. Wuhan residents are considered a high-risk public population that is likely to experience more psychological consequences of this pandemic than the general Chinese population or populations in other countries that undergo less severe lockdown restrictions. Further, this study also allows us to look at how different coping strategies or types of stressors experienced during lockdown might be linked to the degree to which psychological well-being and

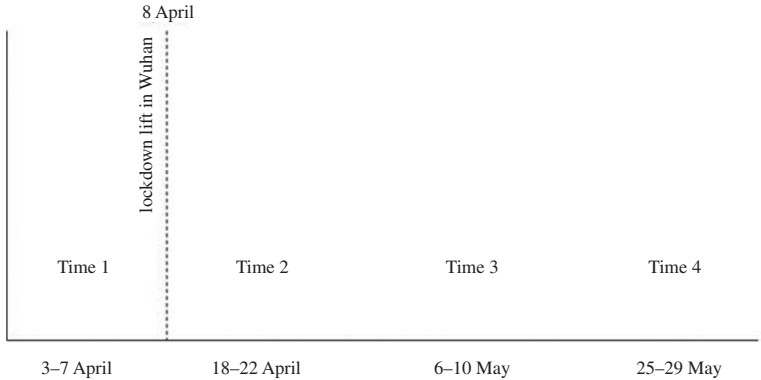

**Figure 1.** Time frame of the assessment in the current study.

ill-being change after lockdown is lifted. Wuhan is now one of the very few places in the world that have experienced life after lockdown long enough to allow for this longitudinal investigation.

Since a vaccine is not yet discovered, another concern about lockdown lift is that it will remove behavioural restrictions around self-isolation—a measure that can ensure slowing down the spread of the virus. Available data collected in other large-scale studies, including the UK COVID-19 Social Study (https://www.covidsocialstudy.org/), the Understanding America Study (https://covid19pulse. usc.edu/) and the International Survey on Coronavirus (https://covid19-survey.org/) all suggested that people around the world have shown high adherence to self-isolating restrictions. Nonetheless, there is little information on whether people are self-isolating because they understand the benefits and value of this behaviour or because they are simply doing it because of government's restrictions. While it makes sense to worry that lack of government restrictions will cause people to behave irresponsibly and stop practising self-isolation, given that the fear around the virus is salient, people might continue to self-isolate out of health concerns for themselves and others. Previous research has distinguished identified motivation—engaging in a behaviour because one sees its values and benefits—from external motivation—doing something because of outside influences [7]. In the context of this pandemic, we will look at both identified motivation for self-isolation—doing it because one sees the benefit of self-isolation to protecting oneself and other people—and external motivation— doing it because one fears social and legal consequences. There are reasons to be concerned that both motivations might go down once lockdown is lifted, such that people will see less value and benefits in self-isolating as well as perceive less external pressure to do it. This study will be the first to investigate whether such concern is valid.

While the data collected from this sample in Wuhan is not generalizable to experiences of the population in China and other countries, observing changes over time of life after lockdown among this subset of WeChat users in Wuhan allows us to navigate our expectations of what life will be like after lockdown is lifted. We will ask the following five questions:

1. Without a vaccine for COVID-19, the risk of infection remains. Will we see a change in motivation for self-isolation after lockdown is lifted?
2. As people are allowed to travel and get back to work, people are likely to go back to setting daily goals and having specific tasks to focus on. Will we see a change in structure of daily life after lockdown is lifted?
3. With the promise of return to normality after lockdown is lifted, should we expect that quality of life will become 'better' after lockdown is lifted compared with during lockdown?
4. Which coping strategies used during lockdown will contribute to psychological well-being and alleviate ill-being after lockdown is lifted?
5. Which stressors experienced during lockdown will undermine psychological well-being and contribute to ill-being after lockdown is lifted?

## 2. Methods

### 2.1. Statement of transparency

All measures, data and code are available on Open Science Framework at https://osf.io/5rxz6/?view_ only=b3a7c3653bd043d189eea2513e66bfde. We have received ethical approval from the University of

Durham's Ethics Committee (PSYCH-2020-03-11T23_41_08) and East China Normal University's Ethics Committee (HR 179-2020).

## 2.2. Measures

### 2.2.1. Measures for descriptive analyses

We have examined demographic information of the current sample (see appendix A). Immediately prior to lockdown being lifted (3–7 April 2020), we have collected data from 462 adults who were inside the city of Wuhan during lockdown (Time 1). Two weeks after lockdown was lifted, 318 adults' data were collected (Time 2). One month after lockdown was lifted, 308 participants joined again in this study (Time 3) and finally one and half months after lockdown was lifted, 306 participants finished the last survey (Time 4). The following information is obtained from data at Time 1 to Time 4.

#### 2.2.1.1. Age

The mean age of our sample is 37.39 (s.d. = 13.15) at Time 1, 35.36 (s.d. = 13.40) at Time 2, 35.76 (s.d. = 13.13) at Time 3 and 35.62 (s.d. = 13.21) at Time 4.

#### 2.2.1.2. Gender

Time 1 sample consists of 166 men (35.93%) and 296 women (64.07%), and sample of Time 2 contained 116 men (36.60%) and 201 women (63.40%), for Time 3, the gender composition was 109 men (35.40%) and 199 women (64.60%). At Time 4, our sample contained 109 men (35.60%) and 197 women (64.40%). These deviate slightly from the gender make-up of Wuhan which is split 51.40% men and 48.60% women.

#### 2.2.1.3. Marital status

There were seven options for marital status, including married, living together as married, divorced, separated, widowed, single/never married, living apart but steady relation (see frequency in appendix A). In general, the majority of our participants were married (63.20%) or single/never married (22.10%) at Time 1. This sample make-up of marital statuses did not change notably at Time 2 (59.00% married, 25.90% single/never married), Time 3 (61.00% married, 25.30% single/never married) or Time 4 (60.80% married, 25.80% single/never married).

#### 2.2.1.4. Employment status

There are seven options for employment status, including 'full-time', 'part-time', 'self-employed', 'retired', 'housewife', 'students', 'unemployed' or 'other'. This question format was taken from the World Value Survey (http://www.worldvaluessurvey.org/wvs.jsp). In general, the majority of our participants were full-time (51.10%), retired (10.60%) and students (15.80%). This sample composition of employment statuses was relatively comparable at Time 2 (50.20% full-time, 9.80% retired and 12.60% students), Time 3 (51.90% full-time, 9.70% retired and 17.20% students) and Time 4 (54.20% full-time, 9.50% retired and 16.00% students).

#### 2.2.1.5. Self-isolated because of COVID-19 in the past two weeks

We asked participants to answer 'yes', 'no' or 'somewhat or in part' to whether they self-isolated in response to the coronavirus outbreak (COVID-19). At Time 1, 80.30% of our participants answered 'yes', this proportion declined over time, with 65.60% answered 'yes' at Time 2, 45.50% answered 'yes' at Time 3 and 36.30% answered 'yes' at Time 4. The frequencies of participants answering 'yes', 'no' or 'somewhat or in part' at each time point is reported in appendix A.

#### 2.2.1.6. Infected by the coronavirus (COVID-19)

Participants answered 'yes' or 'no' to the question of whether they were infected by the virus. At Time 1, only three people in this sample answered yes to this question, and there is no one reported infected by the virus at Time 2, Time 3 and Time 4.

### 2.2.1.7. Knowledge of people who are infected by COVID-19

Participants answered 'yes' or 'no' to whether they knew anyone who has been infected by the virus. At Time 1, 32.20% of the sample reported yes to this question, and a comparable sample answered yes at Time 2 (31.40%), Time 3 (35.40%) and Time 4 (34.00%).

### 2.2.1.8. General state of health

There are six options for levels of perceived health, including 'very poor', 'poor', 'fair', 'good', 'very good' or 'I'm not sure'. This question format was taken from the World Value Survey (http://www. worldvaluessurvey.org/wvs.jsp). In general, the majority of our participants reported being in healthy state at each time point, with 41.80–46.40% of participants reported in good health, and 42.60–47.30% of participants reported in very good health.

## 2.2.2. Measure used for quality control

### 2.2.2.1. Subjective perception of normality

In the surveys administered after lockdown lift (Time 2–4), we asked the participants to answer the question 'Think about your life before the COVID-19 outbreak, how much of your life has returned to normal?' on a scale between 0 and 1. We should expect that the percentage of life returning to normality to increase between Time 2, Time 3 and Time 4.

## 2.2.3. Measures used as covariates

### 2.2.3.1. Stressors during quarantine

In Time 2, Time 3 and Time 4 assessments, we used a stressor checklist adapted from a nationwide study of stressful life events experienced by the general population in Mainland China by Zheng & Lin [8]. To ascertain that we understood the unique experiences that Wuhan residents have experienced as a result of the COVID-19 outbreak, we included an open-ended question asking 'In a few words, please describe how self-isolation has influenced your life' in the Time 1 assessment. The responses have been coded and included in appendix B.

Based on the open-ended responses at Time 1, we added new items to the stressor checklist used in Time 2, Time 3 and Time 4 (see appendix C). We asked the participants to indicate whether each event has happened during time in lockdown (measured at Time 2) and after lockdown was lifted (measured at Time 3 and Time 4) and, if it did, how intensely the event influenced their life (0 = never happened, 1 = extremely mild, 2 = mild, 3 = moderate, 4 = severe, 5 = extremely severe). We calculated the average of all the items that belong to each type of stressors so scores range between 0 and 5 and reflect how intensely each type of stressor has affected the participants' life during lockdown and after lockdown.

### 2.2.3.2. Coping strategies

To evaluate Wuhan residents' coping strategies, we asked participants about coping strategies during the lockdown at Time 2, Time 3 and Time 4. We used the Coping Style Scale that has been validated on Chinese samples by Xiao & Xu [9]. We asked the participants to answer 'yes' or 'no' to whether they used each of the strategies listed to cope with events in their life. Coping strategies are categorized into (i) problem solving (e.g. 'Able to deal with difficulties rationally', 'Try to change the situation and make it better'), (ii) self-blame (e.g. 'Give up on myself', 'Often complain of my own competence'), (iii) help-seeking (e.g. 'Often like to talk with someone to ease the worry', 'Ask others for help in overcoming difficulties'), (iv) fantasizing (e.g. 'Think of something happy to comfort myself', 'Fantasize some unrealistic things to eliminate the trouble'), (v) avoidance (e.g. 'Avoid difficulties for peace of mind', 'Drink or smoke to escape difficulties'), (vi) rationalization (e.g. 'Think "life's experience is suffering", "frustration is a test of myself"'). If the participant chose 'no' as a response, the answer was coded as 0, and if the participant chose 'yes', it was coded as 1. We calculated the average of all the items that belong to each coping style, so scores range between 0 and 1 and reflect the prevalence of each coping strategy in the present sample.

## 2.2.4. Measures of dependent variables

We measured the following dependent variables at all four time points: immediately prior to when lockdown was lifted (3–7 April 2020, Time 1), two weeks after lockdown was lifted (18–22 April

2020, Time 2), four weeks after lockdown was lifted (6–10 May 2020, Time 3), and six weeks after lockdown was lifted (25–29 May 2020, Time 4). Internal consistency for each measure is reported in table 1, along with the correlations between those variables with one another at each time point.

### 2.2.4.1. Identified motivation for self-isolating

We used the same items reported in Weinstein & Nguyen's [3] study, which surveyed participants in the US and UK about their experiences during lockdown in response to COVID-19. These items are adapted from Ryan & Connell's [10] self-regulation questionnaire (SRQ) and have been modified to assess reasons for engaging in self-isolation for personally meaningful reasons like health benefits or protection of others' health. Items for identified regulation have been validated in Chinese in relation to sport behaviours [11]. In this study, there are five items in total: 'because self-isolation was beneficial to me', 'because I really valued the importance of self-isolating', 'because self-isolation was important for protecting my health', 'because it was beneficial to me' and 'because self-isolation was beneficial for others who are important to me'. We asked the participants to rate how true each of the items are to them on a 7-point scale, ranging from 1 = not at all true to 7 = very true.

### 2.2.4.2. External motivation for self-isolating

Similarly, we used the same items reported in Weinstein & Nguyen's [3] study, which surveyed participants in the US and UK about their experiences during lockdown in response to COVID-19. These items are adapted from Ryan & Connell's [10] SRQ and have been modified to assess reasons for engaging in self-isolation to meet others' expectations or to avoid externally imposed consequences. Items for external regulation have been validated in Chinese in relation to sport behaviours [11]. In this study, there are five items in total: 'because I was afraid there would be harsh consequences if I didn't self-isolate', 'because of some external circumstances that make me', 'because I would get in trouble with others if I didn't', 'because I was forced into it' and 'because I felt I had no choice'. We asked the participants to rate how true each of the items are to them on a 7-point scale, ranging from 1 = not at all true to 7 = very true.

### 2.2.4.3. Life structure

We developed a scale to assess the extent to which residents had clear goals and tasks to fill their time during the time in isolation. There are five items in total: 'I had a specific task or tasks to focus my attention', 'I was clear about what I would be doing', 'I had clear and specific goals for how my day would go', 'I was unsure about what I would do throughout the day' (reverse scored) and 'I did not have clear goals guiding how I used my time' (reverse scored). We asked the participants to rate how true each of the items were to them on a 6-point scale, ranging from 1 = not at all true to 6 = very true.

### 2.2.4.4. Loneliness

To measure loneliness, we used 20 items from the revised loneliness rating scale [12] to assess affective experiences related to feelings of being low in energy and effectiveness (depletion; e.g. 'drained', 'empty', 'numb') and feelings of being rejected by one's social circles or community (isolation; e.g. 'unloved', 'worthless', 'hopeless'). In consideration of space, we did not include 20 other items that concern lonely feelings related to relational frustration and depression-related states [13], since they are not central to our research question and can be redundant with the measure of depressive symptoms. We asked the participants to rate how frequently they have had those emotions in the past two weeks on a 7-point scale, ranging from 1 = never to 7 = always.

### 2.2.4.5. Depressive symptoms

We used the 10-item Center for Epidemiologic Studies depression scale (CESD; [14]) to measure the extent to which participants experienced any symptoms of depressed mood in the past two weeks. Examples of items reflecting depressed mood are 'I felt depressed', 'I felt bothered by things that usually don't bother me', and reverse coded items are 'I felt hopeful about the future'. We asked the participants to rate how frequently they experienced depressed mood on a 6-point scale, ranging from 1 = none at all to 6 = most or all of the time.

**Table 1.** Means, standard deviations and correlations between main variables.

| variable | α | M | s.d. | self-isolation | COVID-19 cases | state of health | 1 | 2 | 3 | 4 | 5 | 6 | 7 |
|---|---|---|---|---|---|---|---|---|---|---|---|---|---|
| **Time 1 (n = 462)** | | | | | | | | | | | | | |
| 2. identified | 0.87 | 6.62 | 0.65 | 0.00 | −0.11 | 0.15** | — | | | | | | |
| 3. external | 0.64 | 4.61 | 1.20 | −0.05 | −0.02 | −0.05 | — | 0.02 | | | | | |
| 4. life structure | 0.76 | 4.19 | 1.06 | 0.10 | 0.13 | 0.12* | — | 0.08 | −0.08 | | | | |
| 5. depletion | 0.95 | 2.82 | 1.33 | 0.13* | −0.14 | −0.20** | — | −0.21** | 0.24** | −0.40** | | | |
| 6. isolation | 0.97 | 2.01 | 1.14 | 0.00 | −0.16 | −0.22** | — | −0.28** | 0.28** | −0.33** | 0.75** | | |
| 7. depression | 0.91 | 2.78 | 0.96 | 0.14* | −0.21* | −0.19** | — | −0.19** | 0.15** | −0.44** | 0.83** | 0.67** | |
| 8. need satisfaction | 0.81 | 4.73 | 0.84 | 0.00 | 0.18 | 0.13* | — | 0.28** | −0.09 | 0.53** | −0.58** | −0.51** | −0.68** |
| **Time 2 (n = 292)** | | | | | | | | | | | | | |
| 1. normality | | 54.78 | 23.73 | −0.23** | −0.14 | 0.17** | — | | | | | | |
| 2. identified | 0.91 | 6.41 | 0.90 | 0.09 | 0.08 | 0.06 | 0.10 | | | | | | |
| 3. external | 0.70 | 4.44 | 1.21 | 0.06 | 0.03 | −0.09 | −0.03 | 0.09 | | | | | |
| 4. life structure | 0.81 | 4.27 | 0.96 | 0.03 | 0.20 | 0.14* | 0.11 | 0.20** | −0.04 | | | | |
| 5. depletion | 0.95 | 2.73 | 1.31 | −0.03 | −0.08 | −0.23** | −0.26** | −0.28** | 0.17** | −0.40** | | | |
| 6. isolation | 0.97 | 2.00 | 1.16 | 0.04 | −0.19 | −0.24** | −0.26** | −0.36** | 0.16** | −0.45** | 0.79** | | |
| 7. depression | 0.89 | 2.76 | 0.89 | −0.02 | −0.15 | −0.26** | −0.26** | −0.28** | 0.13* | −0.55** | 0.78** | 0.71** | |
| 8. need satisfaction | 0.80 | 4.75 | 0.87 | 0.07 | 0.19 | 0.24** | 0.24** | 0.26** | −0.14* | 0.60** | −0.56** | −0.54** | −0.68** |
| **Time 3 (n = 284)** | | | | | | | | | | | | | |
| 1. normality | | 65.3 | 2.51 | −0.27** | −0.23* | 0.11 | | | | | | | |
| 2. identified | 0.95 | 6.29 | 1.06 | 0.04 | 0.08 | 0.06 | 0.13* | | | | | | |

(*Continued.*)

**Table 1.** (*Continued.*)

| variable | α | M | s.d. | self-isolation | COVID-19 cases | state of health | 1 | 2 | 3 | 4 | 5 | 6 | 7 |
|---|---|---|---|---|---|---|---|---|---|---|---|---|---|
| 3. external | 0.75 | 4.32 | 1.31 | 0.06 | 0.01 | −0.20** | −0.10 | 0.15* | | | | | |
| 4. life structure | 0.80 | 4.24 | 0.94 | 0.03 | 0.21* | 0.23** | 0.02 | 0.16** | −0.11 | | | | |
| 5. depletion | 0.96 | 2.46 | 1.17 | 0.07 | −0.17 | −0.33** | −0.04 | −0.28** | 0.18** | −0.42** | | | |
| 6. isolation | 0.97 | 1.88 | 0.98 | 0.04 | −0.07 | −0.34** | −0.14* | −0.23** | 0.22** | −0.41** | 0.60** | | |
| 7. depression | 0.91 | 2.53 | 0.87 | 0.06 | −0.10 | −0.42** | −0.22** | −0.24** | 0.16** | −0.60** | 0.63** | 0.72** | |
| 8. need satisfaction | 0.83 | 4.78 | 0.87 | −0.03 | 0.12 | 0.27** | 0.12* | 0.23** | −0.12 | 0.61** | −0.40** | −0.52** | −0.66** |
| Time 4 (*n* = 279) | | | | | | | | | | | | | |
| 1. normality | | 68.42 | 19.67 | −0.32** | −0.14 | 0.16** | | | | | | | |
| 2. identified | 0.95 | 6.15 | 1.10 | 0.08 | 0.06 | 0.12* | 0.17** | | | | | | |
| 3. external | 0.79 | 4.12 | 1.33 | 0.01 | −0.16 | −0.21** | −0.08 | 0.08 | | | | | |
| 4. life structure | 0.81 | 4.34 | 0.93 | −0.01 | 0.22* | 0.26** | 0.12* | 0.08 | −0.15* | | | | |
| 5. depletion | 0.96 | 2.35 | 1.16 | −0.05 | −0.19 | −0.30** | −0.19** | −0.18** | 0.18** | −0.41** | | | |
| 6. isolation | 0.97 | 1.92 | 1.03 | 0.04 | −0.03 | −0.36** | −0.20** | −0.18** | 0.28** | −0.42** | 0.63** | | |
| 7. depression | 0.91 | 2.66 | 0.88 | 0.04 | −0.15 | −0.37** | −0.19** | −0.18** | 0.26** | −0.59** | 0.61** | 0.66** | |
| 8. need satisfaction | 0.85 | 4.85 | 0.88 | 0.03 | 0.14 | 0.33** | 0.15* | 0.20** | −0.27** | 0.63** | −0.48** | −0.48** | −0.69** |

*$p < 0.05$, **$p < 0.01$, ***$p < 0.001$. $\alpha$ = Chronbach's alpha.

### 2.2.4.6. Psychological need satisfaction

We are using nine-item psychological need satisfaction scale [15] to measure the extent to which participants feel free to make choices (autonomy need), feel loved and cared for by people that are important to them (relatedness need), and feel effective and competent in their daily life (competence need). A similar measure of psychological need satisfaction has been validated on Chinese participants and shows good internal reliabilities and convergent validity with other well-being constructs [16]. In consideration of space, we used the shortened nine-item version, and combined all three needs into one single construct. Example items reflecting autonomy need satisfaction are 'I felt free to be who I am' and 'I felt pressured to do certain things or be certain ways' (reverse coded). Examples of items reflecting relatedness need satisfaction are 'I felt loved and cared about' and 'I felt a lot of distance from others' (reverse coded). Examples of items reflecting competence need satisfaction are 'I felt like a competent person' and 'I felt inadequate or incompetent' (reverse coded). We asked the participants to rate how true each of the items are to them in the past two weeks on a 7-point scale, ranging from 1 = not at all true to 7 = very true. All nine items for basic psychological need satisfaction were combined into the same variable. At Time 1, the internal consistency for the nine-item psychological need satisfaction measure was 0.81.

## 3. Hypotheses

### 3.1. QUESTION 1. Without a vaccine for COVID-19, the risk of infection remains. Will we see a change in motivation for self-isolation after lockdown is lifted?

#### 3.1.1. Motivation for self-isolating

*Hypothesis 1.1.* Stringent measures have been taken in Wuhan, where the Chinese government has enforced lockdown rather than making it voluntary like other places. In fact, it was reported that Wuhan went under the most stringent lockdown in the world. In that case, after lockdown is lifted, we expected that external motivation (self-isolating due to being forced or fear of punishments) would decrease over time.

*Hypothesis 1.2.* Wuhan has observed a steady low number of confirmed cases after lockdown is lifted. By 24 April, the number of confirmed cases in Wuhan is 47. Therefore, it is possible that identified motivation for self-isolation (see the importance in self-isolating) would decrease over time.

### 3.2. QUESTION 2. As people are allowed to travel and get back to work, people are likely to go back to setting daily goals and having specific tasks to focus on. Will we see a change in structure of daily life after lockdown is lifted?

#### 3.2.1. Structure of daily life

*Hypothesis 2.1.* Inability to carry out normal daily tasks would be one of the main life disruptions during lockdown. So we expected that perceived structure would increase after lockdown is lifted.

### 3.3. QUESTION 3. With the promise of return to normality after lockdown is lifted, should we expect that quality of life will become 'better' after lockdown is lifted compared with during lockdown?

*Hypothesis 3.1.* An optimistic prediction would be that psychological well-being would go up after lockdown is lifted with the promise of life returning to normality. One study showed that the prevalence of anxiety symptoms among Middle East respiratory syndrome (MERS) patients went from 47.2% to 19.4% four to six months after they were removed from isolation [17]. Among those who were isolating due to contact with the patients but did not contract the illness, the prevalence of anxiety symptoms decreased from 7.6% to 3.0%. This was the only study to our knowledge that tracked change in symptoms between time in isolation and time after removal from isolation. Based on that finding, we should expect psychological well-being to increase and ill-being decrease after lockdown lift. As such, from the perspective that self-isolation and life disruptions during lockdown causes a significant impact on psychological well-being, it was expected that when those burdens are

removed after lockdown lift, psychological well-being should increase. In other words, this should mean that psychological need satisfaction (i.e. autonomy, relatedness, competence) would increase, while loneliness and depression would decrease over time after lockdown was lifted.

*Hypothesis 3.2.* Nonetheless, other considerations could also lead to a competing hypothesis. Wuhan residents might experience a relief in symptoms and boost in well-being at Time 1 when they were aware that lockdown would soon be lifted, and might show a downward change when facing the negative economic impact of the outbreak. Based on the expectation from the World Bank, 'Significant economic pain seems unavoidable in all countries and the risk of financial instability is high' [18]. On this basis, both central and local governments have introduced a series of economic measures to revive the economy and promote consumption in Wuhan. However, the spread of the virus around the world is also creating fears of a global recession, which further decreases the demand for Chinese products. Thus, despite that the optimism might be heightened at Time 1 when lockdown lift was announced, we ought to also anticipate that quality of life could continue to drop as a function of the *economic damage* of this pandemic and its psychological effects. Therefore, from the perspective that the economic impact of lockdown would probably take a toll on people's psychological well-being, it was expected that psychological well-being would decrease. In other words, this should mean that psychological need satisfaction would decrease, while loneliness and depression would increase over time after lockdown was lifted.

## 3.4. QUESTION 4. Which coping strategies used during lockdown contributed to psychological well-being and alleviated ill-being?

Previous research looking at the link between the six coping strategies assessed in this research showed that coping with stress through problem solving and seeking help were linked to lower depressive symptoms for first-year university students [19,20], and correlated with post-traumatic growth among survivors of traumatic events [21,22]. On the other hand, other coping strategies like self-blame, avoidance, fantasizing or rationalization correlated with greater depressive symptoms [19,20]. While self-blame and avoidance have been shown to be associated with greater post-traumatic growth [21], results in relation to the perception of post-traumatic growth should be interpreted with caution. It is important to note that post-traumatic growth does not indicate real change but rather can reflect a coping mechanism whereby an individual who has gone through trauma, retrospectively, reinterprets their experience in positive light [23]. As such, in this study, we relied on evidence suggesting that problem solving and help-seeking pertain to more adaptive coping strategies, whereas self-blame, avoidance, fantasizing and rationalization pertain to less adaptive ones.

*Hypothesis 4.1.* We predicted that higher endorsement of strategies related to problem solving and help-seeking, reported at Time 2 (during lockdown), would be associated with greater initial levels of psychological need satisfaction and lower initial levels of loneliness and depression (measured at Time 1—end of lockdown).

*Hypothesis 4.2.* We predicted that higher endorsement of strategies related to problem solving and help-seeking, reported at Time 2 (during lockdown), would be associated with lower initial levels of psychological need satisfaction and higher initial levels of loneliness and depression (measured at Time 1—end of lockdown).

*Hypothesis 4.3.* We predicted that higher endorsement of strategies related to problem solving and help-seeking, reported at Time 2 (during lockdown), would be linked to an increase in psychological need satisfaction over time and a decrease in loneliness and depression over time.

*Hypothesis 4.4.* We predicted that higher endorsement of strategies related to self-blame, avoidance, fantasizing and rationalization, reported at Time 2 (during lockdown), would be linked to a decrease in psychological need satisfaction and an increase in loneliness and depression.

## 3.5. QUESTION 5. Which stressors experienced during lockdown undermined psychological well-being and contribute to ill-being?

In considering which stressors are likely to more indicative of levels and change in well-being and ill-being after lockdown, we rely on a recent report by the Cheung Kong Graduate School of Business (https://english.ckgsb.edu.cn/blog/psychological-resilience-before-and-after-work-resumption-during-covid-19-episode-1-tracking-work-resumption/) surveying close to 6000 entrepreneurs and employees after people went back to work in China. The data suggest that a majority of participants reported

stress related to this pandemic (58.66%) and financial problems (4.19%). Therefore, we predicted that financial stressors (e.g. difficulties with family finances, frustration with work, salary and career disruptions), and health-related stressors (e.g. death of close one, one's own health issues), experienced during lockdown, are likely to be two major predictors of levels and change in well-being and ill-being after lockdown is lifted.

*Hypothesis 5.1.* We predicted that greater intensity of financial stress and health-related problems, reported at Time 2 (during lockdown), would be associated with lower initial levels of psychological need satisfaction and higher initial levels of loneliness and depression (measured at Time 1—end of lockdown).

*Hypothesis 5.2.* We predicted that greater intensity of financial stress and health-related problems, reported at Time 2 (during lockdown), would be linked to a decrease in psychological need satisfaction and an increase in loneliness and depression over time.

# 4. Procedures

The present research uses a within-subject design to understand the self-isolation experience of people who have been isolated because of the COVID-19 outbreak in Wuhan city, China. Longitudinal study was designed to be conducted with a two-week interval, and data were collected four times. There is no randomization. This study is based on observational data.

## 4.1. Sampling method

This study used convenience and snowball sampling techniques by recruiting participants via the WeChat app, a Chinese 'super' app which combines multi-purpose messaging, social media and mobile payment functions developed by Tencent. WeChat is the most popular social media platform in China. By October 2018, this platform had 1056 billion monthly active users, and penetrated 79% of mobile phone users in China by January 2019. Besides, 98.5% of 50- to 80-year-old smartphone users in China used WeChat by September 2018, according to a *technode* article [24]. This would allow us to get access to the senior population.

## 4.2. Survey design platform

We used the *Wenjuanxing* online platform to upload the participant information sheet, privacy notice, consent form, questionnaires and debriefing sheet. Because all the items in the questionnaires were set as required questions, participants need to answer all the questions before moving to the next page. Thus, there was no missing data within each submitted survey. At the same time, the participants could withdraw the survey at any time without giving a reason; on this basis, the unfinished questionnaires would be labelled as invalid and not be included into this study. Participants who completed the survey of each wave received 6 Chinese Yuan (approximate 0.85 USD).

## 4.3. Inclusion criteria

For this study, we only recruited adults over 18 who were living inside Wuhan during the lockdown period. As such, two inclusion criteria are location and age.

### 4.3.1. Location

According to the aim of the research, we used the function of 'restrict location' in the Wenjuanxing platform, which only allows those people who live in the Wuhan city to get access to the survey. This function was only used in the Time 1 data collection, because during Time 1 survey, Wuhan was still under lockdown. While for Time 2, 3 and 4, the location was not restricted, because people could travel around the country (or the world) after the lockdown was lifted.

Participants were recruited through WeChat groups and WeChat moments. For WeChat groups, a research assistant, a Wuhan citizen, posted Time 1 survey link in different groups that were organized spontaneously by Wuhan citizens to exchange information about purchasing daily supplies. All the members in groups used their nickname and there was no personal information disclosed within the group communication. Additionally, using WeChat moments, our research assistant shared the survey link on the timeline of a unique WeChat account specifically created for this research study. At the end of Time 1 survey, we also encouraged participants to share this research link with their families and friends.

## 4.3.2. Age

There were two approaches to guarantee that all the participants are adults. First, in the recruitment advert that we posted in the WeChat group and WeChat moment, we emphasized that participants should only be over the age of 18. Second, the function of 'restrict age' in the Wenjuanxing platform was used to filter the participants under the age of 18. The questionnaires with the participants under the age of 18 were identified as invalid and excluded from the data collection. We did not find any participants under the age of 18.

## 4.4. Process of ensuring data anonymity

Even though participants have provided us their WeChat IDs to be invited back for later surveys, to ensure anonymity, we also collected WeChat aliases to match participants' data. This is a preferred approach to protect anonymity because WeChat aliases are nicknames that users can choose for themselves and they are not connected to their phone number or any other personal information like WeChat IDs. All WeChat aliases are unique in this sample. When we combined data across four waves, we also checked participants' gender and age in each wave to ensure we match the data correctly. Once we finished data collection and matched the data, all WeChat aliases were removed to ensure that the data we shared on the Open Science Framework are completely anonymized.

## 4.5. Method of data collection

This study was designed as a longitudinal research, with four waves of data collection and the interval is two weeks between each wave. Time arrangement is shown in table 2.

The first wave of data was obtained during the period of Wuhan lockdown. By the time Wuhan was lifted from lockdown on 8 April 2020, and we had already collected data from 462 individuals by the end of 7 April, which met our sample size estimation. Thus, we decided to terminate data collection.

We used WeChat to directly track and connect with the participants who completed Time 1 surveys. There are two ways we offered to reach each individual. First, the participants were encouraged to leave their WeChat ID for us to contact. The second method involves giving the participants our WeChat ID (created specifically for this study) so the participants could add us to their contact list voluntarily. With these two approaches, we were able to reach 342 participants at Time 2.

As seen in the diagram above, at the Time 1, 342 participants allowed us to contact them again at Time 2. Out of these 342 participants, 292 returned to complete the survey at the Time 2. Twenty-five new participants joined the study at this wave because they found the survey link that was shared on WeChat groups and WeChat moments at Time 1. However, because the survey link was updated at every time point to allow enrolled participants conveniently to find the link to complete later surveys, any new participants that joined in at later time points completed later surveys rather than starting with Time 1 survey. While we continue to include those new participants in our contact list to invite them to complete later surveys, data from those participants were not analysed. This decision was because new participants that joined in later did not have the same baseline (before lockdown lift) as those who were recruited at Time 1, and change from before lockdown lift to after lockdown lift was important to our research questions. In total, there were 38 participants that joined in at later assessments (25 at Time 2, seven at Time 3 and six at Time 4). Removing these participants, only those with IDs that could be matched with Time 1 IDs were included in the analyses, including 292 participants at Time 2, 284 at Time 3 and 279 at Time 4.

## 4.6. Participants

We recruited 462 participants via WeChat apps at Time 1. Only those who allowed us to contact them for later surveys were invited to come back. In total, 313 participants returned to complete at least one of the later surveys. We provide demographic description of the current sample at each time point in appendix A. In the sample recruited at Time 1 that returned to complete later surveys, participants' average age is 36.16 (s.d. = 13.31, median = 34). There are 110 men and 203 women. The majority of the sample report self-isolating in response to the coronavirus outbreak (79.20% answered yes, 2.60% answered in part and 18.30% answered no). Percentages of participants that reported yes to self-isolating went down at Time 2 (67.80%), Time 3 (47.50%) and Time 4 (38.70%) (see appendix A and figure 3). At all four waves of data, a

**Table 2.** Time arrangement and procedure.

| | Time 1 | Time 2 | Time 3 | Time 4 |
|---|---|---|---|---|
| date | 3–7 April | 18–22 April | 6–10 May | 25–29 May |
| event | under the lockdown | two weeks after | four weeks after | six weeks after |
| procedure | Posted the survey in the WeChat group and moment; 462 participants completed the online survey, 342 of them voluntarily contacted us. | Contacted 342 participants, 292 people replied to us and finished the survey, 25 participants were new at this wave. | Contacted 367 participants (this includes 342 contacts collected after Time 1 and 25 new contacts collected after Time 2); 301 people recruited from previous waves and finished the survey, and there were seven new participants at this wave. | Contacted 374 participants (this includes 342 contacts collected after Time 1, 25 new contacts after Time 2, and seven new contacts after Time 3); 300 people recruited from previous waves and finished the survey, and there were six new participants at this wave. |
| sample size | 462 | 318 | 308 | 306 |
| number of participants that can be matched with Time 1 | | 292 | 284 | 279 |

Note: Wuhan's lockdown was lifted on the 8 April 2020.

notable proportion have reported knowing at least one person who has been confirmed to be infected by the virus (33.50% at Time 1, 31.00% at Time 2, 34.50% at Time 3 and 33.20% at Time 4).

It is important to note that the majority of our sample is in good health at Time 1, with 43.10% of the sample that reports being in very good health and 43.80% that reports being in good health. Because we also relied on convenience and snowball sampling to recruit participants, the data presented in this study is vulnerable to self-selection bias. We discuss this limitation in the Discussion section.

## 4.7. Drop-out analysis

We conducted the $\chi^2$-test to examine whether there was an association between demographics and voluntarily joined the following study at Time 1 (N drop-out = 120, N remain = 342). This allows us to examine whether there were particularly higher rates of drop-out in any demographic groups. It was shown that there was no significant association between drop-out and gender, $\chi^2_1 = 0.001$, $p = 0.98$, health condition, $\chi^2_5 = 2.68$, $p = 0.75$ and infection by virus, $\chi^2_1 = 0.09$, $p = 0.77$. Significant relationships were found between drop-out after Time 1 and marital status, $\chi^2_6 = 17.33$, $p < 0.01$, with more single individuals willing to be invited back for later surveys instead of married ones; and between drop-out and employment status, $\chi^2_7 = 16.68$, $p = 0.02$, with more students rather than retired

people willing to leave contacts to be followed up again at later time points. We considered this limitation of our longitudinal sample in the Discussion section.

## 4.8. Missing data analysis

$\chi^2$-tests were conducted to examine missing data at later time points. We considered only participants that filled out Time 2, Time 3 and Time 4 surveys (not including those who left contacts but did not fill out surveys) and can be matched with Time 1. In other words, new participants at later assessments were not included into these analyses. This allows us to understand whether participants from certain demographic groups might be more likely to return to complete later surveys. The proportion of missing data at Time 2 (N missing = 170, N remaining = 292) does not differ by gender, $\chi^2_1 = 0.02$, $p = 0.88$, general state of health, $\chi^2_5 = 2.77$, $p = 0.74$ and infection of coronavirus, $\chi^2_1 = 1.18$, $p = 0.28$. However, a significant relationship was found between the proportion of missing data and marital status, $\chi^2_6 = 17.89$, $p < 0.01$, with married individuals' data lost at Time 2. Employment status was also significantly associated with missing data, $\chi^2_7 = 19.49$, $p < 0.01$, with some full-timers lost at Time 2.

At Time 3 (N missing = 178, N remaining = 284), compared with Time 1, there is no evidence that the missing data were associated with gender, $\chi^2_1 = 0.17$, $p = 0.68$, general state of health, $\chi^2_1 = 2.02$, $p = 0.85$, infection of coronavirus, $\chi^2_1 = 1.01$, $p = 0.32$, marital status, $\chi^2_6 = 9.48$, $p = 0.15$ and employment status, $\chi^2_7 = 1.01$, $p = 0.19$.

At Time 4 (N missing = 184, N remaining = 279), compared with Time 1, there was no evidence that the missing data were associated with gender, $\chi^2_1 = 0.03$, $p = 0.86$, general state of health, $\chi^2_5 = 1.64$, $p = 0.90$, infection of coronavirus, $\chi^2_1 = 0.91$, $p = 0.34$, marital status, $\chi^2_6 = 1.53$, $p = 0.10$ and employment status, $\chi^2_7 = 11.55$, $p = 0.12$.

## 4.9. Data exclusion

We did not use any exclusion criteria in this study. Due to the modest sample size, we planned to take full advantage of all available data using full-information maximum likelihood (FIML) to estimate models with missing data. We also planned to continue reporting demographic make-up at each assessment to allow assessment of whether certain demographic groups might have higher attrition rates than others and allow for assessment of generalizability of the presented findings.

## 4.10. Quality check

In the surveys administered after lockdown lift (Time 2–4), we asked the participants to answer the question 'Think about your life before the COVID-19 outbreak, how much of your life has returned to normal?' on a scale between 0 and 1. We should expect that the percentage of life returning to normality to increase between Time 2, Time 3 and Time 4.

## 4.11. Power analysis

We anticipated that we would get at least 200 participants whose data could be matched with Time 1 data. We relied on previous recommendations using Monte Carlo simulations to estimate effect sizes that could be detected with type I error rates set at 0.05 and power of 0.80 or above for a sample of 200. According to Fan & Fan [25], a sample of 200 allows us 0.80 or greater power to detect a linear growth of a small effect size ($d = 0.20$) between time points. According to Diallo et al. [26], a sample of at least 200 allows us 0.80 or greater power to detect a quadratic growth of a small effect size ($R^2 = 0.30$, slope = 0.30). Assuming we achieve growth curve reliability greater than 0.90 or more (suggesting that measures across time points correlate with one another at greater than 0.90), according to Hertzog et al. [27], a sample of at least 200 allows us 0.80 or greater power to detect a correlation between a covariate and slope of ±0.6 or above.

Our final sample is 313, which is bigger than what we anticipated. We used FIML to account for missing values so we can take advantage of the whole sample. Using LIFESPAN [28] to estimate achieved power based on the observed parameters, we had sufficient power (greater than 0.80) to detect significant parameters in the slope models (Model 4) for identified motivation, external motivation, life structure, depressive symptoms and need satisfaction, but not for isolation and depletion.

## 4.12. Data analytic plan

We conducted latent growth models (LGMs) to examine change across time. Previous research suggested that LGMs allow for higher statistical power to detect growth than dependent *t*-tests and repeated-measures ANOVAs, particularly under conditions where there is minimal growth with small to moderate sample sizes [25]. All our hypotheses concerned changes across four time points for external and identified motivation for self-isolation (Hypotheses 1.1 and 1.2), perceived life structure (Hypothesis 2.1), psychological need satisfaction, loneliness and depressive symptoms (Hypotheses 3.1. and 3.2). Further, we considered whether certain coping strategies or types of stressors might predict different levels of well-being and ill-being outcomes (Hypotheses 4.1 and 4.2) and different slopes of change over time (Hypotheses 4.3, 4.4, 5.1 and 5.2). We took the following steps to conduct latent growth models to determine developmental growth trajectories of each dependent measures:

**Model 1.** First, we conducted a fully constrained latent intercept model, not allowing the intercept to vary either between or within individuals. This model, therefore, assumed that there is no difference between people and no change across time. In this model, only the intercept mean and residual variance were estimated, and the residual variance was set to be fixed for all time points. Setting the residual variance to be fixed for all time points assumed that error around each time point is the same.

**Model 2.** In the second model, we allowed the intercept to vary across participants. This means that we allowed for people to have different averages on the dependent variables, yet we still assume no change over time. In this model, only the intercept mean, intercept variance and residual variance were estimated. The residual variance was set to be fixed for all time points.

**Model 3.** In the third model, we began to estimate slope variance, but not yet determine a slope intercept. We added a linear slope into the model (0 1 2 3) but still restrict slope mean to 0. This means we assumed that people might start off with different averages on the dependent variables at Time 1, and might vary in how they respond to each survey across time, but overall there is still no change. In this model, the intercept mean, intercept variance, slope variance and residual variance were estimated. The residual variance was set to be fixed for all time points.

**Model 4.** In the fourth model, we allowed both latent intercept and latent linear slope to vary. We removed the restriction on slope mean. This means we allowed for people to start off with different averages on the dependent variables at Time 1, and to change across four time points in varied ways. In this model, the intercept mean, intercept variance, slope mean, slope variance and residual variance were estimated. Additionally, we also let the latent intercept and the latent linear slope covary with one another; this allows us to see whether people that start off at different averages on the dependent variables might display different slopes of change over time. The residual variance was set to be fixed for all time points.

Because this model allows us to examine whether there is change across time, it allows us to test Hypotheses 1.1, 1.2, 2.1, 3.1 and 3.2. If this model does not improve fitness significantly compared with Model 3 (using $\chi^2$ to compare models) this suggests that there is not significant linear change across time. If this model's fitness is significantly improved compared with Model 3, a positive coefficient of slope intercept will indicate a linear increase and a negative coefficient of slope intercept will indicate a linear decrease.

**Model 5.** The fifth model was a fully unconstrained model, as we now removed restrictions on residual variance for all time points. This allows for different errors around each time point. This model allows us to observe whether the variability of each dependent variable is unequal across time points.

**Model 6.** We added a new latent quadratic slope (0 1 4 9) to Model 5 to see if there is a nonlinear trend. We allowed this quadratic slope to have a residual variance that freely covaried with the latent intercept and the latent linear slope. We planned that, if this model did not significantly improve model fitness compared with Model 5, we would continue on to the next step with the linear slope model. This model allows us to further clarify Hypotheses 1.1, 1.2, 2.1, 3.1 and 3.2 and see whether change over time might happen nonlinearly.

**Model 7.** After we determined the optimal growth model where all parameters are freely estimated, the conditional LGM was estimated to further test whether certain predictors contributed to explain the intercepts and slopes in psychological need satisfaction, loneliness and depression. In other words,

**Table 3.** Summary of model building and fitness indices.

| Model | 1 | 2 | 3 | 4 | 5 | 6 | 7a | 7b |
|---|---|---|---|---|---|---|---|---|
| intercept mean | x | x | x | x | x | x | x | x |
| intercept variance | | x | x | x | x | x | x | x |
| residual variance (fixed) | x | x | x | x | | | | |
| residual variance (free) | | | | | x | x | x | x |
| linear slope mean | | | | x | x | x | x | x |
| quadratic slope mean | | | | | | x | (x) | (x) |
| linear slope variance | | | x | x | x | x | x | x |
| quadratic slope variance | | | | | | x | (x) | (x) |
| intercept—linear slope covariance | | | | x | x | x | x | x |
| intercept—quadratic slope covariance | | | | | | x | (x) | (x) |
| linear—quadratic slope covariance | | | | | | x | (x) | (x) |
| coping strategies predicting intercept | | | | | | | x | |
| coping strategies predicting slope | | | | | | | x | |
| stressors predicting intercept | | | | | | | | x |
| stressors predicting slope | | | | | | | | x |
| $\chi^2$(d.f.) | | | | | | | | |
| RMSEA | | | | | | | | |
| SRMR | | | | | | | | |
| CFI | | | | | | | | |
| change in $\chi^2$ | | | | | | | | |

Notes: x indicates that the parameter is estimated in the model; (x) indicates that the parameter is only estimated if the previous model improves fitness significantly.

we examined whether the initial level and trajectory of well-being and ill-being were conditional on any coping strategies and stressors that participants experienced during lockdown.

(a) To examine how different coping strategies that participants used during lockdown predict varied initial levels of well-being and ill-being at Time 1 and their changes over time, we simultaneously added six coping strategies, including problem solving, help-seeking, self-blame, avoidance, fantasizing and rationalization, measured at Time 2 (during lockdown) as time-invariant covariates to predict the intercepts and slopes of psychological need satisfaction, loneliness and depression. This model allows us to simultaneously test Hypotheses 4.1, 4.2, 4.3 and 4.4, and evaluate between-person differences in coping strategies during lockdown on the stability or change of the outcome variables over time.

(b) To examine how different stressors that participants experienced during lockdown predict varied initial levels of well-being and ill-being at Time 1 and their changes over time, we simultaneously added six stressors, including marriage and romantic relationship problems, family problems, interpersonal relationship problems, health-related problems, problems with work and finance, and daily disruptions, measured at Time 2 (during lockdown) as time-invariant covariates to predict the intercepts and slopes of psychological need satisfaction, loneliness and depression. This model allows us to simultaneously test Hypotheses 5.1 and 5.2, and evaluate between-person differences in life stressors during lockdown on the stability or change of the outcome variables over time.

Analyses were conducted in R program using 'Lavaan' package. All annotated and reproducible codes are shared on Open Science Framework.

## 4.13. Handling missing data

As reported above, from our missing data analysis, marital and employment statuses were the two characteristics that were linked to whether or not someone returned to complete later surveys. This

# Table 4. Registered report plan at Stage 1 and the findings at Stage 2.

| question | hypothesis | sampling plan (e.g. power analysis) | analysis plan | interpretation given different outcomes | summary of results |
|---|---|---|---|---|---|
| **QUESTION 1. Without a vaccine for COVID-19, the risk of infection remains. Will we see a change in motivation for self-isolation after lockdown is lifted?** | **Hypothesis 1.1.** External motivation (self-isolating due to being forced or fear of punishment) will decrease over time. | According to Xitao & Xiaotao [25], a sample of 200 will allow us 0.80 or greater power to detect a linear growth of a small effect size (*d* = 0.20) between time points. According to Diallo *et al.* [26], a sample of at least 250 will allow us 0.80 or greater power to detect a quadratic growth of a small effect size ($R^2 = 0.3$, slope = 0.30). Assuming we achieve growth curve reliability of 0.90 or more (suggesting that measures across time points correlate with one another at 0.91) according to Hertzog *et al.* [27], a sample of at least 200 will allow us 0.80 or greater power to detect a correlation between a covariate and slope of ±0.6 or above. Therefore, at Time 4, we will try to recruit at least 200 returning participants whose data can be matched across four time points. And we will use FIML to account for missing values so we can take advantage of the whole sample of 462. | **Model 1.** Only the intercept mean and residual variance will be estimated, and residual variance will be set to be fixed for all time points. **Model 2.** The intercept mean, intercept variance and residual variance will be estimated. The residual variance is set to be fixed for all time points. **Model 3.** The intercept mean, intercept variance, slope variance and residual variance will be estimated. The residual variance is set to be fixed for all time points. **Model 4.** the intercept mean, intercept variance, linear slope mean (0 1 2 3), slope variance and residual variance will be estimated. The residual variance is set to be fixed for all time points. **Model 5.** We will conduct a fully unconstrained model, as we now will remove restrictions on residual variance for all time points. **Model 6.** Finally, we will add a new latent quadratic slope (0 1 4 9) to Model 5 to see if there is a nonlinear trend. We will allow this quadratic slope to have a residual variance and the latent linear slope. **Model 7a.** Once an optimal growth model is determined (linear versus quadratic), we will add six coping strategies measured at Time 1 as covariates to predict the intercepts and slopes of need satisfaction, loneliness and depression. **Model 7b.** Once an optimal growth model is determined (linear versus quadratic), we will add six stressors measured at Time 1 as covariates to predict the intercepts and slopes of need satisfaction, loneliness and depression. | If this model shows sufficient fit, this suggests that there is little variance between individuals or across time points. If this model shows significantly better fit than Model 1, this suggests that participants differ in their averages on the dependent variable. If this model shows significantly better fit than Model 2, this suggests that participants start off at different levels on the dependent variables and fluctuate over time in varied ways. If this model shows significantly better fit than Model 3, this suggests that participants start off at different levels on the dependent variables, and show significant linear change over time in varied ways. If the estimate of intercept-linear slope covariance is significant at *p* < 0.05, this means that those starting off at different averages on the dependent variables display different slopes over time. This model will allow us to test Hypotheses 1.1, 1.2, 2.1, 3.1 and 3.2). If this model shows significantly better fit than Model 4, this suggests that variability of each dependent variable is unequal across time points. We will discuss this issue of heteroscedasticity in the Discussion. If this model shows significantly better fit than Model 5, this suggests that a quadratic trend fits the data better than a linear trend. If this model yields significantly better fit than the unconditional model (Model 5 or Model 6), this suggests that intercepts and slopes of need satisfaction, loneliness and depression depend on different levels of coping strategies used during lockdown (measured at Time 2). We will rely on parameter estimates to determine which coping strategy significantly predicts either the initial levels of the outcome or the slope of change in outcome over time (Hypotheses 4.1–4.4). If this model yields significantly better fit than the unconditional model (Model 5 or Model 6), this suggests that intercepts and slopes of need satisfaction, loneliness and depression depend on different levels of stressors experienced during lockdown (measured at Time 2). We will rely on parameter estimates to determine whether health-related or finance-related stressors significantly predict either the initial levels of the outcome or the slope of change in outcome over time (Hypotheses 5.1 and 5.2). | **Hypotheses 1.1 and 1.2 were supported.** Both external and identified motivation for self-isolation decreased over time. |
| | **Hypothesis 1.2.** Identified motivation for self-isolation (self-isolating because one sees its importance in protecting oneself and other) will decrease over time. | | | | |
| **QUESTION 2. As people are allowed to travel and get back to work, people are likely to go back to setting daily goals and having specific tasks to focus on. Will we see a change in structure of daily life after lockdown is lifted?** | **Hypothesis 2.1.** Life structure will increase over time after lockdown is lifted. | | | | **Hypothesis 2.1 was supported.** Life structure *slightly increased* over time after lockdown is lifted. |
| **QUESTION 3. With the promise of return to normality after lockdown is lifted, should we expect that quality of life will become 'better' after lockdown is lifted compared with during lockdown?** | **Hypothesis 3.1.** Basic psychological needs will increase, while loneliness and depressive symptoms will decrease over time. | | | | **Hypothesis 3.1 was supported.** Basic psychological needs *slightly increased*, while loneliness and depressive symptoms *slightly decreased* after lockdown was lifted. |
| | **Hypothesis 3.2.** Basic psychological needs will decrease, while loneliness and depressive symptoms will increase over time. | | | | |
| **QUESTION 4. Which coping strategies used during lockdown will contribute to psychological well-being and alleviate ill-being?** | **Hypothesis 4.1.** We predict that higher endorsement of strategies related to problem solving and help-seeking, reported at Time 2 (during lockdown), will be associated with greater initial levels of psychological need satisfaction and lower initial levels of loneliness and depression (measured at Time 1—end of lockdown). | | | | **Hypotheses 4.1 and 4.2 were supported, while hypotheses 4.3 and 4.4 were not supported.** The higher endorsement of problem solving and help-seeking strategies were associated with greater levels of psychological well-being at the end of lockdown. The higher endorsement of strategies related to self-blame, avoidance, fantasizing and rationalization were associated with lower levels of psychological well-being at the end of lockdown. In general, our data *did not support* the relationships between the endorsement of coping strategies during lockdown with rates of *change* in psychological well-being after lockdown was lifted. |
| | **Hypothesis 4.2.** We predict that higher endorsement of strategies related to self-blame, avoidance, fantasizing and rationalization, reported at Time 2 (during lockdown), will be associated with lower initial levels of psychological need satisfaction and higher initial levels of loneliness and depression (measured at Time 1—end of lockdown). | | | | |
| | **Hypothesis 4.3.** We predict that higher endorsement of strategies related to problem solving and help-seeking, reported at Time 2 (during lockdown), will be linked to an increase in psychological need satisfaction over time and a decrease in loneliness and depression over time. | | | | |
| | **Hypothesis 4.4.** We predict that higher endorsement of strategies related to self-blame, avoidance, fantasizing and rationalization, reported at Time 2 (during lockdown), will be linked to a decrease in psychological need satisfaction and an increase in loneliness and depression over time. | | | | |
| **QUESTION 5. Which stressors experienced during lockdown will undermine psychological well-being and contribute to ill-being?** | **Hypothesis 5.1.** We predict that greater intensity of financial stress and health-related problems, reported at Time 2 (during lockdown), will be associated with lower initial levels of psychological need satisfaction and higher initial levels of loneliness and depression (measured at Time 1—end of lockdown). | | | | **Neither hypotheses 5.1 nor 5.2 were supported.** *Daily disruptions* during lockdown, instead of financial stress or health-related problems, was associated with lower levels of psychological well-being reported at the end of lockdown. However, none of the stressors experienced during lockdown was associated with notably different rates of change after lockdown was lifted. |
| | **Hypothesis 5.2.** We predict that greater intensity of financial stress and health-related problems, reported at Time 2 (during lockdown), will be linked to a decrease in psychological need satisfaction and an increase in loneliness and depression over time. | | | | |

indicates that our data were not missing completely at random (MCAR). Our missingness pattern fit the definition of 'missing at random' as the likelihood of missingness depended on the measured characteristics of the participants [29].

We used FIML, which is a robust technique to handle missing data when data are missing at random [30,31], without any changes to the data. FIML allowed us to carry out our analyses with all available raw data while simultaneously accounting for missing data and estimating parameters and errors. Further, ML procedure has been shown to yield less biased estimates in the case where the assumption of multivariate normality is violated [32].

## 4.14. Summary of model building and fitness indices

By following the above planned-out steps, we were able to observe how model fitness improves across models as model restrictions are removed. Results are expressed in table 3.

In the current study, the comparative fit index (CFI), with values greater than 0.90, the standardized root mean residual (SRMR), with values less than 0.05, the root mean squared error of approximation (RMSEA), with values less than 0.08, are indicative of well-fitting models. The likelihood ratio $\chi^2$ is also reported to compare model fitness.

The Stage 1 registered report plan and the summary of the stage 2 results are shown in table 4.

# 5. Results

## 5.1. Tests for multivariate normality

We entered sets of four measures of all dependent variables assessed at four time points (i.e. identified motivation, external motivation, life structure, depletion, isolation, depression, need satisfaction) into multiple tests for multivariate normality, using the mult.norm function in R program. Based on distributions of Mahalanobis's distances, we observed that the assumption of multivariate normality was violated for all our dependent variables, with many of the distributions positively skewed. We proceeded with our LGM using FIML algorithm, which has been shown to yield less biased parameter estimation [32] and also allows us to count cases with missing data across time points without imputation.

## 5.2. Within the first two months after lockdown lift, how much of life has returned to normal?

We asked this question to those that returned to complete surveys at later times. At Time 2 ($n = 292$), two weeks after lockdown lift, on average participants reported 54.78% (s.d. = 23.73) of life has returned to normality. At Time 3 ($n = 284$), a month after lockdown lift, 65.30% (s.d. = 2.51) of life has returned to normality and this increased to 68.42% (s.d. = 19.67) at Time 4 ($n = 279$), six weeks after lockdown lift. This means that whatever changes we reported on our dependent variables in this study reflects a gradual return to normality for this sample (figure 2). At the same time that life is perceived to return to normality, the number of people responding 'yes' to the question of whether people continue to self-isolate due to COVID-19 also dropped over time (figure 3). To the extent that people perceived greater percentage of their life having gone back to normality, they were less likely to report self-isolating (Time 2: $r = -0.23$, $p < 0.01$; Time 3: $r = -0.27$, $p < 0.01$; Time 4: $r = -0.32$, $p < 0.01$). Nonetheless, the likelihood of participants' self-isolating after lockdown was lifted yielded little association with their motivation for self-isolating (table 1), suggesting that whether someone self-isolated after lockdown had little to do with whether they perceived self-isolating to be beneficial or whether they felt there were external pressures to self-isolate.

## 5.3. Within the first two months after lockdown lift, does motivation for self-isolating decrease?

We measured two types of motivation for self-isolating. Identified motivation refers to reasons for self-isolating because one sees the benefits in self-isolating, particularly to protect one's own health as well as important others. On the other hand, external motivation refers to reasons for self-isolating due to external restrictions or pressures. As shown in table 1, these two types of motivation were not related to one another, and participants reported endorsing identified motivation for self-isolating more than

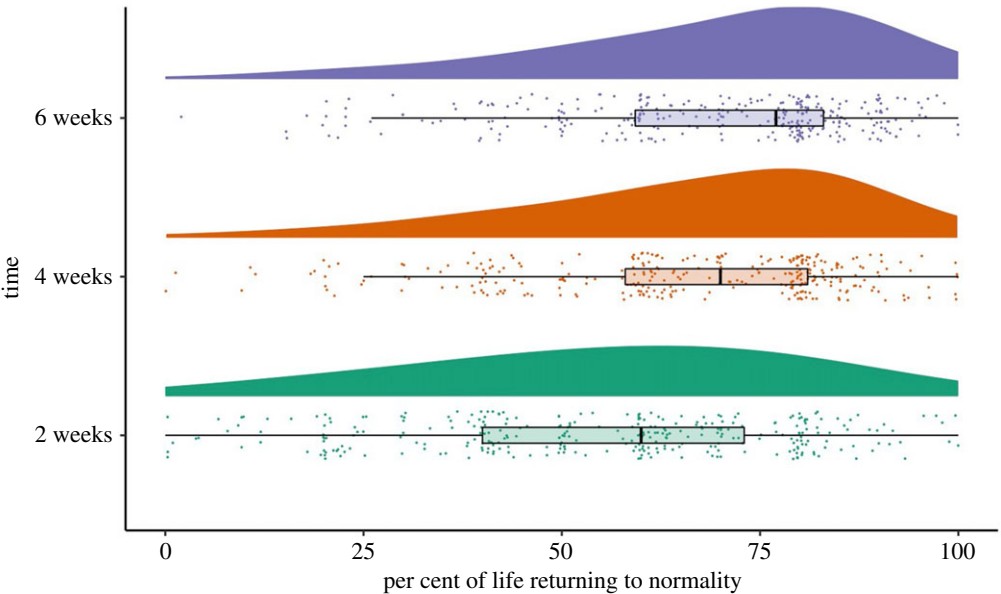

**Figure 2.** Graph illustrating the rate of life returning to normality within the first two months after lockdown.

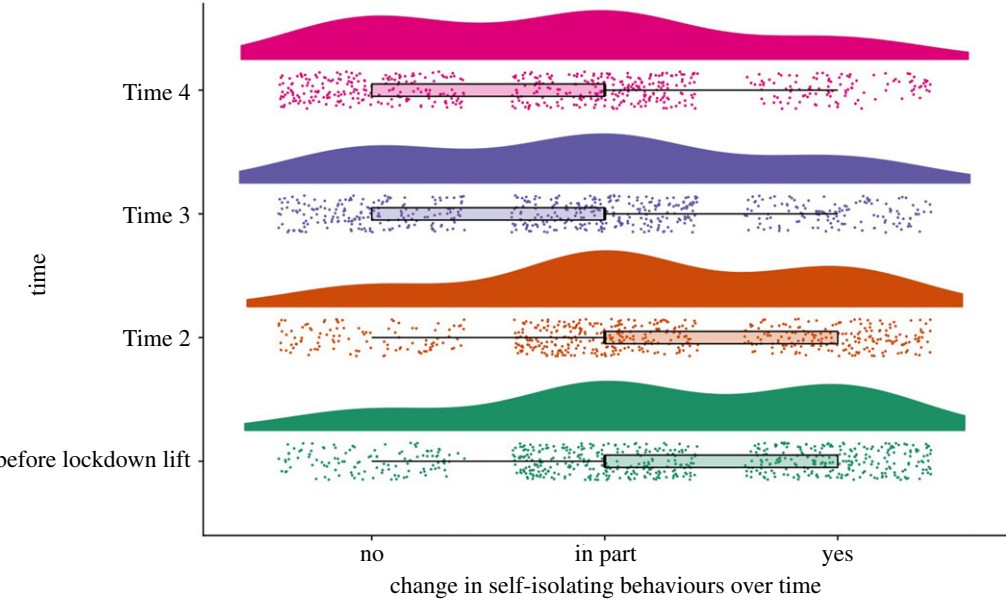

**Figure 3.** Change in distribution of the sample that self-isolate during lockdown and after lockdown.

external motivation, suggesting that on average, participants recognized the benefits of self-isolating and did not just do it because they were forced to or because of government's restrictions.

For both types of motivation, LGM suggested that the models with linear slopes added yielded satisfactory model fit (see tables 5 and 6). Overall, the results suggested that there were decreases in both identified and external motivation at relatively the same rates. Therefore, Hypotheses 1.1 and 1.2 were both supported.

For identified motivation specifically, Model 6 ($\chi^2_1 = 0.21$, RMSEA = 0.00, SRMR = 0.00, CFI = 1.00) yielded the best fit compared with Model 4 and Model 5; nonetheless, the quadratic slope was not significant ($B = 0.02$, $SE.B = 0.02$, $z = 1.10$, $p = 0.27$) (table 5). The linear slope was significant ($B = -0.22$, $SE.B = 0.07$, $z = -3.37$, $p = 0.001$), but change over time in identified motivation was mainly driven by the drop in 'strongly agree' responses. Particularly, as seen in figure 4, at Time 1 data gather around the highest possible response for identified motivation, with the median of 7 for both Time 1 and Time 2. The median drops to 6.8 at Time 3 and 6.6 at Time 4, and data distribution became more spread out (s.d.$_{T1}$ = 0.65, s.d.$_{T2}$ = 0.90, s.d.$_{T3}$ = 1.06, s.d.$_{T4}$ = 1.10). As such, the decrease of the means in

**Table 5.** Models investigating change in identified motivation for self-isolating within the first six weeks after lockdown lift.

| | 1 | 2 | 3 | 4 | 5 | 6 |
|---|---|---|---|---|---|---|
| intercept mean | 6.37*** | 6.38*** | 6.48*** | 6.59*** | 6.60*** | **6.62*** |
| intercept variance | | 0.36*** | 0.18*** | 0.10* | 0.18*** | **0.30** |
| residual variance (fixed) | 0.91*** | 0.55*** | 0.43*** | 0.45*** | | |
| residual variance (free) | | | | | 0.25, 0.52, 0.53, 0.40 | **0.12, 0.37, 0.44, 0.05** |
| linear slope mean | | | | −0.15*** | −0.15*** | **−0.22** |
| quadratic slope mean | | | | | | **0.02** |
| linear slope variance | | | 0.08*** | 0.04** | 0.06*** | **0.68*** |
| quadratic slope variance | | | | | | **0.07*** |
| intercept—linear slope covariance | | | | 0.07*** | 0.03 | **−0.13** |
| intercept—quadratic slope covariance | | | | | | **0.04** |
| linear—quadratic slope covariance | | | | | | **−0.21*** |
| $\chi^2$(d.f.) | 403.57*** | 198.76*** | 104.32*** | 48.06*** | 26.04*** | **0.21** |
| RMSEA | 0.32 | 0.23 | 0.17 | 0.13 | 0.12 | **0** |
| SRMR | 0.45 | 0.37 | 0.19 | 0.12 | 0.06 | **0** |
| CFI | 0 | 0.29 | 0.64 | 0.85 | 0.92 | **1** |
| d.f. | 12 | 11 | 10 | 8 | 5 | **1** |
| change in $\chi^2$ | | 204.81 | 94.44 | 56.26 | 22.02 | **25.83** |
| achieved power | | | | 1.00 | | |

$*p < 0.05$, $**p < 0.01$, $***p < 0.001$. Final model is bolded.

**Table 6.** Models investigating change in external motivation for self-isolating within the first six weeks after lockdown lift.

| | 1 | 2 | 3 | 4 | 5 | 6 |
|---|---|---|---|---|---|---|
| intercept mean | 3.78*** | 4.38*** | 4.45*** | **4.60*** | 4.60*** | 4.61*** |
| intercept variance | | 0.82*** | 0.72*** | **0.72*** | 0.69*** | 1.17*** |
| residual variance (fixed) | 1.62*** | 0.81*** | 0.69*** | **0.69*** | | |
| residual variance (free) | | | | | 0.76, 0.69, 0.74, 0.48 | 0.27, 0.65, 0.63, 0.33 |
| linear slope mean | | | | **−0.16*** | −0.16*** | −0.15 |
| quadratic slope mean | | | | | | −0.00 |
| linear slope variance | | | 0.07*** | **0.05** | 0.06** | 0.97** |
| quadratic slope variance | | | | | | 0.07** |
| intercept—linear slope covariance | | | | **0.01** | 0.02 | −0.58* |
| intercept—quadratic slope covariance | | | | | | 0.15* |
| linear—quadratic slope covariance | | | | | | −0.25** |
| $\chi^2$(d.f.) | 402.26*** | 79.02*** | 50.79*** | **15.70*** | 9.91 | 0.86 |
| RMSEA | 0.32 | 0.14 | 0.11 | **0.06** | 0.06 | 0 |
| SRMR | 0.36 | 0.11 | 0.09 | **0.04** | 0.03 | 0.01 |
| CFI | 0 | 0.82 | 0.89 | **0.98** | 0.99 | 1.00 |
| d.f. | 12 | 11 | 10 | **8** | 5 | 1 |
| change in $\chi^2$ | | 323.24 | 28.23 | **35.09** | 5.79 | 9.05 |
| achieved power | | | | 1.00 | | |

*$p < 0.05$, **$p < 0.01$, ***$p < 0.001$. Final model is bolded.

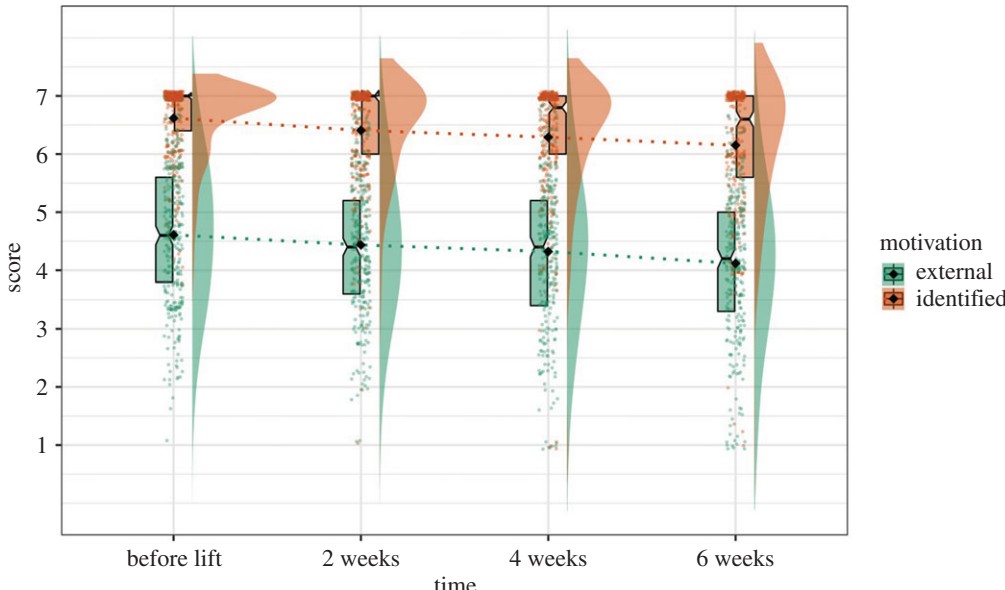

**Figure 4.** Graph illustrating the rates of identified and external motivations decreasing within the first two months after lockdown.

identified motivation over time was mainly due to some data drifting down to the lower-scored responses (see data distribution in figure 4), yet most people continue to be highly motivated to self-isolate to protect their health and their loved ones' health.

For external motivation, Model 4 yielded the best fit ($\chi^2_8 = 15.70$, RMSEA = 0.06, SRMR = 0.04, CFI = 0.98), and setting the residual variance free did not improve the model fit significantly (table 6). Therefore, we chose Model 4 as our final model. Data for external motivation were more normally distributed compared with identified motivation, with most participants scoring around the midpoint (median = 4.6 for Time 1, 4.4 for Time 2 and Time 3, and 4.2 for Time 4). Errors around the means at each time point were relatively equal (s.d.$_{T1}$ = 1.20, s.d.$_{T2}$ = 1.21, s.d.$_{T3}$ = 1.31, s.d.$_{T4}$ = 1.33) and scores gradually decreased over time with more participants choosing the lowest possible responses at Time 3 and Time 4 (figure 4). This means that, after government's lockdown restrictions were lifted for Wuhan, compared with the previous time points, participants perceived fewer external pressures and reported less fear of punishments and consequences for not self-isolating.

## 5.4. Does structure of everyday life improve after lockdown is lifted?

Perceived structure in one's daily life were measured with five self-reported items about the extent to which participants had specific tasks to focus on or had clear goals and plans of how to use their time. According to the LGM, Model 5 which included freely estimated linear slope yielded the best fit ($\chi^2_5 = 1.91$, RMSEA = 0.06, SRMR = 0.03, CFI = 0.99) and showed significant $\chi^2$ improvement compared with Model 4. Therefore, we chose Model 5 as the final model for perceived life structure. On average, compared with previous time points, participants reported improvement in their life structure by 0.05 point on 6-point scale at every later time point (table 7). This suggested that after lockdown was lifted, participants gradually and increasingly perceived their life as having clear goals and plans (figure 5). Therefore, Hypothesis 2.1 was supported, showing small increase in life structure over time after lockdown was lifted.

## 5.5. Does quality of life improve after lockdown is lifted?

### 5.5.1. Basic psychological need satisfaction

We assessed participants' levels of satisfaction of their basic psychological needs for autonomy, relatedness and competence, during lockdown and after lockdown was lifted. We combined all nine items assessing the three needs into one composite. The results from LGM suggested that Model 4 yielded the best fit ($\chi^2_8 = 2.32$, RMSEA = 0.07, SRMR = 0.03, CFI = 0.98) (table 8). Therefore, we chose Model 4 as our final model, which indicates linear increase in basic need satisfaction over time.

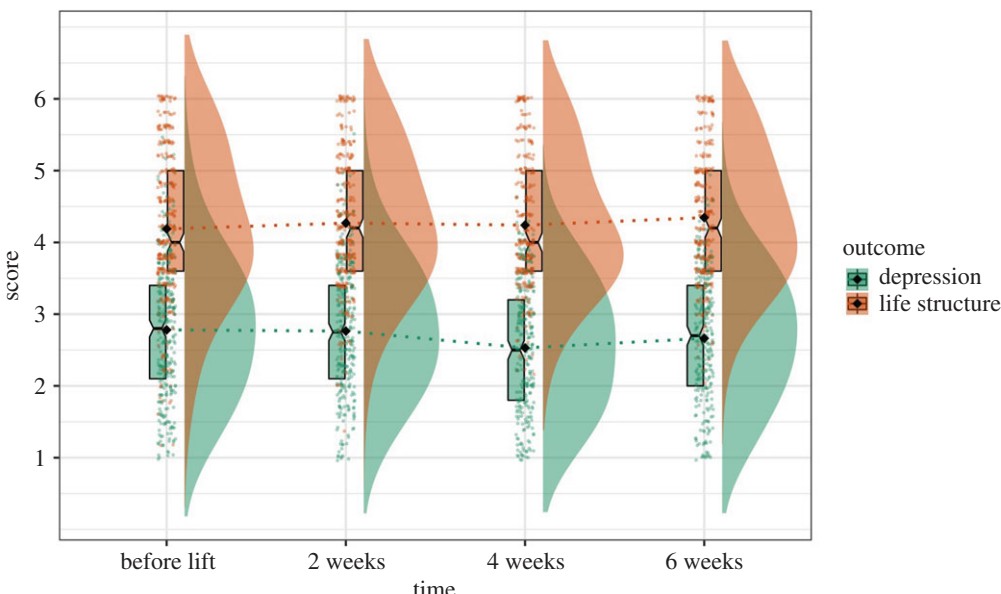

**Figure 5.** Graph illustrating the rates of depressive symptoms and life structure changing within the first two months after lockdown.

**Table 7.** Models investigating change in perceived life structure within the first six weeks after lockdown lift.

| | 1 | 2 | 3 | 4 | **5** | 6 |
|---|---|---|---|---|---|---|
| intercept mean | 4.26*** | 4.26*** | 4.25*** | 4.19*** | **4.19**\*** | 4.20*** |
| intercept variance | | 0.54*** | 0.54*** | 0.72*** | **0.64**\*** | 0.81*** |
| residual variance (fixed) | 0.95*** | 0.40*** | 0.37* | 0.33*** | | |
| residual variance (free) | | | | | **0.52, 0.31, 0.29, 0.25** | 0.31, 0.30, 0.24, 0.23 |
| linear slope mean | | | | 0.05* | **0.05**\* | 0.00 |
| quadratic slope mean | | | | | | 0.02 |
| linear slope variance | | | 0.02* | 0.04*** | **0.03**\*\* | 0.41* |
| quadratic slope variance | | | | | | 0.03* |
| intercept—linear slope covariance | | | | −0.08*** | **−0.05**\* | −0.27 |
| intercept—quadratic slope covariance | | | | | | 0.05 |
| linear—quadratic slope covariance | | | | | | −0.10* |
| $\chi^2$(d.f.) | 511.34*** | 61.59*** | 55.79*** | 29.69*** | **10.91** | 4.29* |
| RMSEA | 0.37 | 0.12 | 0.12 | 0.09 | **0.06** | 0.10 |
| SRMR | 0.39 | 0.08 | 0.11 | 0.04 | **0.03** | 0.02 |
| CFI | 0 | 0.90 | 0.91 | 0.96 | **0.99** | 0.99 |
| d.f. | 12 | 11 | 10 | 8 | **5** | 1 |
| change in $\chi^2$ | | 449.75 | 5.80 | 26.10 | **18.78** | 6.62 |
| achieved power | | | | 1.00 | | |

*$p < 0.05$, **$p < 0.01$, ***$p < 0.001$. Final model is bolded.

**Table 8.** Models investigating change in depletion within the first six weeks after lockdown lift.

| | 1 | 2 | 3 | 4 | 5 | 6 | 7a | 7b |
|---|---|---|---|---|---|---|---|---|
| intercept mean | 2.60*** | 2.60*** | 2.61*** | 2.82*** | **2.82\*\*\*** | 2.85*** | 2.68*** | 1.52*** |
| intercept variance | | 1.01*** | 1.01*** | 1.20*** | **1.17\*\*\*** | 1.31*** | 0.61*** | 0.69*** |
| residual variance (fixed) | 1.60*** | 0.60*** | 0.59*** | 0.55*** | | | | |
| residual variance (free) | | | | | **0.58, 0.62, 0.52, 0.36** | 0.47, 0.81, 0.51, 0.83 | 0.67, 0.56, 0.54, 0.37 | 0.65, 0.58, 0.53, 0.38 |
| linear slope mean | | | | −0.15*** | **−0.15\*\*\*** | −0.28*** | −0.19 | 0.03 |
| quadratic slope mean | | | | | | 0.04* | | |
| linear slope variance | | | 0.00 | 0.01 | **0.02** | −0.22 | 0.01 | 0.01 |
| quadratic slope variance | | | | | | −0.06** | | |
| intercept—linear slope covariance | | | | −0.07* | **−0.06** | −0.21 | −0.01 | −0.01 |
| intercept—quadratic slope covariance | | | | | | 0.03 | | |
| linear—quadratic slope covariance | | | | | | 0.12 | | |
| $\chi^2$(d.f.) | 629.68*** | 103.02*** | 102.87*** | 35.96*** | **27.07\*\*\*** | 10.64** | 49.50*** | 55.98*** |
| RMSEA | 0.41 | 0.16 | 0.17 | 0.11 | **0.12** | 0.18 | 0.08 | 0.09 |
| SRMR | 0.44 | 0.11 | 0.12 | 0.06 | **0.06** | 0.03 | 0.04 | 0.04 |
| CFI | 0 | 0.84 | 0.84 | 0.95 | **0.96** | 0.98 | 0.96 | 0.95 |
| d.f. | 12 | 11 | 10 | 8 | **5** | 1 | 17 | 17 |
| change in $\chi^2$ | | 526.66 | 0.15 | 66.91 | **8.89** | 16.43 | 22.43 | 28.91 |
| achieved power | | | | 0.36 | | | | |

* $p < 0.05$, ** $p < 0.01$, *** $p < 0.001$. Negative variances in Model 6 suggested that the model is mis-specified. As such, even though $\chi^2$-test yielded significantly better fit, we chose Model 5 as the best model; achieved power for slope model is low.

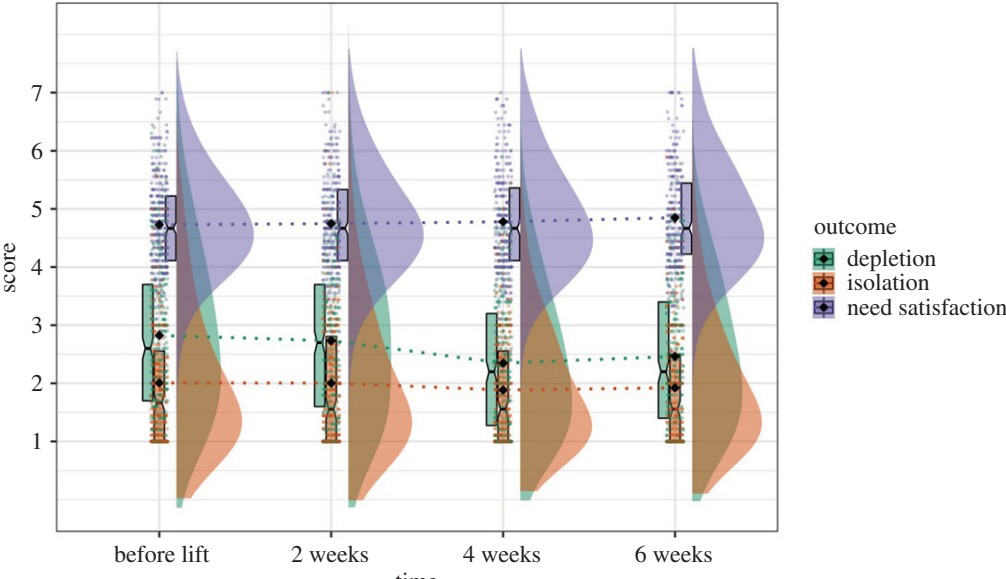

**Figure 6.** Graph illustrating the rates of depletion, isolation and need satisfaction changing within the first two months after lockdown.

However, on average basic need satisfaction increases by 0.04 out of 7-point scale at every later time point, suggesting that change was minimal. As seen in figure 6, errors around the means are relatively equal across time points, and the means generally were above the midpoint of 7-point scales. This indicates that participants' perceptions of how much autonomy, relatedness and competence experienced during and after lockdown did not suffer too much from the extreme measures that have been taken in Wuhan, and need satisfaction appeared to recover, albeit to a very small degree, after lockdown was lifted.

### 5.5.2. Mental health

We used three measures to assess mental health, including depletion and isolation subscales from the revised loneliness rating scale [12], and the 10-item depression scale by Andresen *et al.* [14]. Depression was measured on 6-point scale, so we presented change in depressive symptoms in the same graph as life structure (figure 5), which was also measured on 6-point scale. Depletion and isolation were presented in the same graph (figure 6) as need satisfaction, as those constructs were measured on 7-point scale.

### 5.5.3. Depressive symptoms

Looking at changes in depressive symptoms, LGM suggested that Model 4 yielded the best fit ($\chi^2_8 = 29.85$, RMSEA = 0.09, SRMR = 0.04, CFI = 0.97) (table 9). Therefore, we chose Model 4 as our final model for depressive symptoms. This indicates that depressive symptoms decreased by 0.07 out of 6-point scale at every later assessment, suggesting that change was minimal over time (figure 5). Furthermore, the variance did not vary much across time points (table 9), but the highest observed scores decreasing from 6 at Time 1, to 5.50 at Time 2, to 4.60 at Time 3 and 4.90 at Time 4. This also means that the standard deviation slightly decreased from Time 1 to Time 4 (s.d.$_{T1}$ = 0.96, s.d.$_{T2}$ = 0.89, s.d.$_{T3}$ = 0.87, s.d.$_{T4}$ = 0.88).

### 5.5.4. Depletion

Looking at changes in depletion measures, LGM suggested that Model 5 yielded the best fit ($\chi^2_5 = 27.07$, RMSEA = 0.12, SRMR = 0.06, CFI = 0.96). While $\chi^2$-test suggested that Model 6 improved fit significantly compared with Model 5, Model 6 returned negative variance, suggesting that Model 6 was mis-specified. Therefore, we chose Model 5 as our final model for depletion, indicating that change over time was linear rather than quadratic. Nonetheless, as seen in figure 6, decrease in depletion over time was minimal, by 0.15 on average out of 7-point scale at every later time point. It is worth noting that the variance reduced

**Table 9.** Models investigating change in isolation within the first six weeks after lockdown lift.

| | 1 | 2 | 3 | 4 | 5 | 6 | 7a | 7b |
|---|---|---|---|---|---|---|---|---|
| intercept mean | 1.96*** | 1.95*** | 1.95*** | 2.02*** | **2.02*** | 2.02*** | 1.99*** | 1.14*** |
| intercept variance | | 0.75*** | 0.76*** | 0.80*** | **0.65*** | 0.80*** | 0.35*** | 0.36*** |
| residual variance (fixed) | 1.17*** | 0.42*** | 0.43*** | 0.42*** | | | | |
| residual variance (free) | | | | | **0.72, 0.49, 0.28, 0.26** | 0.51, 0.65, 0.22, 0.78 | 0.76, 0.45, 0.30, 0.26 | 0.76, 0.47, 0.28, 0.28 |
| linear slope mean | | | | −0.04* | **−0.04** | −0.05 | −0.06 | 0.07 |
| quadratic slope mean | | | | | | 0.00 | | |
| linear slope variance | | | −0.003 | −0.00 | **−0.02** | −0.03 | −0.02 | −0.03* |
| quadratic slope variance | | | | | | −0.04 | | |
| intercept—linear slope covariance | | | | −0.02 | **0.04** | −0.10 | 0.06* | 0.07** |
| intercept—quadratic slope covariance | | | | | | 0.02 | | |
| linear—quadratic slope covariance | | | | | | 0.06 | | |
| $\chi^2$(d.f.) | 636.88*** | 76.13*** | 75.87*** | 68.79*** | **13.75** | 1.02 | 37.44*** | 29.43* |
| RMSEA | 0.41 | 0.14 | 0.15 | 0.16 | **0.08** | 0 | 0.06 | 0.05 |
| SRMR | 0.44 | 0.09 | 0.09 | 0.08 | **0.05** | 0 | 0.04 | 0.04 |
| CFI | 0 | 0.89 | 0.89 | 0.90 | **0.99** | 1 | 0.97 | 0.98 |
| d.f. | 12 | 11 | 10 | 8 | **5** | 1 | 17 | 17 |
| change in $\chi^2$ | 560.75 | | 0.26 | 7.08 | **55.04** | 12.73 | 23.69 | 28.41 |

*$p < 0.05$, **$p < 0.01$, ***$p < 0.001$. Negative variances in all slope models suggested that the models are mis-specified, suggesting that estimates of slope parameters can be biased. Achieved power cannot be calculated for linear slope model due to negative slope variance.

over time (table 10), with the highest observed scores decreasing from 6.4 at Time 1 and 7 at Time 2, to 5.90 at Time 3 and 5.50 at Time 4. This also means that the standard deviation decreased from Time 1 to Time 4 (s.d.$_{T1}$ = 1.33, s.d.$_{T2}$ = 1.31, s.d.$_{T3}$ = 1.16, s.d.$_{T4}$ = 1.17).

### 5.5.5. Isolation

Looking at changes in isolation measures, LGM suggested that Model 5 yielded the best fit ($\chi^2_5 = 13.75$, RMSEA = 0.08, SRMR = 0.05, CFI = 0.99). Note that isolation models returned negative variance of the linear slope, suggesting that the models were mis-specified and thus could yield biased parameter estimations (tables 11 and 12). Therefore, we restrained from interpreting the estimates yielded by these models. As seen in figure 6, the change in isolation over time was very minimal after lockdown was lifted, mainly due to floor effects observed at all time points. In other words, a majority of participants that reported not experiencing isolation during lockdown as well as after lockdown.

Overall, our data yielded support for Hypothesis 3.1 instead of Hypothesis 3.2 for need satisfaction, depletion and depressive symptoms. After lockdown was lifted, participants' basic psychological needs increased while depressive symptoms and depletion decreased. Nonetheless, changes over time on all mental health indicators were minimal. Participants showed the largest change over time for depletion measure, which pertain to feeling low in energy and effectiveness. Furthermore, as can be seen in figure 6, change in means over time for depletion was driven by two factors: (i) the highest observed scores, and (ii) the medians on both scales dropped after Time 2. Observation of the scatter plot indicates that fewer participants chose 'strongly agree', and more participants chose 'strongly disagree' to experiencing depletion and isolation at Time 3 and Time 4.

## 5.6. Which coping strategies contributed to quality of life experienced at the end of lockdown and improvement after lockdown was lifted?

Generally, at Time 2 when asked about strategies that they used to cope during lockdown ($n = 292$), most participants reported using problem solving and help-seeking more often to cope with life during lockdown (figure 7). On average, strategies related to problem solving was chosen 81% of the time, followed by strategies related to help-seeking chosen 57% of the time. On the other hand, less optimal strategies like rationalization (e.g. justify the bad situation; 42%), fantasizing (e.g. wishing the problem would go away; 38%), avoidance (e.g. not wanting to face the problem; 40%) or self-blame (e.g. blaming oneself for the bad situation; 23%) were endorsed less often.

Six strategies were simultaneously entered into linear slope models (Model 4 or 5), based on the best-fitted models, to predict both the intercepts and the slopes of basic need satisfaction, depression, depletion and isolation. As such, the coefficient of each coping strategy on the intercept of the dependent variable reflects the link between the coping strategy with the baseline level of that dependent variable (Time 1; measured before lockdown lift), and coefficient of each coping strategy on the slope reflects the link between the coping strategy with the trajectory of change of that dependent variable over time. While we measured coping strategies at Time 2, Time 3 and Time 4, following the analytic plan in Stage 1 Registered Report, we only used the coping strategies measured at Time 2 (i.e. strategies used during lockdown) as predictors in these models.

For basic psychological need satisfaction, more reliance on problem solving during lockdown related to greater need satisfaction experienced at the end of lockdown (baseline), right before the lift ($B = 1.49$, $SE.B = 0.21$, $z = 7.07$, $p < 0.001$). On the other hand, blaming oneself for bad events related to lower need satisfaction experienced at the end of lockdown ($B = -0.66$, $SE.B = 0.22$, $z = -2.98$, $p = 0.003$). There was no significant effect of other coping strategies on the baseline level of need satisfaction and no significant effect of coping strategies on the slope of change over time for need satisfaction. The need satisfaction model that included coping strategies as time-invariant covariates of intercept and slope yielded satisfactory fit ($\chi^2_{20} = 54.72$, $p < 0.001$; RMSEA = 0.07, SRMR = 0.03, CFI = 0.96) (table 8).

For depressive symptoms, more reliance on problem solving during lockdown predicted lower depressive symptoms reported at the end of lockdown ($B = -1.19$, $SE.B = 0.23$, $z = -5.12$, $p < 0.001$). On the other hand, self-blame ($B = 0.69$, $SE.B = 0.24$, $z = 2.85$, $p = 0.004$), fantasizing, such as wishing the problem would go away ($B = 0.89$, $SE.B = 0.29$, $z = 3.10$, $p = 0.002$), or using avoidant strategies, such as drinking or smoking to escape ($B = 0.71$, $SE.B = 0.29$, $z = 2.42$, $p = 0.015$), predicted higher depressive symptoms reported at the end of lockdown. The depression model that included coping strategies

**Table 10.** Models investigating change in depressive symptoms within the first six weeks after lockdown lift.

| | 1 | 2 | 3 | 4 | 5 | 6 | 7a | 7b |
|---|---|---|---|---|---|---|---|---|
| intercept mean | 2.89*** | 2.69*** | 2.71*** | **2.78*** | 2.78*** | 2.81*** | 2.93*** | 1.78*** |
| intercept variance | | 0.58*** | 0.58*** | **0.66*** | 0.63*** | 0.61*** | 0.37*** | 0.41*** |
| residual variance (fixed) | 0.82*** | 0.25*** | 0.23*** | **0.21** | | | 0.21*** | 0.21*** |
| residual variance (free) | | | | | 0.27, 0.21, 0.21, 0.17 | 0.30, 0.23, 0.21, 0.22 | | |
| linear slope mean | | | | **−0.07*** | −0.07*** | −0.15** | −0.05 | 0.02 |
| quadratic slope mean | | | | | | 0.03 | | |
| linear slope variance | | | 0.01* | **0.02** | 0.01* | −0.07 | 0.02 | 0.01 |
| quadratic slope variance | | | | | | −0.01 | | |
| intercept—linear slope covariance | | | | **−0.04* | −0.02 | −0.00 | −0.02 | −0.02 |
| intercept—quadratic slope covariance | | | | | | −0.01 | | |
| linear—quadratic slope covariance | | | | | | 0.03 | | |
| $\chi^2$(d.f.) | 757.46*** | 65.09*** | 58.92*** | **29.85*** | 23.02*** | 17.72*** | 47.60*** | 51.69*** |
| RMSEA | 0.45 | 0.13 | 0.13 | **0.09** | 0.11 | 0.23 | 0.07 | 0.07 |
| SRMR | 0.47 | 0.07 | 0.09 | **0.04** | 0.04 | 0.04 | 0.03 | 0.03 |
| CFI | 0 | 0.93 | 0.93 | **0.97** | 0.98 | 0.98 | 0.97 | 0.96 |
| d.f. | 12 | 11 | 10 | **8** | 5 | 1 | 20 | 20 |
| change in $\chi^2$ | | 692.37 | 6.17 | **29.07** | 6.83 | 5.30 | 17.75 | 21.84 |
| achieved power | | | | **1.00** | | | | |

*$p < 0.05$, **$p < 0.01$, ***$p < 0.001$. Final model is bolded.

**Table 11.** Models investigating change in need satisfaction within the first six weeks after lockdown lift.

| | 1 | 2 | 3 | 4 | 5 | 6 | 7a | 7b |
|---|---|---|---|---|---|---|---|---|
| intercept mean | 4.77*** | 4.77*** | 4.76*** | **4.71*** | 4.70*** | 4.73*** | 3.84*** | 5.54*** |
| intercept variance | | 0.53*** | 0.51*** | **0.50*** | 0.49*** | 0.48*** | 0.29*** | 0.34*** |
| residual variance (fixed) | 0.75*** | 0.22*** | 0.20*** | **0.21*** | | | 0.21*** | 0.21*** |
| residual variance (free) | | | | | 0.25, 0.18, 0.22, 0.18 | 0.22, 0.24, 0.20, 0.43 | | |
| linear slope mean | | | | **0.04** | 0.05*** | −0.03 | 0.13 | −0.01 |
| quadratic slope mean | | | | | | 0.03* | | |
| linear slope variance | | | 0.01* | **0.01 | 0.00 | −0.09 | 0.01 | 0.01 |
| quadratic slope variance | | | | | | −0.02** | | |
| intercept—linear slope covariance | | | | **0.01 | 0.01 | 0.04 | 0.01 | 0.02 |
| intercept—quadratic slope covariance | | | | | | −0.01 | | |
| linear—quadratic slope covariance | | | | | | 0.05 | | |
| $\chi^2$(d.f.) | 751.93*** | 37.76*** | 32.19*** | **20.32*** | 15.64** | 0.10 | 54.72*** | 39.92** |
| RMSEA | 0.44 | 0.09 | 0.08 | **0.07 | 0.08 | 0.00 | 0.07 | 0.06 |
| SRMR | 0.47 | 0.06 | 0.05 | **0.03 | 0.04 | 0.00 | 0.03 | 0.03 |
| CFI | 0.003 | 0.96 | 0.97 | **0.98 | 0.99 | 1.00 | 0.96 | 0.98 |
| d.f. | 12 | 11 | 10 | **8 | 5 | 1 | 20 | 20 |
| change in $\chi^2$ | 714.17 | 714.17 | 5.57 | **11.87 | 4.68 | 15.54 | 34.40 | 19.60 |
| achieved power | | | | **0.87 | | | | |

$*p < 0.05$, $**p < 0.01$, $***p < 0.001$. Negative variances in Model 6 suggested that the model is mis-specified. As such, even though $\chi^2$-test yielded significantly better fit, we chose Model 4 as the best model. Final model is bolded.

**Table 12.** Regression coefficients reflecting the association between coping strategies and stressors experienced during lockdown with need satisfaction, depressive symptoms, depletion, isolation.

| | need satisfaction | | depressive symptoms | | depletion | | isolation | |
|---|---|---|---|---|---|---|---|---|
| | intercept | linear slope | intercept | linear slope | intercept | linear slope | intercept | linear slope |
| *coping strategies during lockdown* | | | | | | | | |
| problem solving | 1.49*** | −0.11 | −1.19*** | 0.06 | −1.05** | 0.11 | −0.77** | −0.01 |
| self-blame | −0.66** | −0.01 | 0.69** | −0.04 | 0.95** | −0.10 | 1.08*** | −0.07 |
| help-seeking | 0.27 | −0.00 | −0.15 | −0.04 | −0.54* | 0.01 | −0.54* | 0.05 |
| fantasizing | −0.34 | 0.08 | 0.89** | −0.02 | 1.64*** | −0.28* | 0.56 | 0.03 |
| avoidance | −0.40 | −0.12 | 0.71* | −0.09 | 0.78 | −0.08 | 0.45 | −0.19 |
| rationalization | −0.09 | 0.06 | 0.25 | 0.05 | 0.26 | 0.28 | 0.72 | 0.18 |
| *stressors during lockdown* | | | | | | | | |
| family relations | −0.04 | −0.03 | 0.07 | 0.01 | 0.10 | 0.01 | 0.25** | −0.05* |
| romantic relations | −0.00 | 0.01 | 0.08 | −0.01 | 0.17 | −0.04 | 0.08 | −0.03 |
| interpersonal relations | −0.11 | 0.03 | 0.14 | 0.02 | 0.26 | 0.09 | 0.19 | 0.16*** |
| health concerns | 0.10 | 0.03 | −0.08 | −0.02 | −0.19 | 0.02 | 0.04 | −0.03 |
| financial stress | −0.11* | −0.01 | 0.09 | 0.00 | 0.10 | −0.02 | 0.11* | −0.02 |
| daily disruptions | −0.33*** | 0.03 | 0.40*** | −0.04** | 0.51*** | −0.08*** | 0.26*** | −0.04 |

*p < 0.05, **p < 0.01, ***p < 0.001.

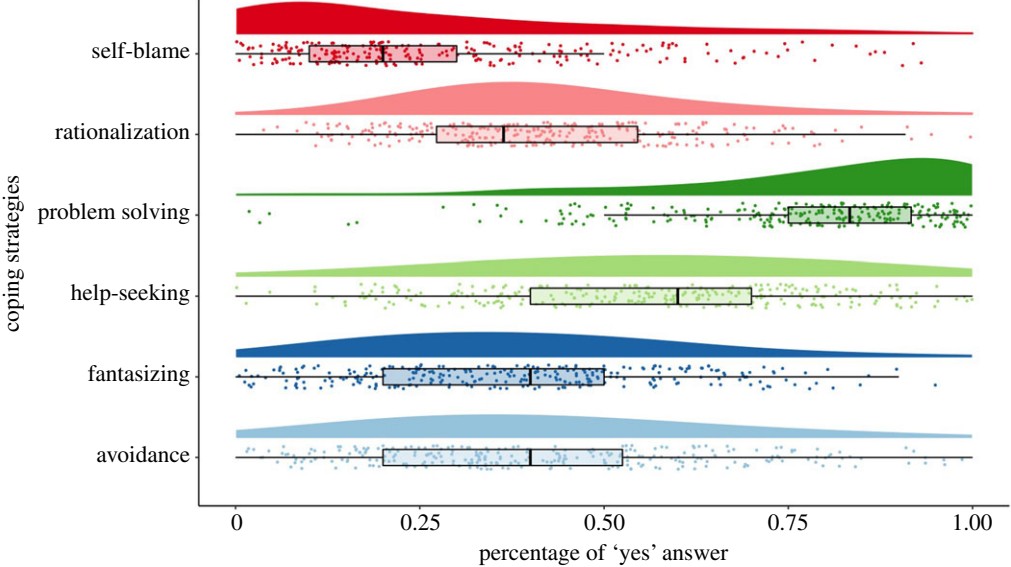

**Figure 7.** Prevalence rate of coping strategies endorsed during lockdown.

as time-invariant covariates of intercept and slope yielded satisfactory fit ($\chi^2_{20} = 47.60$, $p < 0.001$; RMSEA = 0.07, SRMR = 0.03, CFI = 0.97) (table 9).

For depletion, more reliance on problem solving ($B = -1.05$, $SE.B = 0.33$, $z = -3.19$, $p = 0.001$) and help-seeking ($B = -0.54$, $SE.B = 0.27$, $z = -2.00$, $p = 0.045$) predicted lower depletion reported at the end of lockdown. On the other hand, self-blame ($B = 0.95$, $SE.B = 0.34$, $z = 2.78$, $p = 0.005$), and fantasizing ($B = 1.64$, $SE.B = 0.40$, $z = 4.09$, $p < 0.001$) predicted greater depletion reported at the end of lockdown. The depletion model that included coping strategies as time-invariant covariates of intercept and slope yielded satisfactory fit ($\chi^2_{17} = 49.50$, $p < 0.001$; RMSEA = 0.08, SRMR = 0.04, CFI = 0.96) (table 10).

For isolation, the model that included coping strategies as time-invariant covariates of intercept and slope yielded moderate fit ($\chi^2_{20} = 94.96$, $p < 0.001$; RMSEA = 0.11, SRMR = 0.05, CFI = 0.90). However, the model returned negative variance for the linear slope, similarly to previous isolation models reported above in table 9, suggesting that the parameter estimates for linear slope could be biased. In relation to the intercept, problem solving ($B = -0.69$, $SE.B = 0.29$, $z = -2.41$, $p = 0.016$) and help-seeking ($B = -0.53$, $SE.B = 0.24$, $z = -2.27$, $p = 0.023$) predicted lower levels of isolation at the end of lockdown, while self-blame ($B = 0.94$, $SE.B = 0.30$, $z = 3.14$, $p = 0.002$) predicted more isolation (table 11).

Coping strategies used during lockdown did not predict different trajectories of change over time. This indicates that while coping strategies used during lockdown might be linked to participants' experiences reported at the end of lockdown, they were not significant factors that predicted the extent to which quality of life improved over time after lockdown was lifted. Overall, we did not find the support that rationalization, through trying to justify one's life events instead of facing the actual problem, predicted participants' experiences of need satisfaction, depressive symptoms, depletion and isolation at the end of lockdown. Nonetheless, as predicted, problem solving and help-seeking during lockdown related with more positive outcomes experienced at the end of lockdown, while self-blame, fantasizing and avoidance predicted more negative outcomes.

## 5.7. Which stressors experienced during lockdown contributed to quality of life experienced at the end of lockdown and improvement after lockdown was lifted?

Generally, participants have not experienced many severe stressors during lockdown (figure 8). The most severe stressor reported during lockdown was daily disruptions such as having to change daily routines, not having space to exercise, not being able to commute to places or communicate with others, but even so, the mean was 2.08 (s.d. = 0.98) on scale ranging from 0 (never happened) to 5 (extremely severe). The second most severe stressor during lockdown was problems with work and finance such as having frustration with work, family's financial difficulty, or career disruption ($M = 1.02$, s.d. = 1.10), followed by family problems, such as difficulties with child rearing and troubles with family members ($M = 0.60$, s.d. = 0.86). The percentages of people who reported experiencing the six types of stressors (choosing values

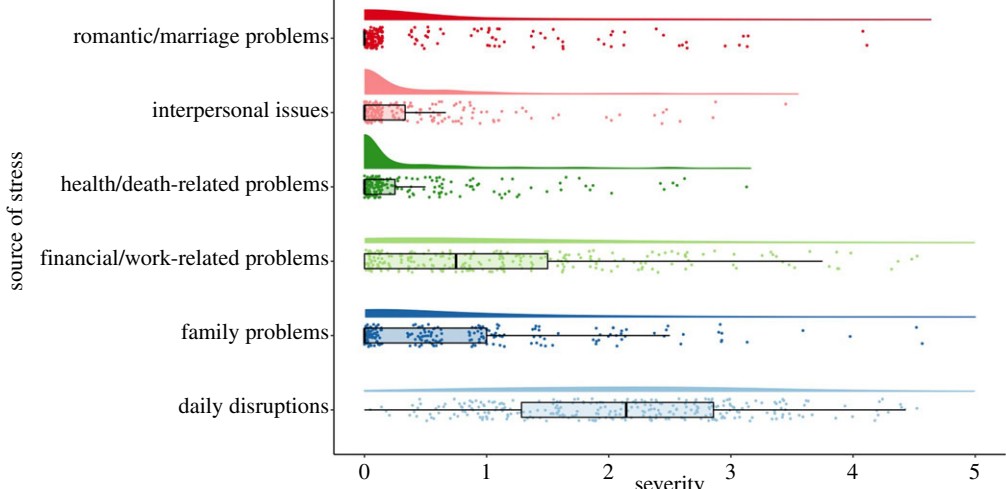

**Figure 8.** Severity levels (1–7) of different stressors experienced during lockdown.

other than 0) are as follows: 99% experienced daily disruptions, 71% experienced financial or work-related problems, 49% experienced family problems, 31% experienced interpersonal issues, 29% experienced health or death-related problems and 22% experienced romantic or marriage problems.

Six types of stressors were simultaneously entered into linear slope models (Model 4 or 5), based on the best-fitted models, to predict both the intercepts and the slopes of basic need satisfaction, depression, depletion and isolation. As such, the coefficient of each type of stressors on the intercept on the dependent variable reflects the link between the stressor with the baseline level of that dependent variable (Time 1; measured before lockdown lift), and coefficient of each type of stressor on the slope reflects the link between the stressor with the trajectory of change of that dependent variable over time. While we measured stressors at Time 2, Time 3 and Time 4, following the analytic plan in Stage 1 Registered Report, we only used the stressors measured at Time 2 (i.e. experienced during lockdown) as predictors in these models.

The models that included types of stressors as time-invariant covariates of intercepts and slopes of the dependent variables yielded satisfactory fit for need satisfaction (table 8; $\chi^2_{20} = 39.92$, $p = 0.005$; RMSEA = 0.06, SRMR = 0.03, CFI = 0.98) and depressive symptoms (table 9; $\chi^2_{20} = 51.69$, $p < 0.001$; RMSEA = 0.07, SRMR = 0.03, CFI = 0.96), and moderate fit for depletion (table 10; $\chi^2_{17} = 55.98$, $p < 0.001$; RMSEA = 0.09, SRMR = 0.04, CFI = 0.95). Fit indexes for the isolation model were also satisfactory (table 11; $\chi^2_{17} = 49.50$, $p < 0.001$; RMSEA = 0.08, SRMR = 0.04, CFI = 0.96) but this model yielded negative slope variance, so we cautioned against interpreting coefficients on the linear slope for this model.

Across all models, we found little support for our hypotheses that financial stress and health or death-related problems were the two main predictors of participants' experiences at the end of lockdown and improvement over time after lockdown was lifted. Health issues of oneself or the death of a family member did not yield any significant coefficient on need satisfaction, depression, depletion or isolation experienced at the end of lockdown nor did it predict change over time after lockdown was lifted (see tables 8–11). Financial stress predicted the intercept of need satisfaction and isolation; experiencing more financial problems during lockdown related to lower need satisfaction ($B = -0.11$, $SE.B = 0.04$, $z = -2.59$, $p = 0.010$), and higher isolation ($B = 0.11$, $SE.B = 0.05$, $z = 2.16$, $p = 0.031$) reported at the end of lockdown.

Daily disruptions—a predictor that we did not include in our Hypotheses 5.1 and 5.2—emerged as the most consistent predictor of quality of life at the end of lockdown. Daily disruptions predicted lower need satisfaction ($B = -0.33$, $SE.B = 0.05$, $z = -7.31$, $p < 0.001$), more depressive symptoms ($B = 0.40$, $SE.B = 0.05$, $z = 8.08$, $p < 0.001$) and lower levels of depletion ($B = 0.51$, $SE.B = 0.07$, $z = 7.35$, $p < 0.001$). Further, experiencing more daily disruptions during lockdown predicted steeper drops of depressive symptoms ($B = -0.04$, $SE.B = 0.02$, $z = -2.64$, $p = 0.008$) and depletion ($B = -0.08$, $SE.B = 0.02$, $z = -3.76$, $p < 0.001$) over time after lockdown was lifted. This suggests that those who experienced more severe daily disruptions during lockdown, such as disruptions in eating habits, daily routines or commuting, etc. experienced greater mental health improvement after lockdown was lifted. Nonetheless, it is worth noting that those coefficients on the slope of change were very small, indicating that daily disruptions explained very minimally the different trajectories of change in depressive symptoms and depletion (see figures 9 and 10).

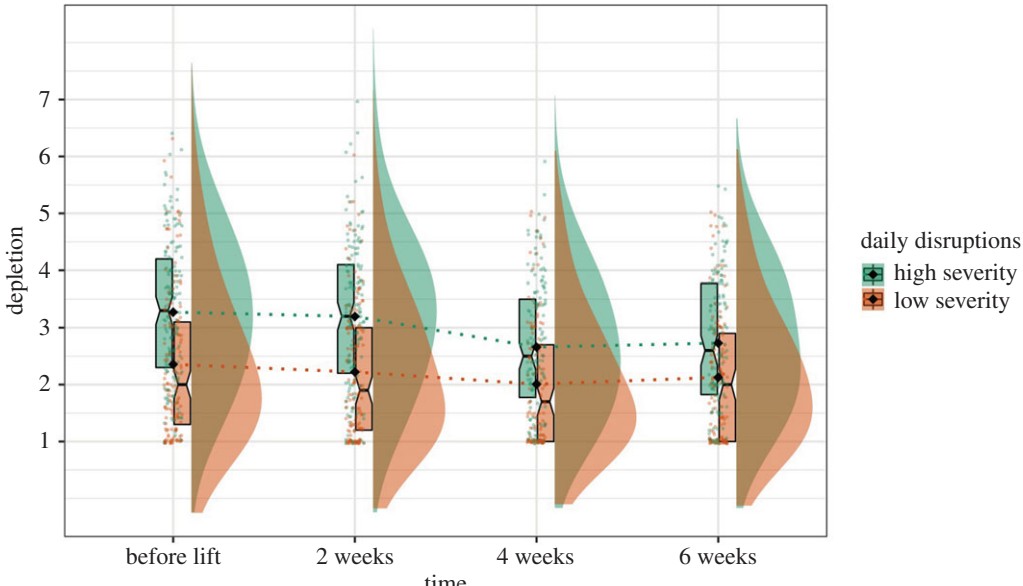

**Figure 9.** Graph illustrating different trajectories of change in depletion within the first two months after lockdown by different levels of daily disruptions reported during lockdown.

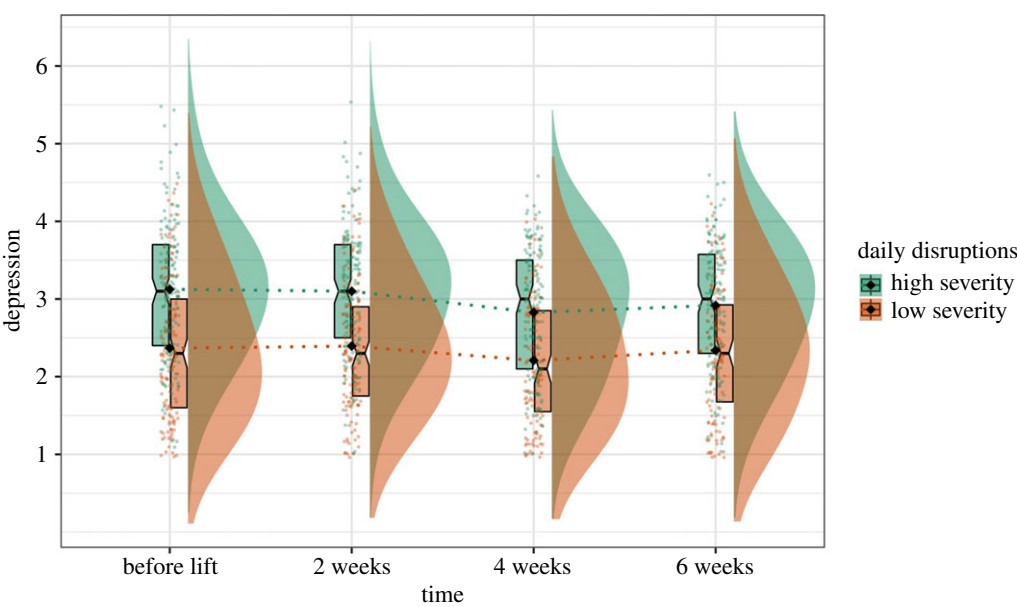

**Figure 10.** Graph illustrating different trajectories of change in depressive symptoms within the first two months after lockdown by different levels of daily disruptions reported during lockdown.

Another stressor that emerged as significant predictor for isolation was stress related to family problems, such as problems with child rearing and troubles with family members ($B = 0.24$, $SE.B = 0.08$, $z = 2.96$, $p = 0.003$). In the sample recruited at the end of lockdown, only 35 participants reported living alone, whereas 52 participants lived with one other person, 130 participants lived with two other people, 85 lived with three and 63 lived with more than three people. As such, this sample represents those in Wuhan who were self-isolating with other people. To the extent that living with others created troubles between people in the same household, this factor contributed to heightened feeling of isolation at the end of lockdown.

Overall, the results of LGM on the motivation of self-isolation, life structure and mental health status in our sample supported the linear tendency after lockdown was lifted in Wuhan. We determined different models (Models 4, 5 and 6) to describe the data due to the following two reasons. First, we followed the data analysis plans that we have registered at Stage 1. Second, although Model 4 supported the linear

trajectories of the change over time, Model 5 or 6 offered more information (e.g. the change of residual variance) and helped us better interpret the data. Nonetheless, taken together, all models give a convergent picture of gradual linear, optimistic change over time after lockdown is lifted.

# 6. Discussion

The present study used LGM to investigate the change in life after lockdown among a sample of residents in Wuhan, China. To our knowledge, this is the first study to illustrate the trajectory of motivation for self-isolation, well-being and mental health outcomes, after the lockdown period in Wuhan. The findings in the current study contribute to the current understanding of whether and how life returns to normality and what factors might undermine this change over time.

For motivation to self-isolate, we see relatively similar downtrend for both identified and external motivation. Given that the decrease of identified motivation for self-isolation was mainly driven by a few individuals choosing the lower-scored responses, the data suggest that most of our participants still display high motivation of self-isolation to protect oneself and others' health after lockdown was lifted, even though the fear of being punished for going outside has gone down.

The findings related to motivation for self-isolation might also be due to several societal factors. First, our participants might still be cautious to prevent a second wave of infections, mainly due to the absence of vaccine and the uncertain infections from asymptomatic COVID-19 carriers. According to Bai *et al.* [33], asymptomatic COVID-19 carriers do not develop notable COVID-19 symptoms at any point in time, but may contribute to the spread of the disease, and the challenge of tracking them caused more concerns on the current worldwide pandemic containment [34]. Second, information about the mechanism of COVID-19 virus transmission continues to be communicated to the public through the official media by Chinese government. Third, as one of the most populous cities in China, Wuhan is a major transportation hub, with dozens of railways, roads and expressways passing through the city and connecting to other major cities. After the lift of lockdown, travel between different areas in China is increasing, and the appearance of imported cases from overseas may also raise the concern of possible COVID-19 infections. As such, these variables might contribute to maintaining most of our participants' identified motivation (e.g. I recognize the protective effect of self-isolation), even though external motivation driven by governments' restrictions and fear of punishment has generally reduced. Note that, however, motivation for self-isolating reflects people's perceptions of why they are self-isolating. We did not have data to speak to whether this directly translates to actual behaviour in day-to-day life, as not everyone who sees the value in self-isolating can shield at home; this largely depends on individual life circumstances.

On the other hand, we found the structure of daily life increased slightly over time. Even though we do not have the baseline level of life structure before the outbreak of COVID-19 among our participants, the evidence that daily disruptions were the most severe type of stressors during lockdown, which we discuss in more details later in this Discussion, may explain the possible reason for the slight increase of life structure after lockdown was lifted. What we do not know with this data are whether this gradual increase began after lockdown was lifted or whether change has occurred during lockdown. In other words, this result might be evidence of psychological resilience under the tremendous transition of self-isolation [35], and our participants may have found suitable ways to adjust their lives over time and thus have experienced gentler transition from lockdown to normality.

On a similar note, our participants did not report severe mental health problems (e.g. isolation, depletion, depressive symptoms) during the lockdown, and our findings suggest slight improvement of psychological need satisfaction after lockdown was lifted. We see relatively low levels of mental health risks in this sample, which could be due to the following two reasons. First, although one-third of our participants know at least one COVID-19-infected person, the majority of our participants were in good health themselves and recognized the value of self-isolation. Therefore, our participants' situations are very different from those who are in obligatory quarantine for being diagnosed with this disease [36]. Thus, our participants may have low chance to experience severe mental sufferings due to fear of contracting the virus.

Our study is not the first to show that mental health risks have not drastically increased during lockdown. Similar results have been found in samples recruited in the United Kingdom [3] and the United States [35]. These findings suggest that, despite concerns around rise in mental health issues during the COVID-19 pandemic self-isolation, the general population might have displayed tremendous resilience throughout this time [35].

From our perspective, evidence of resilience is only inferred rather than directly measured; that is, in time of a pandemic when we should expect mental health to suffer, we did not find evidence for such problem and we take that as a sign of resilience. Many societal factors can explain such resilience in the Chinese context. One paper speculated that since the lockdown just happened 1 day before the Chinese New Year's Eve, the holiday might serve as a buffer that helped lessen stress around the virus [37]. For example, by the time the lockdown was imposed, most people in China had returned back home to reunite with their families and had the opportunity to reserve sufficient amount of food and daily necessities due to the traditional custom to spend the 7-day national holiday.

The more important point is that this sample did not show drastic decrease in psychological well-being nor increase in mental health risks. We have predicted this trend as a competing hypothesis (Hypothesis 3.2). We did indeed observe a few individuals that displayed high mental health risks (e.g. isolation, depletion and depressive symptoms) in this sample, but the number of those high values dropped over time, as indicated by the distribution tail approaching lower values. At the same time, it is worth noting that the majority of our participants are adults in good health, which separated them from other more vulnerable populations, such as children, COVID-19 patients and medical care workers. Some emerged evidence showed that these populations would face more mental health risks due to the COVID-19 pandemic [38–40]. As such, supporting previous take-aways by Weinstein & Nguyen [3], and by Luchetti *et al.* [35], we see evidence of no drastic increase in mental health risks in general population as an encouraging sign.

Further, it is also important to contextualize findings on mental health risks during COVID-19 based on the time of data collection. For example, our data were collected at the beginning of April at the end of the lockdown period, whereas another study by Gao *et al.* [41] collected data from the Chinese general population before lockdown was imposed in Wuhan. Gao *et al.*'s [41] study showed high prevalence of depressive symptoms (48.3%) and anxiety (22.6%). Lacking longitudinal data that track those participants over time, it is difficult to know whether depressive symptoms and anxiety in the sample of Gao *et al.*'s [41] study remained the same, went up or dropped after that. In other data collected from population-representative sample in the US (Understanding America Study; https://covid19pulse.usc.edu/), it was shown that depression and anxiety were heightened during the period leading up to the country's border closure but gradually dropped after that. So the difference between our findings and Gao *et al.* [41] results could be due to the timeframe when data were collected.

Besides, another issue on the time of data collection is that we only collected data in the first two months after lockdown was lifted in Wuhan, China. In other words, our data could only draw the picture of the short-term effect of COVID-19 impact on mental health risks. As the global COVID-19 outbreak sparked more concerns on the economic crisis and recession, we could not speak to the long-term effect of the pandemic on states of mental health in the general population. To investigate long-term effects of lockdown, we will need more assessments, but unfortunately our limited funding has not allowed us to continue this endeavour. Therefore, we can only speak to the immediate short-term effect of lockdown within the first two months after the lift of restrictions.

When investigating predictors of well-being and mental health risks, as predicted, problem solving and help-seeking during lockdown were found as the protective coping strategies for well-being and mental health, while self-blame, fantasizing and avoidance used during lockdown were found to increase the risks at the end of lockdown. This finding is in line with previous studies, that finding ways to resolve issues at hand and seeking help from others have been shown to be adaptive coping strategies for Chinese people, and linked to lower psychological maladjustment. By contrast, self-blame, fantasizing and avoidance were related to maladaptive coping strategies, and linked with more negative outcomes [19,20]. However, we did not find the support for the prediction of any coping strategies on trajectory of change over time for psychological well-being and mental health risks. As discussed before, the changes of psychological well-being and mental health risks were minimal in the current study.

For the stressors experienced during lockdown, contrary to our research hypotheses, daily disruptions, instead of financial stress and health or death-related problems, was found as the main predictor of participants' well-being and mental health risks at the end of lockdown. In the current study, daily disruptions refer to the disruptions of original life's living habits and routines. Indeed, this COVID-19 outbreak caused dramatic shift in the routines and rituals of individual's life. For instance, people cannot visit relatives and friends during the Chinese New Year as usual, the school closure caused more caregiving burdens for parents, and working at home made the establishment of work–life boundary even harder. Previous studies suggested that routines and rituals in families could provide stability and continuity during period of stress and promote the strength and solidarity of the family [42,43]. Studies on children believed that routine is critical in the establishment of their

sense of stability and feelings of security [44,45], by offering the predictability in the environment [46]. Besides, a recently study argued that part of the protective functions of the family adaptive system can be achieved through daily routines and rituals [47], which in turn would promote family resilience when family members are facing risks or difficulties. Taken together, the function of routines and rituals might be important in the current COVID-19 pandemic. When consistent expectations of daily life through maintaining regular daily routines are disrupted, people could find home-confined time challenging and distressing.

We were surprised at the lack of evidence for health concerns being predictive of psychological ill-being and mental health risks at the end of lockdown. Other papers have discussed the impact of health anxiety, especially exacerbated by exposure to social media, on mental health [3,41,48]. While our sample does not include individuals who were infected by the virus, more than one-third of our participants reported knowing at least one COVID-19-infected person with a few knowing more than 10 people being infected. However, note that by April, the Chinese government has taken preventive measures to control the spread. Starting on 5 February, Hubei provincial authorities vowed to leave no virus-infected patient unattended (Xinhuanet), and established Fangcang shelter hospitals, as well as rapidly converted existing public venues to large-scale, temporary hospitals, to provide additional medical resources [49]. As such, for those who were not infected with the virus but self-isolated at homes, they may be less concerned about catching the disease or not having immediate access to hospital services, which could explain why health-related stressors did not predict ill-being and mental health risks.

It was also surprising to us that financial and work-related issues were not aggravators of mental health risks. Since we relied on convenience sampling approach by recruiting WeChat users, we were less likely to reach those who faced financial and work-related difficulties. In other words, we do not know whether the lack of negative impact of financial and work-related stressors on mental health could be due to the characteristics of the current sample. Chinese households that are financially secure are likely to have precautionary savings put aside to deal with uncertain circumstances [50–52]. Studies suggested that those precautionary savings could smooth people's consumption when facing high income uncertainties or when adverse economic shocks occur [51,53]. Further, we also did not find evidence suggesting that financial and work-related stressors aggravated mental health risks over time after lockdown was lifted and after people went back to work. What our data could not speak for is whether change over time after lockdown lift could have been more pronounced, had we been able to recruit a larger sample with more diverse financial and employment circumstances. For example, future studies could compare a group of people that faced unemployment as a result of COVID-19 with those who did not. Using longitudinal data collected from more than 24 000 adults in Germany, Lucas et al. [54] found that after facing unemployment, people's life satisfaction did bounce back but not to the baseline that they had prior to unemployment. Overall, our study yields evidence of optimistic, albeit slow, changes after lockdown. The most pronounced changes occur in people's motivation for self-isolating. Participants in this sample gradually perceived less pressure from the government to self-isolate and less fear of legal consequences for leaving their homes. Nonetheless, most participants continued seeing benefits in self-isolating to protect their own health and other people's health, even though there is a small, increasing, number of those who reported seeing little benefit. In relation to life structure, well-being and mental health risks, there appears to be a slow return to normality with small increase over time in life structure and need satisfaction, and small decrease in isolation, energy depletion, and depressive symptoms. Coping strategies and stressors experienced during lockdown do not seem to be important factors that predict the different trajectory of change in well-being and mental health risks. However, coping through problem solving and seeking help from others are predictive of better well-being and mental health, while blaming oneself, avoiding the problem or rationalizing around the ongoing pandemic are linked to lower well-being and mental health at the end of lockdown. During the time of lockdown, disruptions that prevent people to carry out their daily routines emerge as the strongest predictor of lower well-being and more mental health risks reported at the end of lockdown.

## 7. Conclusion

This study presented descriptive data on a sample of residents living in Wuhan during the lockdown in response to the novel coronavirus outbreak. We included data on Wuhan participants' motivation for self-isolation, life stressors, coping strategies and different indicators of well-being and mental health. These quantitative measures could be clarified with further qualitative data to understand why the

results here showed little change after lockdown was lifted. For example, other researchers [35] have suggested that psychological resilience might be a possible explanation of the presented findings. However, this interpretation needs to be solidified and complemented with interviews or focus groups to investigate what resilience means behaviourally or emotionally during the lockdown period in Wuhan. Indeed, our sample reported engaging mostly in problem solving and help-seeking, and these two types of strategies are linked to positive well-being and mental health outcomes during lockdown. What we do not know is how people used these strategies in their daily life to get through the disruptions caused by the outbreak of COVID-19. As such, the follow-up studies could be complemented with qualitative research to contextualize the findings we presented in this paper.

Taken together, this set of findings do not yield evidence that there are dramatic changes in motivation for isolating or well-being and mental health indicators after lockdown was lifted in Wuhan, China, which was considered one of the high-risk areas in the current ongoing pandemic. Nonetheless, the shifts in motivation in a few participants warrant further research to zoom into characteristics of individuals who saw little benefits in self-isolating after lockdown is lifted. With respect to well-being and mental health indicators, the results favour the direction of change which reflects that this sample of individuals from Wuhan, China, have coped adaptively and remained psychologically stable, if not improved, after lockdown was lifted. Consistent with other longitudinal data in the UK and US showing the stability of mental health and well-being over time during lockdown [3,35], our results suggest that for our participants life after lockdown may have settled into the 'new normal'.

Ethics. We have received ethical approval from University of Durham's Ethics Committee (PSYCH-2020-03-11T23_41_08) and East China Normal University's Ethics Committee (HR 179-2020).

Data accessibility. All measures, data and code are available on Open Science Framework at https://osf.io/5rxz6/?view_only=b3a7c3653bd043d189eea2513e66bfde.

Authors' contributions. T.Z. and T.T.N. worked collaboratively to draft the Stage 1 manuscript. T.T.N. and T.Z. conceptualized the study, developed methods, ethical documentation, prepared registrations and study materials. T.Z. gathered questionnaires available in Chinese as well as translated all surveys that were not available in Chinese. J.Z. helped recruit participants, and T.Z. collected data and followed up with participants. T.Z. prepared the data and wrote up the syntax for variable computation in SPSS. T.T.N. analysed the data in R and wrote up the Results section. T.T.N. and T.Z. worked collaboratively on the Discussion section. J.L. funded the current project, supervised the data collection process and provided feedback to the Stage 2 manuscript.

Competing interests. We declare we have no competing interests.

Funding. This work was supported by the Fundamental Research Funds for the Central Universities (2017ECNU-HLYT013, 2018ECNU-QKT015) to J.L.

# Appendix A. Demographic information of the final sample that is included in confirmatory analyses

| categories | Time 1 (n = 313) | Time 2 (N = 292)[a] | Time 3 (N = 284)[b] | Time 4 (N = 279)[c] |
| --- | --- | --- | --- | --- |
| gender | | | | |
| men | 110 | 106 | 102 | 99 |
| women | 203 | 186 | 182 | 180 |
| marital status | | | | |
| married | 188 | 172 | 171 | 169 |
| living together as married | 9 | 9 | 9 | 8 |
| divorced | 9 | 9 | 9 | 8 |
| separated | 0 | 0 | 1 | 2 |
| widowed | 7 | 4 | 4 | 4 |
| single/never married | 78 | 76 | 73 | 71 |
| living apart but steady relation (married, cohabitation) | 22 | 22 | 17 | 17 |

(Continued.)

| categories | Time 1 (*n* = 313) | Time 2 (*N* = 292)[a] | Time 3 (*N* = 284)[b] | Time 4 (*N* = 279)[c] |
|---|---|---|---|---|
| general state of health | | | | |
| very poor | 9 | 8 | 4 | 2 |
| poor | 7 | 9 | 6 | 6 |
| fair | 20 | 14 | 20 | 27 |
| good | 137 | 126 | 128 | 116 |
| very good | 135 | 134 | 126 | 129 |
| I'm not sure | 5 | 1 | 0 | 0 |
| self-isolate in response to the coronavirus (COVID-19) outbreak | | | | |
| yes | 247 | 198 | 135 | 108 |
| no | 57 | 54 | 93 | 106 |
| in part | 8 | 40 | 56 | 65 |
| infected by the coronavirus (COVID-19) | | | | |
| yes | 1 | 0 | 0 | 0 |
| no | 312 | 292 | 284 | 279 |
| employment status | | | | |
| full time | 155 | 145 | 148 | 148 |
| part time | 3 | 3 | 8 | 6 |
| self-employed | 12 | 11 | 10 | 8 |
| retired | 32 | 30 | 30 | 29 |
| housewife/husband | 10 | 12 | 12 | 9 |
| students | 63 | 59 | 49 | 48 |
| unemployed | 11 | 8 | 11 | 11 |
| other | 27 | 24 | 16 | 20 |
| knowledge of at least one person infected by COVID-19 | | | | |
| yes | 105 | 97 | 108 | 104 |
| no | 208 | 195 | 176 | 175 |
| how many people? | 2.96 (2.68) | 3.03 (2.31) | 3.38 (3.33) | 3.44 (3.19) |

[a]Participants who filled out the Time 2 survey and can be matched with Time 1 were taken into consideration.
[b]Participants who filled out the Time 3 survey and can be matched with Time 1 were taken into consideration.
[c]Participants who filled out the Time 4 survey and can be matched with Time 1 were taken into consideration.

## Appendix B. Coding for responses to the question 'How has self-isolation influenced your life?' and the number of times each category mentioned

| categories | times of mention |
|---|---|
| negative influences | |
| lack of opportunities for interpersonal communication | 18 |
| inconvenience of grocery shopping | 102 |
| lack of freedom | 20 |
| negative body condition | 5 |

| categories | times of mention |
|---|---|
| transportation difficulty | 11 |
| unable to work | 13 |
| financial difficult | 37 |
| negative emotion | 107 |
| general routine change | 12 |
| sleeping time | 2 |
| eating habit | 4 |
| lack of exercise | 14 |
| positive influences | |
| positive emotion | 12 |
| solitude and independence | 4 |
| opportunity for self-regulation | 3 |
| opportunity with families | 1 |
| no influence | 100 |

## Appendix C. Stressor list administered at Time 2, Time 3, and Time 4

| categories | items from Zheng & Lin [6] |
|---|---|
| problem with romantic relationship or marriage | |
| interference by a romantic relationship or marriage | x |
| broken relationship with spouse or lover | x |
| family problems | |
| difficulties with child rearing | x |
| trouble with family members | x |
| problems with interpersonal relationship | |
| trouble with boss or leaders | x |
| trouble with colleagues or neighbours | x |
| quarrel with others over trivial matters | x |
| problems with health | |
| death of spouse | x |
| acute or severe sickness | x |
| acute or severe sickness of relatives or friends | x |
| death of relatives or friends | x |
| problems with work and finance | |
| family financial difficulty | x |
| frustration with work | x |
| deferred payment of salary or bonus | |
| career disruption | |
| daily disruptions | |
| changes in eating habits | |
| changes in daily routine | x |

(*Continued*.)

| categories | items from Zheng & Lin [6] |
| --- | --- |
| changes in activities area | |
| the living environment is under threat | |
| change in commuting | |
| lack of opportunities for interpersonal communication | |
| inconvenience of grocery shopping | |

Notes: (x) mark items adapted from Zheng & Lin's [8] nationwide study of stressful life events in Mainland China

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
