## [Reviewer comments · Royal Society Open Science]

Review History

RSOS-200705.R0 (Original submission)

Review form: Reviewer 1 (Daniel Dunleavy)

Do you have any ethical concerns with this paper?

No

Recommendation?

Major revision

Comments to the Author(s)

Comments to the authors:

I thank the authors for there submission. Overall, I think the submission targets a timely and important issue. The following questions, comments, suggestions, etc. are posed in hopes of improving or clarifying aspects of this submission.

1. It is unclear to me if the target population is restricted to adults only. If so, how do the authors ensure that respondents are adults? For example, is there a filtering question provided at the beginning of the survey?

2. How is the survey distributed? I understand participants are approached via the WeChat app. Is the survey hosted through another provider (e.g. SoSciSurvey or Qualtrics, or some other alternative)?
3. Similarly, where will the data be hosted? I see that the author are using the Open Science Framework to post their survey variables. I presume data would be hosted there?
4. What measures will be taken to ensure that the data remains anonymized/confidential; while still adhering to the requirement of making data/materials publicly available?
5. It is unclear to me how participants are tracked from Time 1 to Time 2 to Time 3. The authors appear to be using a form of convenience and/or snowball sampling through the WeChat app. I am unfamiliar with the app, so please excuse my ignorance. Will the participants need to have continued to use the app throughout the study for researchers to have access/connect to them? Or are there alternative ways of connecting once they have enrolled in the study? Further, is their a participant ID linked to the participant? Do they know this ID and re-enter it when completing Time 2 and Time 3? Or do the researchers connect the ID to the participant somehow (e.g. using WeChat profile information)?
6. The Registered Report proposal was submitted before Time 2 data collection was completed. Now that I am reviewing it, I believe data collection has stopped. Is there a way you can update us on what the sample size for Time 2 was? And further what % of participants remained in the study? This may be an important indicator for how many people continue through to Time 3 and has implications for the sensitivity analysis performed as part of the proposal. For example, it may affect how large the smallest effect size of interest is.
7. How will the authors handle missing data? Can participants skip questions or are they forced to answer each question before moving to the next?
8. Was there missing data for Time 1? What percentage? [This can also apply to Time 2 now that it has been collected].
9. Are participants being compensated in any way?

I hope these questions, comments, etc. have helped improve the proposal in some way.

Thank you for your consideration of them.

Take care,
Daniel Dunleavy

Review form: Reviewer 2

Do you have any ethical concerns with this paper?

No

Recommendation?

Major revision

Comments to the Author(s)

From my understanding, the authors have started data collection for an online study with citizens of Wuhan, where a strict lockdown was imposed and has been lifted. The aim of the

research was to understand impact of easing lockdown measures. Specifically, examining changes in motivation to self-isolate, the structure of daily life, as well as the psychosocial impact. The researchers identified a small cohort of general population, and took measurement while the lockdown was in place. These people were followed-up over two time points; specifically, at 2-weeks post-lockdown with an additional follow-up planned at 4-weeks post-lockdown.

1. The scientific validity of the research question(s)

The research addresses some interesting questions around the transitioning out of lockdowns. The study addresses this question in population that is ahead of other countries in terms of COVID-19 management and has the potential to offer a unique insight that feeds into our understanding of the impact national movement restrictions have on individuals and whether these are readily reversed with less stringent confinement measures, despite a lack of vaccine availability.

2. The logic, rationale, and plausibility of the proposed hypotheses

The psychosocial impact of a lockdown during a global pandemic is widely discussed currently and research investigating this is quickly being initiated. However, the slightly longer-term impact remains somewhat unaddressed. However, as countries will eventually transition of lockdowns, understanding how populations adapt is relevant.

It may also be useful to not only examine internal experience but also examine self-reported behaviour more closely (as this data is being collected) in order to provide preliminary insights into how populations behave in response to easing of restrictions. Thus, potentially informing transition strategies. I.e. understanding if people remain careful without external pressures may help understand if a gradual transition is necessary. It would be beneficial for the authors to discuss the impact of their research for extensively, over and above scientific inquiry. Examining behaviour may substantiate the potential impact.

It would seem more natural to frame this research in exploratory, observational terms rather than strongly hypothesis-driven. There are a lot of assumptions being made about establishing baseline severity which aren't strictly necessary if the interest is in a trend change over time, as indicated by the research questions. A later more qualitative comparison of baseline severity in Wuhan vs. other countries could be discussed. Given the sparsity of research in this area, any directed hypothesis may be premature. I.e. people may continue to have an intrinsic motivation to self-isolate within reasonable ramifications, which may be interesting in and of itself. Also, the evidence cited seems to somewhat contradict the last hypothesis so the reason for adhering to this hypothesis remains unclear.

3. The soundness and feasibility of the methodology and analysis pipeline

Generally, the research is sensible and feasible, as demonstrated by the successful collection of data so far. In places, more detail and greater justification of choices would be helpful.

There could be greater clarity on the design of the research study. From my understanding this is intended to be a small cohort of the general population that was recruited during lockdown and followed-up at two time points post-lockdown to assess a change in outcomes in lockdown vs. after lockdown. The timeframe could be specified more clearly early on describing when the lockdown was imposed, when it was lifted and at which time points measurements were taken. This only becomes somewhat apparent later on, with details of how long the lockdown lasted missing. The introduction conveys a sense that the first measurement was taken prior to lockdown.

The questionnaires are adapted so they may therefore lose some of their inherent properties; however, they appear to have face validity. I was unable to access the revised Loneliness Scale (so I am unable to comment on this) but it may be beneficial to explain why only the subscales were

used. It is unclear why not all measures were taken at all time points. However, all self-developed questions are open available to view.

The authors have chosen to opt for a multiple of t-tests, without consideration/explanation of the potential impact. An analysis of changes in raw scores over time rather than dichotomised outcomes based on the midpoints of questionnaires simpler and more intuitive. Furthermore, the analysis could be more parsimonious if the interest was more in the relative change over time, as stated by the research questions.

The authors may want to consider analysing all time points collectively. This could be easily implemented through mixed models, be it linear or additive in nature (depending on the expected trend). Additionally, this could allow for effects examined (i.e. it may be plausible to assume that motivation to self-isolate could vary by prior or present COVID-19 infection). However, if hypothesis driven research is preferred, a justification of statistical analysis would be useful as well as the potential limitations of the approach and why they were chosen nonetheless.

The power calculations appear logical. However, it bears the question of what kind of effect sizes you would expect to find. Framing and justifying the threshold used in terms of effect sizes found in previous literature would be helpful.

4. Clarity and degree of methodological detail for replication

The measures and analyses are explained in detail and most measures are openly available. The procedural elements of the methodology could be expanded more to allow for greater transparency and reproducibility. For example, are any measures being taken to follow-up non-responders as missing data from non-responders will likely be important to discuss? While the demographics are very useful, it would be beneficial to have a greater elaboration of the representativeness of your sample to the population you would like to extrapolate the results to and to what degree you have achieved this. I.e. the user base of WeChat, the types of “forums” being posted in, comparisons to demographic structure of Wuhan citizens, etc. This will contribute to an understanding of the population that is being recruited, which may ultimately speak to the generalisability of your findings. It would also allow for replication in other countries if i.e. cross-cultural variations are ever explored.

5. Potential identification of deviations from protocol

Similar to above, the report appears to be sufficiently explained to pinpoint deviations from the analysis. The procedural aspect of the methods could be elaborated on.

6. Sufficient outcome-neutral conditions (e.g. positive controls) for ensuring that the results obtained are able to test the stated hypotheses

Given the design of the research, there appears to be limited capacity to implement this. However, the additional check of asking people generally if their lives changed over time seems sensible.

Review form: Reviewer 3

Do you have any ethical concerns with this paper?

No

Recommendation?

Accept with minor revision

Comments to the Author(s)

The authors present an interesting study proposal on how life changes in the first month after COVID19 lockdowns have been lifted in Wuhan. The expected results, especially regarding motivation for self-isolation, will be important given the recent (sometimes controversial) public discussions on relaxing lockdown regulations and fear of a repeated spread of the virus. It will furthermore be interesting to see whether and how well-being and mental health changes on average. I want to compliment the authors on preparing this study in the midst of the COVID-19 crisis and for deciding to submit a Registered Report to undergo peer review before analysing the data.

Overall, the study's motivation and proposed methodology seem well-rounded and strong. Anyway, I will make some points that may help to improve the design and analyses in the following. I have ordered my points according to their importance for the current submission as a Stage 1 report:

- I understand the authors rationale for the onset hypotheses in self-isolation and structure of daily life measures. The current justification for the effect sizes of interest seem reasonable, but I still fear that the previous data that inform the criteria are not ideal. Maybe there just isn't enough data to inform a specific value that could be considered "normal" for self-isolation and structure of daily life measures? I would still find it very difficult to draw appropriate inferences on whether these measures are on average elevated during lockdown (T1) in the absence of a pre-lockdown measure or previous data (means and variances) of the population of interest (e.g. residents of Wuhan or China).

Thus, I might be inclined to prefer if the authors stick with their descriptive design and drop hypotheses 1.1 and 2.1. The focus on changes after the lockdown lift is anyway more interesting and more informative here, I find. (Please note that I think there is enough data to inform the onset hypotheses of the well-being measures, i.e. hypothesis 4.1., and perhaps 3.1.)

- One important question, at least regarding the mental health outcomes, is whether there was an announcement of the lockdown lift before or during data collection of T1? If so, one might expect that the anticipation of the lift already alleviated some psychological distress. Although I am not sure, it might even be possible that there psychological distress is lowest at T1, while it increases again slightly at T2 and / or T3.

- I really like that the authors propose different models for mean-level changes in their variables during their three time points. The planned analyses of the longitudinal data are what I would call "minimally sufficient" for the proposed research questions. However, conducting a range of t-tests brings about some issues, mainly multiple comparison issues. Have the authors considered alternative analyses that allow to analyse all three time points simultaneously, such as repeated-measures ANOVA or even better mixed-effects models (which allow to estimate the model with all available data)?

- I assume all of the proposed tests (and according power analyses) are two-sided?

- Regarding the effect sizes of interest and the power calculations: If I understand correctly, the authors chose relatively criteria for power analyses are fairly conservative ($\alpha = .001$, power = .95) and calculate the smallest Cohen's d that would be detectable given the (expected) sample size. This would be a value of $d = 0.3$. According to my calculations in R, d would even slightly higher given these criteria, namely $d = 0.33$:

```
>power.t.test(n = 226, sig.level = 0.001, power = 0.95,
+           type = "paired",
+           alternative = "two.sided")
```

Paired t test power calculation

```

n = 226
delta = 0.3323347
sd = 1
sig.level = 0.001
power = 0.95
alternative = two.sided

```

NOTE: n is number of *pairs*, sd is std.dev. of *differences* within pairs

Anyway, I think that $d = 0.33$ is a fairly high effect size for the effects to be studied. As there are few previous data to inform the determination of the smallest effect of interest, I would also consider less conservative criteria ($\alpha = .05$ and $\text{power} = .85$) appropriate to detect effects of $d < .20$. Alternatively, one could set α to $.025$ to correct for multiple comparisons (2 tests: T1 vs. T2; T2 vs. T3) for each set of trend analyses.

- Have there been previous studies on motivation during and after lockdown? If not, this aspect might be further highlighted as a unique strength, given its relevance for preventing a spread of COVID-19.
- There are some theories and previous research on psychological well-being and mental health that might be relevant to inform the current study (hypotheses 4.1. and hypotheses 4.2). In the depression literature, I would recommend to look into life events research. This paper might be a good starting point: <https://www.ncbi.nlm.nih.gov/pubmed/9046559>. There are some theories in the well-being literature that propose that well-being doesn't change permanently following life events, although recent studies show that there might be enduring effects. A good overview is probably this paper: <https://www.jstor.org/stable/20183166?seq=1>. Although of course a lockdown and its lift are unique events that haven't been considered before, it might be worth embedding the current study in this literature (at least regarding the well-being and mental health measures). It might also help to discuss a potential absence of effects on mental health.
- If I understand correctly, several scales that have been designed for this study (life structure) or have been used in this population for the first time (e.g. psychological need satisfaction)? In general, it would be important to provide some basic information on internal consistency (e.g. Cronbach's alpha) and validity for all measures, and these scales in particular. For example, it would be good to know to what extent the motivation for self-isolating measure correlates with actual self-isolating.
- Have the authors considered robust alternatives to t-tests if the data are skewed (which is to be expected, at least for the mental health data)?
- In the Stage 2 report, I would recommend providing a drop-out analysis, if you haven't planned to do so anyway. Drop-out analyses would be important to see whether people with certain characteristics preferentially dropped out of the study, which might bias the results and their interpretation.
- Have the authors considered an exploratory analysis to predict motivation for self-isolation at T2 and T3 (e.g. using multiple regression)? I could imagine that, for example, age, financial difficulties, and self-rated health might be associated with internal motivation to self-isolate.
- In the abstract (l. 37 or 38) it says "prior to lockdown life"; I assume it should read "prior to lockdown lift"?

- If possible, I would generally recommend that the authors share their data. Given that there are many studies being conducted in China and other countries, open data might help to allow cross-country comparisons and meta-analyses.

Review form: Reviewer 4

Do you have any ethical concerns with this paper?

No

Recommendation?

Major revision

Comments to the Author(s)

The proposed study aims to describe levels of motivation for self-isolation, structure of daily life, and quality of life in Wuhan during and after lockdown due to the Covid-19 pandemic. The study proposes collecting questionnaire data at 3 time points (T1 during lockdown, T2 shortly after lockdown was lifted, T3 further after lockdown was lifted). T1 and T2 data have already been collected; hence, I will not focus on evaluating or criticising the specific measures used for this Stage 1 Registered Report, as I acknowledge that these cannot be changed at this stage in the research, and I have no major concerns about them. The research question is an important and valid one, and the hypotheses are plausible and clearly described, but I have important concerns about the proposed analysis strategy, particularly the method by which conclusions are drawn from effect sizes, which I have outlined below:

1. The authors propose basing their conclusions on effect size, specifically Cohen's d . However, their proposed contrasts are all reliant on within-subject contrasts (paired samples t -tests or one sample t -tests). As such, the appropriate effect size would be d_{within} – note that this is calculated differently to d (see Lakens, 2013, *Frontiers in Psychology* for a description). If t -tests are to be computed, will non-parametric alternatives be used if assumptions such as normality are broken, and/or will any data transformations be attempted? If non-parametric tests are used, other effect sizes (e.g. r) may be more appropriate.
2. Furthermore, I am not clear that the proposed method for concluding a difference between time points was reached is appropriate. In section 5, the authors outline the results of a sensitivity power analysis ($\alpha = .001$, power = .95), based on their anticipated sample size of 226. I have a number of concerns about this strategy:
 - a. The research is described as 'descriptive', yet a number of empirical hypotheses are outlined, many with the criteria of observing $d = 0.3$ for concluding a difference between conditions. Given that the justification for $d = 0.3$ relies upon power analysis using an alpha level of .001, it would make just as much sense to rely on $p < .001$ to conclude a difference between conditions. I am aware that this is not typical of descriptive research, yet this is essentially how the analysis is framed anyway, as the alpha level of .001 used in the power analysis is used to calculate the 0.3 effect size figure. An alternative perhaps more in line with the ethos of descriptive research would be to report 95% (or 99%) confidence intervals for the observed effect sizes (which may also solve another problem, outlined below).
 - b. The authors outline a strategy for concluding whether any observed effects are linear ($d = 0.3$ between T1 and T2, then again $d = 0.3$ between T2 and T3), or non-linear ($d = 0.3$ between two conditions but not between the others). However, unless I have misunderstood, based on their proposed strategy, a T1-T2 difference of $d = 0.3$, then a T2-T3 difference of $d = 0.29$ would be interpreted as 'non-linear', which does not seem like a valid conclusion. Likewise, a T1-T2 difference of $d = 0.8$, followed by a T2-T3 difference of $d = 0.3$ would be interpreted as linear, which again would not seem valid. Without using more complex modelling, one approach, as

outlined above, could be to base conclusions on 95% confidence intervals around the effect size. So, for example, if the T1-T2 difference was $d = 0.4$ [CI = 0.3, 0.5] and T2-T3 difference was $d = 0.2$ [CI = 0.1, 0.3], this would provide some evidence for non-linearity (due to non-overlapping confidence intervals, yet both CIs not overlapping 0), whereas if the two effects were $d = 0.3$ [0.2, 0.4] and $d = 0.2$ [0.1, 0.3] this would not provide evidence for non-linearity. This approach would also solve the issue outlined in point c below re: justifying an effect size of 0.3 – in this approach, one would just rely on CIs not overlapping 0, rather than falling above or below an arbitrary effect size. There may be issues with the sample size leading to quite wide confidence intervals – it could be justified to relax to 90% confidence intervals if fully preregistered.

c. I am not convinced that settling on $d = 0.3$ is justified well in the manuscript. There is no justification that this effect size would be ‘meaningful’ (or that smaller effect sizes would not be meaningful). The only justification is based on the available sample size, yet this sample size was opportunistically gathered so does not provide justification for what is a ‘meaningful’ effect size in the real world. Note that my concern here is not with the opportunistic sample, but the justification of the effect size.

d. The anticipated figure of 226, based on 30% attrition at T2 and 30% attrition at T3 seems reasonable, though can the authors point to previous similar research to justify this figure? In this preregistration, the authors should provide a plan for if attrition is higher than this. For example, if attrition reaches 40% rather than 30%, the final sample size would be substantially lower ($N \sim 166$ compared to 226). At what point would attrition be judged ‘too high’?

3. There is also an issue of bias due to missing data. The authors plan to conduct some analysis on T1 data ($N = 462$) and some analysis on T1-2-3 data (anticipated $N = 226$), yet the two analyses may not be comparable if attrition is linked to any of the outcome measures. It is not inconceivable, for example, that participants suffering lower quality of life (e.g., the depression or loneliness measures) are more likely to drop out of the study. I suppose one method to deal with this would be to recalculate effects for the T1 data using only participants that did not later drop-out, though this of course leaves issues of generalizability. There are numerous other techniques for dealing with missing data (though I am not an expert in that area so can provide limited advice).

4. It should be acknowledged that the sample gained so far at T1 does not seem particularly representative of the wider Wuhan population, being mainly female. Presumably, there are also issues regarding who is likely to use the WeChat social media platform? With an unrepresentative demographic, at the very least the authors should clearly outline that we should be *very* cautious in generalizing this to the wider Chinese population (and even moreso to the whole world).

5. In a number of places, the manuscript proposes comparing to a midpoint of a Likert scale using a one-sample t-test, though there are a number of aspects of this that I find unclear.

a. Firstly, the authors repeatedly classify 3.5 as the midpoint of a 7-point scale and 3 as the midpoint of a 6-point scale, when presumably this should be 4 (1-2-3 and 5-6-7 on each side) and 3.5 (1-2-3 and 4-5-6 on each side) respectively? (Please correct me if I have misunderstood something here!).

b. Secondly, I assume that the authors plan to sum the scores from questionnaires and use a one-sample t-test to compare to the score a participant would receive if they responded at the midpoint for all items – if this is the case, this should be outlined and the specific scores to be compared against mentioned.

c. Thirdly, I think the justification for comparing against a midpoint is quite weak – for example, for hypothesis 3.1, the justification is that in the UK the ‘mean happiness’ was 6.8 out of 10, which the authors interpret as ‘the middle range’, though 6.8 does not seem to be in the middle! Furthermore, the extent to which the mean happiness of UK citizens (using a different scale) is comparable to Wuhan citizens is debatable. Is a mean score from previous research with each scale (in this case the Psychological Need Satisfaction scale) available, and if so could that be used as a benchmark instead?

Review form: Reviewer 5

Do you have any ethical concerns with this paper?

No

Recommendation?

Accept with minor revision

Comments to the Author(s)

Nguyen and colleagues propose an interesting and timely descriptive study of the psychological impacts of the government enforced lockdown in Wuhan, China, following the outbreak of COVID-19. Important considerations are below.

1. The scientific validity of the research question(s)

The authors have asked a number of important and timely research questions, which appear scientifically valid.

2. The logic, rationale, and plausibility of the proposed hypotheses

The proposed hypotheses are logical based on existing data and have been well justified by the authors.

3. The soundness and feasibility of the methodology and analysis pipeline (including statistical power analysis where applicable).

The power calculations are sound and appear to reflect estimated effect sizes reported in the literature.

4. Whether the clarity and degree of methodological detail would be sufficient to replicate exactly the proposed experimental procedures and analysis pipeline

The authors have done well to explain the variables used in the study. The nature of the circumstances of the study (a lockdown in a specific region precipitated by a viral outbreak) will make these results inherently difficult to replicate. However, the authors have done well to capture these data in a timely fashion and replication of their findings should be considered in light of these circumstances. The authors should include in their forthcoming manuscript exactly how all variables were handled/coded/transformed in analyses as well as some comments regarding possible clustering of participants within families/households (as this may impact independence of observations).

5. Whether the authors provide a sufficiently clear and detailed description of the methods to prevent undisclosed flexibility in the experimental procedures or analysis pipeline

It will be important for the authors to explicitly describe how variables are dealt with in this study and include a STROBE checklist for their reports of missing data, etc.

Additionally, I have some concern of some of the adapted items on the 'Self-Regulation Questionnaire', with respect to redundancy in both the "Identified motivation for self-isolation" and "External motivation for self-isolating". For example: "because self-isolation was beneficial to me" and "because it was beneficial to me" are almost identical, and "because I was afraid there would be harsh consequences if I didn't self-isolate" and "because I would get into trouble with others if I didn't" are also similar. The authors should explain how they will deal with these items.

Additionally, the authors should describe any possible relatedness between study participants (e.g. spouses, family members, shared household).

6. Whether the authors have considered sufficient outcome-neutral conditions (e.g. positive controls) for ensuring that the results obtained are able to test the stated hypotheses

Given the circumstances, I think it will be difficult for authors to consider outcome-neutral conditions (as all study participants will have undergone lock-down). However it would be helpful to provide comparisons with other reports (in Wuhan/China if these exist) regarding scores on the included measures to provide some context.

Decision letter (RSOS-200705.R0)

Dear Dr Nguyen,

The Editors assigned to your stage one Registered Report ("A COVID-19 descriptive study of life after lockdown in Wuhan, China") have now received comments from reviewers. We would like you to revise your paper in accordance with the referee and editors suggestions which can be found below (not including confidential reports to the Editor). Please note this decision does not guarantee eventual acceptance.

When submitting your revised manuscript, you must respond to the comments made by the referees and upload a file "Response to Referees" in "Section 2 - File Upload". Please use this to document how you have responded to the comments, and the adjustments you have made. In order to expedite the processing of the revised manuscript, please be as specific as possible in your response.

Kind regards,
Andrew Dunn
Royal Society Open Science
openscience@royalsociety.org

on behalf of Professor Chris Chambers (Registered Reports Editor, Royal Society Open Science)
openscience@royalsociety.org

Associate Editor Comments to Author (Professor Chris Chambers):

Associate Editor: 1

Comments to the Author:

The manuscript has now been assessed by five reviewers and a specialist editor. As you will see, the reviews are very detailed and constructive. All find merit in the proposal while also raising concerns within all areas of the Stage 1 review criteria, from the rationale underpinning the

hypotheses, the level of methodological detail (including a wide range of issues from the power analysis and justification of the effect size of interest to the handling of missing data), and clarity/consistency of presentation. Although the criticisms are substantial, they do fall within scope for a Stage 1 RR and therefore a Major Revision is invited. Please be sure to respond point-by-point to every comment of every reviewer and the specialist editor, whose comments you can find at the end of the letter.

Comments to Author:

Reviewer: 1

Comments to the Author(s)

I thank the authors for their submission. Overall, I think the submission targets a timely and important issue. The following questions, comments, suggestions, etc. are posed in hopes of improving or clarifying aspects of this submission.

1. It is unclear to me if the target population is restricted to adults only. If so, how do the authors ensure that respondents are adults? For example, is there a filtering question provided at the beginning of the survey?
2. How is the survey distributed? I understand participants are approached via the WeChat app. Is the survey hosted through another provider (e.g. SoSciSurvey or Qualtrics, or some other alternative)?
3. Similarly, where will the data be hosted? I see that the author are using the Open Science Framework to post their survey variables. I presume data would be hosted there?
4. What measures will be taken to ensure that the data remains anonymized/confidential; while still adhering to the requirement of making data/materials publicly available?
5. It is unclear to me how participants are tracked from Time 1 to Time 2 to Time 3. The authors appear to be using a form of convenience and/or snowball sampling through the WeChat app. I am unfamiliar with the app, so please excuse my ignorance. Will the participants need to have continued to use the app throughout the study for researchers to have access/connect to them? Or are there alternative ways of connecting once they have enrolled in the study? Further, is their a participant ID linked to the participant? Do they know this ID and re-enter it when completing Time 2 and Time 3? Or do the researchers connect the ID to the participant somehow (e.g. using WeChat profile information)?
6. The Registered Report proposal was submitted before Time 2 data collection was completed. Now that I am reviewing it, I believe data collection has stopped. Is there a way you can update us on what the sample size for Time 2 was? And further what % of participants remained in the study? This may be an important indicator for how many people continue through to Time 3 and has implications for the sensitivity analysis performed as part of the proposal. For example, it may affect how large the smallest effect size of interest is.
7. How will the authors handle missing data? Can participants skip questions or are they forced to answer each question before moving to the next?
8. Was there missing data for Time 1? What percentage? [This can also apply to Time 2 now that it has been collected].
9. Are participants being compensated in any way?

I hope these questions, comments, etc. have helped improve the proposal in some way.

Thank you for your consideration of them.

Take care,
Daniel Dunleavy

Reviewer: 2

Comments to the Author(s)

From my understanding, the authors have started data collection for an online study with citizens of Wuhan, where a strict lockdown was imposed and has been lifted. The aim of the research was to understand impact of easing lockdown measures. Specifically, examining changes in motivation to self-isolate, the structure of daily life, as well as the psychosocial impact. The researchers identified a small cohort of general population, and took measurement while the lockdown was in place. These people were followed-up over two time points; specifically, at 2-weeks post-lockdown with an additional follow-up planned at 4-weeks post-lockdown.

1. The scientific validity of the research question(s)

The research addresses some interesting questions around the transitioning out of lockdowns. The study addresses this question in population that is ahead of other countries in terms of COVID-19 management and has the potential to offer a unique insight that feeds into our understanding of the impact national movement restrictions have on individuals and whether these are readily reversed with less stringent confinement measures, despite a lack of vaccine availability.

2. The logic, rationale, and plausibility of the proposed hypotheses

The psychosocial impact of a lockdown during a global pandemic is widely discussed currently and research investigating this is quickly being initiated. However, the slightly longer-term impact remains somewhat unaddressed. However, as countries will eventually transition of lockdowns, understanding how populations adapt is relevant.

It may also be useful to not only examine internal experience but also examine self-reported behaviour more closely (as this data is being collected) in order to provide preliminary insights into how populations behave in response to easing of restrictions. Thus, potentially informing transition strategies. I.e. understanding if people remain careful without external pressures may help understand if a gradual transition is necessary. It would be beneficial for the authors to discuss the impact of their research for extensively, over and above scientific inquiry. Examining behaviour may substantiate the potential impact.

It would seem more natural to frame this research in exploratory, observational terms rather than strongly hypothesis-driven. There are a lot of assumptions being made about establishing baseline severity which aren't strictly necessary if the interest is in a trend change over time, as indicated by the research questions. A later more qualitative comparison of baseline severity in Wuhan vs. other countries could be discussed. Given the sparsity of research in this area, any directed hypothesis may be premature. I.e. people may continue to have an intrinsic motivation to self-isolate within reasonable ramifications, which may be interesting in and of itself. Also, the evidence cited seems to somewhat contradict the last hypothesis so the reason for adhering to this hypothesis remains unclear.

3. The soundness and feasibility of the methodology and analysis pipeline

Generally, the research is sensible and feasible, as demonstrated by the successful collection of data so far. In places, more detail and greater justification of choices would be helpful.

There could be greater clarity on the design of the research study. From my understanding this is intended to be a small cohort of the general population that was recruited during lockdown and followed-up at two time points post-lockdown to assess a change in outcomes in lockdown vs. after lockdown. The timeframe could be specified more clearly early on describing when the lockdown was imposed, when it was lifted and at which time points measurements were taken. This only becomes somewhat apparent later on, with details of how long the lockdown lasted missing. The introduction conveys a sense that the first measurement was taken prior to lockdown.

The questionnaires are adapted so they may therefore lose some of their inherent properties; however, they appear to have face validity. I was unable to access the revised Loneliness Scale (so I am unable to comment on this) but it may be beneficial to explain why only the subscales were used. It is unclear why not all measures were taken at all time points. However, all self-developed questions are open available to view.

The authors have chosen to opt for a multiple of t-tests, without consideration/explanation of the potential impact. An analysis of changes in raw scores over time rather than dichotomised outcomes based on the midpoints of questionnaires simpler and more intuitive. Furthermore, the analysis could be more parsimonious if the interest was more in the relative change over time, as stated by the research questions.

The authors may want to consider analysing all time points collectively. This could be easily implemented through mixed models, be it linear or additive in nature (depending on the expected trend). Additionally, this could allow for effects examined (i.e. it may be plausible to assume that motivation to self-isolate could vary by prior or present COVID-19 infection). However, if hypothesis driven research is preferred, a justification of statistical analysis would be useful as well as the potential limitations of the approach and why they were chosen nonetheless.

The power calculations appear logical. However, it bears the question of what kind of effect sizes you would expect to find. Framing and justifying the threshold used in terms of effect sizes found in previous literature would be helpful.

4. Clarity and degree of methodological detail for replication

The measures and analyses are explained in detail and most measures are openly available. The procedural elements of the methodology could be expanded more to allow for greater transparency and reproducibility. For example, are any measures being taken to follow-up non-responders as missing data from non-responders will likely be important to discuss? While the demographics are very useful, it would be beneficial to have a greater elaboration of the representativeness of your sample to the population you would like to extrapolate the results to and to what degree you have achieved this. I.e. the user base of WeChat, the types of "forums" being posted in, comparisons to demographic structure of Wuhan citizens, etc. This will contribute to an understanding of the population that is being recruited, which may ultimately speak to the generalisability of your findings. It would also allow for replication in other countries if i.e. cross-cultural variations are ever explored.

5. Potential identification of deviations from protocol

Similar to above, the report appears to be sufficiently explained to pinpoint deviations from the analysis. The procedural aspect of the methods could be elaborated on.

6. Sufficient outcome-neutral conditions (e.g. positive controls) for ensuring that the results obtained are able to test the stated hypotheses

Given the design of the research, there appears to be limited capacity to implement this.

However, the additional check of asking people generally if their lives changed over time seems sensible.

Reviewer: 3

Comments to the Author(s)

The authors present an interesting study proposal on how life changes in the first month after COVID19 lockdowns have been lifted in Wuhan. The expected results, especially regarding motivation for self-isolation, will be important given the recent (sometimes controversial) public discussions on relaxing lockdown regulations and fear of a repeated spread of the virus. It will furthermore be interesting to see whether and how well-being and mental health changes on average. I want to compliment the authors on preparing this study in the midst of the COVID-19 crisis and for deciding to submit a Registered Report to undergo peer review before analysing the data.

Overall, the study's motivation and proposed methodology seem well-rounded and strong. Anyway, I will make some points that may help to improve the design and analyses in the following. I have ordered my points according to their importance for the current submission as a Stage 1 report:

- I understand the authors rationale for the onset hypotheses in self-isolation and structure of daily life measures. The current justification for the effect sizes of interest seem reasonable, but I still fear that the previous data that inform the criteria are not ideal. Maybe there just isn't enough data to inform a specific value that could be considered "normal" for self-isolation and structure of daily life measures? I would still find it very difficult to draw appropriate inferences on whether these measures are on average elevated during lockdown (T1) in the absence of a pre-lockdown measure or previous data (means and variances) of the population of interest (e.g. residents of Wuhan or China).

Thus, I might be inclined to prefer if the authors stick with their descriptive design and drop hypotheses 1.1 and 2.1. The focus on changes after the lockdown lift is anyway more interesting and more informative here, I find. (Please note that I think there is enough data to inform the onset hypotheses of the well-being measures, i.e. hypothesis 4.1., and perhaps 3.1.)

- One important question, at least regarding the mental health outcomes, is whether there was an announcement of the lockdown lift before or during data collection of T1? If so, one might expect that the anticipation of the lift already alleviated some psychological distress. Although I am not sure, it might even be possible that there psychological distress is lowest at T1, while it increases again slightly at T2 and / or T3.

- I really like that the authors propose different models for mean-level changes in their variables during their three time points. The planned analyses of the longitudinal data are what I would call "minimally sufficient" for the proposed research questions. However, conducting a range of t-tests brings about some issues, mainly multiple comparison issues. Have the authors considered alternative analyses that allow to analyse all three time points simultaneously, such as repeated-measures ANOVA or even better mixed-effects models (which allow to estimate the model with all available data)?

- I assume all of the proposed tests (and according power analyses) are two-sided?

- Regarding the effect sizes of interest and the power calculations: If I understand correctly, the authors chose relatively criteria for power analyses are fairly conservative ($\alpha = .001$, power = .95) and calculate the smallest Cohen's d that would be detectable given the (expected) sample size. This would be a value of $d = 0.3$. According to my calculations in R, d would even slightly higher given these criteria, namely $d = 0.33$:

```
> power.t.test(n = 226, sig.level = 0.001, power = 0.95,
+             type = "paired",
+             alternative = "two.sided")
```

Paired t test power calculation

```
      n = 226
  delta = 0.3323347
      sd = 1
sig.level = 0.001
  power = 0.95
alternative = two.sided
```

NOTE: n is number of *pairs*, sd is std.dev. of *differences* within pairs

Anyway, I think that $d = 0.33$ is a fairly high effect size for the effects to be studied. As there are few previous data to inform the determination of the smallest effect of interest, I would also consider less conservative criteria ($\alpha = .05$ and $\text{power} = .85$) appropriate to detect effects of $d < .20$. Alternatively, one could set α to $.025$ to correct for multiple comparisons (2 tests: T1 vs. T2; T2 vs. T3) for each set of trend analyses.

- Have there been previous studies on motivation during and after lockdown? If not, this aspect might be further highlighted as a unique strength, given its relevance for preventing a spread of COVID-19.
- There are some theories and previous research on psychological well-being and mental health that might be relevant to inform the current study (hypotheses 4.1. and hypotheses 4.2). In the depression literature, I would recommend to look into life events research. This paper might be a good starting point: <https://www.ncbi.nlm.nih.gov/pubmed/9046559>. There are some theories in the well-being literature that propose that well-being doesn't change permanently following life events, although recent studies show that there might be enduring effects. A good overview is probably this paper: <https://www.jstor.org/stable/20183166?seq=1>. Although of course a lockdown and its lift are unique events that haven't been considered before, it might be worth embedding the current study in this literature (at least regarding the well-being and mental health measures). It might also help to discuss a potential absence of effects on mental health.
- If I understand correctly, several scales that have been designed for this study (life structure) or have been used in this population for the first time (e.g. psychological need satisfaction)? In general, it would be important to provide some basic information on internal consistency (e.g. Cronbach's alpha) and validity for all measures, and these scales in particular. For example, it would be good to know to what extent the motivation for self-isolating measure correlates with actual self-isolating.
- Have the authors considered robust alternatives to t-tests if the data are skewed (which is to be expected, at least for the mental health data)?
- In the Stage 2 report, I would recommend providing a drop-out analysis, if you haven't planned to do so anyway. Drop-out analyses would be important to see whether people with certain characteristics preferentially dropped out of the study, which might bias the results and their interpretation.
- Have the authors considered an exploratory analysis to predict motivation for self-isolation at T2 and T3 (e.g. using multiple regression)? I could imagine that, for example, age, financial difficulties, and self-rated health might be associated with internal motivation to self-isolate.

- In the abstract (l. 37 or 38) it says “prior to lockdown life”; I assume it should read “prior to lockdown lift”?

- If possible, I would generally recommend that the authors share their data. Given that there are many studies being conducted in China and other countries, open data might help to allow cross-country comparisons and meta-analyses.

Reviewer: 4

Comments to the Author(s)

The proposed study aims to describe levels of motivation for self-isolation, structure of daily life, and quality of life in Wuhan during and after lockdown due to the Covid-19 pandemic. The study proposes collecting questionnaire data at 3 time points (T1 during lockdown, T2 shortly after lockdown was lifted, T3 further after lockdown was lifted). T1 and T2 data have already been collected; hence, I will not focus on evaluating or criticising the specific measures used for this Stage 1 Registered Report, as I acknowledge that these cannot be changed at this stage in the research, and I have no major concerns about them. The research question is an important and valid one, and the hypotheses are plausible and clearly described, but I have important concerns about the proposed analysis strategy, particularly the method by which conclusions are drawn from effect sizes, which I have outlined below:

1. The authors propose basing their conclusions on effect size, specifically Cohen’s d . However, their proposed contrasts are all reliant on within-subject contrasts (paired samples t -tests or one sample t -tests). As such, the appropriate effect size would be d_{z} - note that this is calculated differently to d (see Lakens, 2013, *Frontiers in Psychology* for a description). If t -tests are to be computed, will non-parametric alternatives be used if assumptions such as normality are broken, and/or will any data transformations be attempted? If non-parametric tests are used, other effect sizes (e.g. r) may be more appropriate.

2. Furthermore, I am not clear that the proposed method for concluding a difference between time points was reached is appropriate. In section 5, the authors outline the results of a sensitivity power analysis ($\alpha = .001$, power = .95), based on their anticipated sample size of 226. I have a number of concerns about this strategy:

a. The research is described as ‘descriptive’, yet a number of empirical hypotheses are outlined, many with the criteria of observing $d = 0.3$ for concluding a difference between conditions. Given that the justification for $d = 0.3$ relies upon power analysis using an alpha level of .001, it would make just as much sense to rely on $p < .001$ to conclude a difference between conditions. I am aware that this is not typical of descriptive research, yet this is essentially how the analysis is framed anyway, as the alpha level of .001 used in the power analysis is used to calculate the 0.3 effect size figure. An alternative perhaps more in line with the ethos of descriptive research would be to report 95% (or 99%) confidence intervals for the observed effect sizes (which may also solve another problem, outlined below).

b. The authors outline a strategy for concluding whether any observed effects are linear ($d \geq 0.3$ between T1 and T2, then again $d \geq 0.3$ between T2 and T3), or non-linear ($d \geq 0.3$ between two conditions but not between the others). However, unless I have misunderstood, based on their proposed strategy, a T1-T2 difference of $d = 0.3$, then a T2-T3 difference of $d = 0.29$ would be interpreted as ‘non-linear’, which does not seem like a valid conclusion. Likewise, a T1-T2 difference of $d = 0.8$, followed by a T2-T3 difference of $d = 0.3$ would be interpreted as linear, which again would not seem valid. Without using more complex modelling, one approach, as outlined above, could be to base conclusions on 95% confidence intervals around the effect size. So, for example, if the T1-T2 difference was $d = 0.4$ [CI = 0.3, 0.5] and T2-T3 difference was $d = 0.2$ [CI = 0.1, 0.3], this would provide some evidence for non-linearity (due to non-overlapping confidence intervals, yet both CIs not overlapping 0), whereas if the two effects were $d = 0.3$ [0.2, 0.4] and $d = 0.2$ [0.1, 0.3] this would not provide evidence for non-linearity. This approach would also solve the issue outlined in point c below re: justifying an effect size of 0.3 - in this approach,

one would just rely on CIs not overlapping 0, rather than falling above or below an arbitrary effect size. There may be issues with the sample size leading to quite wide confidence intervals - it could be justified to relax to 90% confidence intervals if fully preregistered.

c. I am not convinced that settling on $d = 0.3$ is justified well in the manuscript. There is no justification that this effect size would be 'meaningful' (or that smaller effect sizes would not be meaningful). The only justification is based on the available sample size, yet this sample size was opportunistically gathered so does not provide justification for what is a 'meaningful' effect size in the real world. Note that my concern here is not with the opportunistic sample, but the justification of the effect size.

d. The anticipated figure of 226, based on 30% attrition at T2 and 30% attrition at T3 seems reasonable, though can the authors point to previous similar research to justify this figure? In this preregistration, the authors should provide a plan for if attrition is higher than this. For example, if attrition reaches 40% rather than 30%, the final sample size would be substantially lower ($N \sim 166$ compared to 226). At what point would attrition be judged 'too high'?

3. There is also an issue of bias due to missing data. The authors plan to conduct some analysis on T1 data ($N = 462$) and some analysis on T1-2-3 data (anticipated $N = 226$), yet the two analyses may not be comparable if attrition is linked to any of the outcome measures. It is not inconceivable, for example, that participants suffering lower quality of life (e.g., the depression or loneliness measures) are more likely to drop out of the study. I suppose one method to deal with this would be to recalculate effects for the T1 data using only participants that did not later drop-out, though this of course leaves issues of generalizability. There are numerous other techniques for dealing with missing data (though I am not an expert in that area so can provide limited advice).

4. It should be acknowledged that the sample gained so far at T1 does not seem particularly representative of the wider Wuhan population, being mainly female. Presumably, there are also issues regarding who is likely to use the WeChat social media platform? With an unrepresentative demographic, at the very least the authors should clearly outline that we should be *very* cautious in generalizing this to the wider Chinese population (and even moreso to the whole world).

5. In a number of places, the manuscript proposes comparing to a midpoint of a Likert scale using a one-sample t-test, though there are a number of aspects of this that I find unclear.

a. Firstly, the authors repeatedly classify 3.5 as the midpoint of a 7-point scale and 3 as the midpoint of a 6-point scale, when presumably this should be 4 (1-2-3 and 5-6-7 on each side) and 3.5 (1-2-3 and 4-5-6 on each side) respectively? (Please correct me if I have misunderstood something here!).

b. Secondly, I assume that the authors plan to sum the scores from questionnaires and use a one-sample t-test to compare to the score a participant would receive if they responded at the midpoint for all items - if this is the case, this should be outlined and the specific scores to be compared against mentioned.

c. Thirdly, I think the justification for comparing against a midpoint is quite weak - for example, for hypothesis 3.1, the justification is that in the UK the 'mean happiness' was 6.8 out of 10, which the authors interpret as 'the middle range', though 6.8 does not seem to be in the middle! Furthermore, the extent to which the mean happiness of UK citizens (using a different scale) is comparable to Wuhan citizens is debatable. Is a mean score from previous research with each scale (in this case the Psychological Need Satisfaction scale) available, and if so could that be used as a benchmark instead?

Reviewer: 5

Comments to the Author(s)

Nguyen and colleagues propose an interesting and timely descriptive study of the psychological impacts of the government enforced lockdown in Wuhan, China, following the outbreak of COVID-19. Important considerations are below.

1. The scientific validity of the research question(s)

The authors have asked a number of important and timely research questions, which appear scientifically valid.

2. The logic, rationale, and plausibility of the proposed hypotheses

The proposed hypotheses are logical based on existing data and have been well justified by the authors.

3. The soundness and feasibility of the methodology and analysis pipeline (including statistical power analysis where applicable).

The power calculations are sound and appear to reflect estimated effect sizes reported in the literature.

4. Whether the clarity and degree of methodological detail would be sufficient to replicate exactly the proposed experimental procedures and analysis pipeline

The authors have done well to explain the variables used in the study. The nature of the circumstances of the study (a lockdown in a specific region precipitated by a viral outbreak) will make these results inherently difficult to replicate. However, the authors have done well to capture these data in a timely fashion and replication of their findings should be considered in light of these circumstances. The authors should include in their forthcoming manuscript exactly how all variables were handled/coded/transformed in analyses as well as some comments regarding possible clustering of participants within families/households (as this may impact independence of observations).

5. Whether the authors provide a sufficiently clear and detailed description of the methods to prevent undisclosed flexibility in the experimental procedures or analysis pipeline

It will be important for the authors to explicitly describe how variables are dealt with in this study and include a STROBE checklist for their reports of missing data, etc.

Additionally, I have some concern of some of the adapted items on the 'Self-Regulation Questionnaire', with respect to redundancy in both the "Identified motivation for self-isolation" and "External motivation for self-isolating". For example: "because self-isolation was beneficial to me" and "because it was beneficial to me" are almost identical, and "because I was afraid there would be harsh consequences if I didn't self-isolate" and "because I would get into trouble with others if I didn't" are also similar. The authors should explain how they will deal with these items.

Additionally, the authors should describe any possible relatedness between study participants (e.g. spouses, family members, shared household).

6. Whether the authors have considered sufficient outcome-neutral conditions (e.g. positive controls) for ensuring that the results obtained are able to test the stated hypotheses

Given the circumstances, I think it will be difficult for authors to consider outcome-neutral conditions (as all study participants will have undergone lock-down). However it would be helpful to provide comparisons with other reports (in Wuhan/China if these exist) regarding scores on the included measures to provide some context.

Reviewer: Specialist Editor

The registered report of Nguyen and colleagues addresses the question of how life changes after the COVID-19 related lockdown. In particular, over three time points the following aspects will be investigated: changes in motivation to self-isolate, changes in structure of daily life and changes in psychological well-being. The study is surely relevant and timely, aiming at observing

the post-lockdown effects in the first city publicly affected by the COVID crisis and one with the stringent measures.

I have several suggestions that the authors should considered in the compilation of their report.

- 1) While the time points of measurement are stated in the abstract, it is only later in the main text that it become clear what t1, t2 and t3 refer to. I would suggest to make them explicit (with dates) already in the introduction
- 2) Related to this, the use of the verb tense is changing from past to future (we collected/we will collect) for the same time points, making difficult to understand if they refer to the same thing. Please use the same tense throughout the manuscript
- 3) Descriptive measures: the division of descriptive measures vs. dependent variables needs to be better specified. While I agree that sociodemographic data can be considered descriptive measure, I do not understand why variables such us 4.6.9.and 10. are not analyzed as dependent variables. Especially changes tracked by item 9. And 10. seem to be particular relevant for later dependent variables (such as psychological well-being)
- 4) If any relationship between descriptive variables and dependent variables is expected, should be (at least briefly) addressed in the report
- 5) In the main text, a change in psychological well-being toward negative outcomes, as possible effect of an economic crisis , is mentioned but not elaborated anymore in the hypotheses section. I think this point is very relevant and I would recommend to consider it further.
- 6) A rationale for the distance between t2 and t3 measure should be provided.
- 7) What is the minimum age for taking part to the survey? Or this is not an inclusion criteria

Author's Response to Decision Letter for (RSOS-200705.R0)

See Appendix A.

RSOS-200705.R1 (Revision)

Review form: Reviewer 1 (Daniel Dunleavy)

Do you have any ethical concerns with this paper?

No

Recommendation?

Accept in principle

Comments to the Author(s)

I thank the authors for their timely response and thoughtful consideration of my feedback. Below are a few more questions and considerations.

1. In the abstract (p. 3) it states:

"We have collected data from 462 participants in Wuhan, China, prior to lockdown lift between the 3rd and 7th of April, 2020 (Time 1), and have followed up with another wave of data collection between 18th and 22nd of April, 2020 (Time 2), and between 6th and 10th of May, 2020 (Time 3). We will continue to collect one more wave of data between 25th and 29th of May, 2020 (Time 4)."

I would suggest amending this to reflect the number of participants collected for Time 2 (i.e. $n = 293$) and Time 3 ($n = 342$).

2. Missing data.

On page 25, I would simply recommend being just a bit more explicit in your reporting of what you are comparing. In other words, just being a bit more clear in what timepoint and which groups you are comparing. By all means, it could just be me, but I found it to be a bit tricky to keep track of the total sample for each timepoint, the number of new participants, the number included in the missing data comparison, and how it might relate to previous timepoints. The table on page 21-22 did help in this regard, but if there is anyway to make that section a bit more clear, I think would be helpful.

Thank you again and take care,
Daniel Dunleavy

Review form: Reviewer 2

Do you have any ethical concerns with this paper?

No

Recommendation?

Accept with minor revision

Comments to the Author(s)

The authors have made substantial changes to the protocol addressing the previous limitations. The rationale is still very relevant and the procedural detail is much improved statistical methodology appears more fitting now.

However, it is unclear as to why new participants were recruited at each stage. I believe it is more common to have a sample at baseline and then only follow these participants up over time. Or, alternatively sampling a cross-sectional population at each time point as previous surveys have done. It would be good to have a rationale for combining the two and also provide some information on how many people from the original cohort were retained at each time point. This also leads to some uncertainty around how much missing data is present for each participant. It can be problematic to not have adequate baseline to primary follow-up time data, even when using methods to account for missingness. I.e. It would be problematic to have a participant who only contributed data at T3. Also, some consideration to the amount of missing data might be given when using methods to address this problem, especially when missingness is expected to be high (from what I understand ~43% , $n = 200$ at T4, of the 462 participants who completed the survey at baseline are expected to respond across all time points).

The model building process could be elaborated on slightly further, including planned model evaluation and slightly more clarity on idiosyncrasies of the models for each research question. I.e. clarifying which models are being built for which hypothesis - is model 7 be relevant to hypothesis 1-3? Given that your data will be open access, it would be good to state software and make your annotated code openly available too to make the analysis pipeline transparent and reproducible.

Review form: Reviewer 3

Do you have any ethical concerns with this paper?

No

Recommendation?

Accept with minor revision

Comments to the Author(s)

I think that the revised stage 1 report has considerably improved by the revision. I especially think that dropping the baseline hypotheses and switching to growth models to model the observed changes will make the study stronger. I also thank the authors for addressing all my comments point.

I only have two small comments: First, Cronbach's alpha, as now provided, is better described as "internal consistency" (rather than internal reliability).

Second, it would be good to be more explicitly how the authors want to discern no linear change, linear increase, and linear decrease. The first (no linear change vs. linear change) should be reflects in model comparison 3 vs. 4. it's good to note that an increase would be reflected in a positive coefficient (slope intercept), and a decrease in a negative one.

Review form: Reviewer 4

Do you have any ethical concerns with this paper?

No

Recommendation?

Accept in principle

Comments to the Author(s)

I think this revised manuscript is much improved on the first submission. My main concerns from the first submission regarded the use of effect sizes and t-tests, so the inclusion of latent growth models has thus improved the analysis strategy. Although I have never used LGMs (and so cannot comment in detail on the proposed specifics of the modelling), this usage seems appropriate and well detailed. A number of the hypotheses which I had queries about have been removed or replaced, which I also think improves this manuscript, and it is good to see that so far there has been a relatively low attrition rate across time points, removing my concerns about possible high attrition. The addition of a fourth time point also seems worthwhile.

I would like to acknowledge that the authors have done a lot of work to improve this manuscript in a relatively short amount of time, and have responded in detail to all reviewers. I believe this will be an important and interesting study and will look forward to reviewing the eventual Stage 2 manuscript.

Review form: Reviewer 5

Do you have any ethical concerns with this paper?

No

Recommendation?

Accept in principle

Comments to the Author(s)

The authors have done a nice job responding to my comments. I have no further critiques and wish them well.

Decision letter (RSOS-200705.R1)

Dear Dr Nguyen,

On behalf of the Editors, I am pleased to inform you that your Manuscript RSOS-200705.R1 entitled "A COVID-19 descriptive study of life after lockdown in Wuhan, China" has been accepted in principle for publication in Royal Society Open Science subject to minor revision in accordance with the referee and editor suggestions. Please find their comments at the end of this email.

The reviewers and handling editors have recommended publication, but also suggest some minor revisions to your manuscript. Therefore, I invite you to respond to the comments and revise your manuscript.

Please you submit the revised version of your manuscript within 7 days (i.e. by the 04-Jun-2020). If you do not think you will be able to meet this date please let me know immediately.

When submitting your revised manuscript, you will be able to respond to the comments made by the referees and you should upload a file "Response to Referees". You can use this to document any changes you make to the original manuscript. In order to expedite the processing of the revised manuscript, please be as specific as possible in your response to the referees.

Full author guidelines can be found here <https://royalsocietypublishing.org/rsos/registered-reports>.

Kind regards,

Royal Society Open Science Editorial Office
Royal Society Open Science
openscience@royalsociety.org

on behalf of Professor Chris Chambers (Subject Editor, Royal Society Open Science)
openscience@royalsociety.org

Associate Editor Comments to Author (Professor Chris Chambers):

Associate Editor: 1

Comments to the Author:

The revised manuscript was returned to the five specialist reviewers who assessed the original Stage 1 submission. The reviewers are now broadly positive and the manuscript is within striking reach of IPA. There are still some relatively minor issues to address concerning the clarity and detail of reporting (see remaining comments of Reviewers 1, 2 and 3) but provided the authors are able to respond comprehensively to these remaining issues in a final revision, IPA should be forthcoming without requiring further in-depth review.

Reviewer comments to Author:

Reviewer: 1

Comments to the Author(s)

I thank the authors for their timely response and thoughtful consideration of my feedback. Below are a few more questions and considerations.

1. In the abstract (p. 3) it states:

"We have collected data from 462 participants in Wuhan, China, prior to lockdown lift between the 3rd and 7th of April, 2020 (Time 1), and have followed up with another wave of data collection between 18th and 22nd of April, 2020 (Time 2), and between 6th and 10th of May, 2020 (Time 3). We will continue to collect one more wave of data between 25th and 29th of May, 2020 (Time 4)."

I would suggest amending this to reflect the number of participants collected for Time 2 (i.e. $n = 293$) and Time 3 ($n = 342$).

2. Missing data.

On page 25, I would simply recommend being just a bit more explicit in your reporting of what you are comparing. In other words, just being a bit more clear in what timepoint and which groups you are comparing. By all means, it could just be me, but I found it to be a bit tricky to keep track of the total sample for each timepoint, the number of new participants, the number included in the missing data comparison, and how it might relate to previous timepoints. The table on page 21-22 did help in this regard, but if there is anyway to make that section a bit more clear, I think would be helpful.

Thank you again and take care,
Daniel Dunleavy

Reviewer: 2

Comments to the Author(s)

The authors have made substantial changes to the protocol addressing the previous limitations. The rationale is still very relevant and the procedural detail is much improved statistical methodology appears more fitting now.

However, it is unclear as to why new participants were recruited at each stage. I believe it is more common to have a sample at baseline and then only follow these participants up over time. Or, alternatively sampling a cross-sectional population at each time point as previous surveys have done. It would be good to have a rationale for combining the two and also provide some information on how many people from the original cohort were retained at each time point. This

also leads to some uncertainty around how much missing data is present for each participant. It can be problematic to not have adequate baseline to primary follow-up time data, even when using methods to account for missingness. I.e. It would be problematic to have a participant who only contributed data at T3. Also, some consideration to the amount of missing data might be given when using methods to address this problem, especially when missingness is expected to be high (from what I understand ~43% , n= 200 at T4, of the 462 participants who completed the survey at baseline are expected to respond across all time points).

The model building process could be elaborated on slightly further, including planned model evaluation and slightly more clarity on idiosyncrasies of the models for each research question. I.e. clarifying which models are being built for which hypothesis - is model 7 relevant to hypothesis 1-3? Given that your data will be open access, it would be good to state software and make your annotated code openly available too to make the analysis pipeline transparent and reproducible.

Reviewer: 3

Comments to the Author(s)

I think that the revised stage 1 report has considerably improved by the revision. I especially think that dropping the baseline hypotheses and switching to growth models to model the observed changes will make the study stronger. I also thank the authors for addressing all my comments point.

I only have two small comments: First, Cronbach's alpha, as now provided, is better described as "internal consistency" (rather than internal reliability).

Second, it would be good to be more explicitly how the authors want to discern no linear change, linear increase, and linear decrease. The first (no linear change vs. linear change) should be reflected in model comparison 3 vs. 4. it's good to note that an increase would be reflected in a positive coefficient (slope intercept), and a decrease in a negative one.

Reviewer: 4

Comments to the Author(s)

I think this revised manuscript is much improved on the first submission. My main concerns from the first submission regarded the use of effect sizes and t-tests, so the inclusion of latent growth models has thus improved the analysis strategy. Although I have never used LGMs (and so cannot comment in detail on the proposed specifics of the modelling), this usage seems appropriate and well detailed. A number of the hypotheses which I had queries about have been removed or replaced, which I also think improves this manuscript, and it is good to see that so far there has been a relatively low attrition rate across time points, removing my concerns about possible high attrition. The addition of a fourth time point also seems worthwhile.

I would like to acknowledge that the authors have done a lot of work to improve this manuscript in a relatively short amount of time, and have responded in detail to all reviewers. I believe this will be an important and interesting study and will look forward to reviewing the eventual Stage 2 manuscript.

Author's Response to Decision Letter for (RSOS-200705.R1)

See Appendix B.

Decision letter (RSOS-200705.R2)

Dear Dr Nguyen,

On behalf of the Editor, I am pleased to inform you that your Manuscript RSOS-200705.R2 entitled "A COVID-19 descriptive study of life after lockdown in Wuhan, China" has been accepted in principle for publication in Royal Society Open Science.

Please read the following email carefully

Your accepted Stage 1 manuscript has been registered under a 2-year private embargo at <https://osf.io/g2t3b>

This embargo will be released by the journal (and the Stage 1 protocol made fully public) at the point of Stage 2 submission or formal withdrawal. Please include the URL to this registered protocol in your eventual Stage 2 manuscript.

Following completion of your study, we invite you to resubmit your paper for peer review as a Stage 2 Registered Report. Please note that your manuscript can still be rejected for publication at Stage 2 if the editors consider any of the following conditions to be met:

- The results were unable to test the authors' proposed hypotheses by failing to meet the approved outcome-neutral criteria.
- The authors altered the Introduction, rationale, or hypotheses, as approved in the Stage 1 submission.
- The authors failed to adhere closely to the registered study procedures. Please note that any deviations from the approved study procedures must be communicated to the editor immediately for approval, and prior to the completion of data collection. Failure to do so can result in revocation of in-principle acceptance and rejection at Stage 2 (see complete guidelines for further information).
- Any post-hoc (unregistered) analyses were either unjustified, insufficiently caveated, or overly dominant in shaping the authors' conclusions.
- The authors' conclusions were not justified given the data obtained.

We encourage you to read the complete guidelines for authors concerning Stage 2 submissions at <https://royalsocietypublishing.org/rsos/registered-reports#ReviewerGuideRegRep>. Please especially note the requirements for data sharing, reporting the URL of the registered protocol, and that withdrawing your manuscript will result in publication of a Withdrawn Registration.

Once again, thank you for submitting your manuscript to Royal Society Open Science and we look forward to receiving your Stage 2 submission. If you have any questions at all, please do not hesitate to get in touch. We look forward to hearing from you shortly with the anticipated submission date for your stage two manuscript.

on behalf of Professor Chris Chambers (Registered Reports Editor, Royal Society Open Science)
 openscience@royalsociety.org

Author's Response to Decision Letter for (RSOS-200705.R2)

See Appendix C.

RSOS-200705.R3 (Revision)

Review form: Reviewer 1 (Daniel Dunleavy)

Is the manuscript scientifically sound in its present form?

Yes

Are the interpretations and conclusions justified by the results?

Yes

Is the language acceptable?

Yes

Do you have any ethical concerns with this paper?

No

Have you any concerns about statistical analyses in this paper?

No

Recommendation?

Accept with minor revision

Comments to the Author(s)

I thank the authors for their hard work on this Registered Report submission. The following comments, suggestions, and questions are meant to help clarify and potentially strengthen aspects of the manuscript.

"Whether the data are able to test the authors' proposed hypotheses by passing the approved outcome-neutral criteria (such as absence of floor and ceiling effects or success of positive controls)"

-I have no concerns on this particular point.

"Whether the Introduction, rationale and stated hypotheses are the same as the approved Stage 1 submission"

-The introduction, rationale, hypotheses and other Stage 1 materials appear to be the same in both submissions, with the exception of the authors having changed verbage from present/future tense to past tense. No major changes (additions/omissions) are noted.

"Whether the authors adhered precisely to the registered experimental procedures"

-I have no concerns on this particular point. Additionally, all code and materials appear to be supplied and are functional.

"Where applicable, whether any unregistered exploratory statistical analyses are justified, methodologically sound, and informative"

-No issues noted on this point. There appear to be no obvious deviations from the analytic plan. The section on missing data analysis is clear and explicit.

"Whether the authors' conclusions are justified given the data"

-I think the authors did a good job reporting the study's limitations. Particularly, I felt they were very transparent and clear, when they stated:

p. 43, lines 4-14: "What we do not know with this data is whether this gradual increase begun after lockdown was lifted or whether change has occurred during lockdown. In other words, this result might be evidence of psychological resilience under the tremendous transition of self isolation (Luchetti, et.al, 2020), and our participants may have found suitable ways to adjust their lives over time and thus have experienced gentler transition from lockdown to normalcy. "

Further, p. 44 goes into good detail about how this study differs from other efforts (e.g. Gao et al. 2020).

My only major feedback to the discussion section, would be to add more detailed content describing how these concepts (e.g. isolation, coping, resilience) may be further investigated in future research or follow-up studies, by use of qualitative or other interviewing methods....methods that may complement this study and help triangulate the results found here.

Other comments:

Reporting

Pages 51-66: While I'm not a huge proponent of using asterisks in tables, I'd suggest that (for each table that uses asterisks for reporting) you include a note at the bottom explicitly stating what each refers to. This is done in some places, but not others.

Page 71: To increase clarity, I might suggest in Appendix B, that the three notes stating: "1. "Participants who filled...were taken into consideration." be phrased as "...are reported here."

Review form: Reviewer 4

Is the manuscript scientifically sound in its present form?

Yes

Are the interpretations and conclusions justified by the results?

Yes

Is the language acceptable?

Yes

Do you have any ethical concerns with this paper?

No

Have you any concerns about statistical analyses in this paper?

No

Recommendation?

Accept with minor revision

Comments to the Author(s)

This Stage 2 Registered Report investigates changes in motivation for isolation, life structure, coping strategies, and mental health variables (depression, loneliness) over time, during and after lockdown in Wuhan, China, in relation to the covid-19 pandemic. The protocol outlined at Stage 1 has been adhered to, the data are able to test the preregistered hypotheses, procedures have been adhered to, and the conclusions are justified by the data (though see one very minor suggestion below), and the authors report and discuss their findings clearly. I have run R code available on OSF, which is easily accessible and clearly commented, and easily managed to reproduce the analysis and figures. I only have a couple of very minor points to make (see below), but otherwise am happy for this to be accepted.

1. The descriptions of internal reliability for some of the self-report measures (page 11 onwards) seem to have been removed (or moved, and I can't find them!) – I think these were useful for the reader to assess the measures, and should be included if possible.
2. I think the conclusion in the very final paragraph regarding mental health (“this set of findings do not yield evidence to support the current media concerns around rise in mental health issues... after imposed lockdown”) is perhaps over-stated. Given that there was no pre-lockdown assessment of the mental health variables, it is hard to conclude whether there was a lockdown-related rise (which has remained steady across the four timepoints) or no change at all. This is discussed well by the authors earlier in the discussion (with reference to the Gao et al., 2020 study), so it is only this final summary sentence that I think should be toned down a bit.
3. I think it may also be worth noting in the discussion further limitations of the conclusions that can be drawn about mental health – for example, we can't say much about long term effect on mental health, nor the effects on vulnerable populations (e.g., children), as has been speculated upon in the media.

Typos

- P26, line 50 – presumably should read ‘sample of 200’ not ‘sample of 2’.
- P39 – ‘table x’ should be replaced with the correct table number.
- P43, line 38 – should read ‘the current study’, not ‘current study’
- P43, line 48 – should read ‘isolation’ not ‘solation’.

Decision letter (RSOS-200705.R3)

Dear Dr Nguyen:

On behalf of the Editor, I am pleased to inform you that your Stage 2 Registered Report RSOS-200705.R3 entitled "A COVID-19 descriptive study of life after lockdown in Wuhan, China" has been deemed suitable for publication in Royal Society Open Science subject to minor revision in

accordance with the referee suggestions. Please find the referees' comments at the end of this email.

The reviewers and Subject Editor have recommended publication, but also suggest some minor revisions to your manuscript. Therefore, I invite you to respond to the comments and revise your manuscript.

Please also ensure that all the below editorial sections are included where appropriate -- if any section is not applicable to your manuscript, please can we ask you to nevertheless include the heading, but explicitly state that the heading is inapplicable. An example of these sections is attached with this email.

- Ethics statement

- Data accessibility

If you wish to submit your supporting data or code to Dryad (<http://datadryad.org/>), or modify your current submission to dryad, please use the following link:
[http://datadryad.org/submit?journalID=RSOS&manu=\(Document not available\)](http://datadryad.org/submit?journalID=RSOS&manu=(Document not available))

- Competing interests

- Authors' contributions

- Acknowledgements

- Funding statement

Because the schedule for publication is very tight, it is a condition of publication that you submit the revised version of your manuscript within 7 days (i.e. by the 14-Aug-2020). If you do not think you will be able to meet this date please let me know immediately.

Kind regards,

Andrew Dunn

on behalf of Professor Chris Chambers
 (Registered Reports Editor, Royal Society Open Science)
 openscience@royalsociety.org

Associate Editor Comments to Author (Professor Chris Chambers):

Associate Editor: 1

Comments to the Author:

The Stage 2 manuscript was returned to two of the five reviewers and the specialist associate editor who assessed the Stage 1 submission. The assessments are positive overall, and the Stage 2 criteria are essentially met. The reviewers and specialist AE do, however, make some useful suggestions for improving clarity in the final stretch, ensuring comprehensive reporting, and enhancing the Discussion.

In addition to addressing the reviewers' comments, please also update the Abstract to summarise the results and conclusions. Also, please move the study design table from Appendix A to either the main text of the Methods, or (better yet in my opinion), update the table to include an extra column on the right that states the *outcome* of each test and place the table at or near the start of the Results section (changing the table from portrait to landscape mode, and reducing the font size, would help fit in this extra information while keeping the table readable). I think an augmented design table with results would be useful addition to the Stage 2 manuscript that will help readers quickly digest the questions posed and answers obtained, but I will leave the authors to decide which approach works best for them (either including the current design table as-is in the Method, or an enhanced version in the Results). Either way, please move it from the Appendix to the main text.

Provided the authors are able to respond comprehensively to all points raised in a revision, Stage 2 acceptance should be forthcoming without requiring further in-depth review.

Comments to Author:

Reviewer: 1

Comments to the Author(s)

I thank the authors for their hard work on this Registered Report submission. The following comments, suggestions, and questions are meant to help clarify and potentially strengthen aspects of the manuscript.

"Whether the data are able to test the authors' proposed hypotheses by passing the approved outcome-neutral criteria (such as absence of floor and ceiling effects or success of positive controls)"

-I have no concerns on this particular point.

"Whether the Introduction, rationale and stated hypotheses are the same as the approved Stage 1 submission"

-The introduction, rationale, hypotheses and other Stage 1 materials appear to be the same in both submissions, with the exception of the authors having changed verbage from present/future tense to past tense. No major changes (additions/omissions) are noted.

"Whether the authors adhered precisely to the registered experimental procedures"

-I have no concerns on this particular point. Additionally, all code and materials appear to be supplied and are functional.

"Where applicable, whether any unregistered exploratory statistical analyses are justified, methodologically sound, and informative"

-No issues noted on this point. There appear to be no obvious deviations from the analytic plan. The section on missing data analysis is clear and explicit.

"Whether the authors' conclusions are justified given the data"

-I think the authors did a good job reporting the study's limitations. Particularly, I felt they were very transparent and clear, when they stated:

p. 43, lines 4-14: "What we do not know with this data is whether this gradual increase begun after lockdown was lifted or whether change has occurred during lockdown. In other words, this result might be evidence of psychological resilience under the tremendous transition of self isolation (Luchetti, et.al, 2020), and our participants may have found suitable ways to adjust their lives over time and thus have experienced gentler transition from lockdown to normalcy. "

Further, p. 44 goes into good detail about how this study differs from other efforts (e.g. Gao et al. 2020).

My only major feedback to the discussion section, would be to add more detailed content describing how these concepts (e.g. isolation, coping, resilience) may be further investigated in future research or follow-up studies, by use of qualitative or other interviewing methods....methods that may complement this study and help triangulate the results found here.

Other comments:

Reporting

Pages 51-66: While I'm not a huge proponent of using asterisks in tables, I'd suggest that (for each table that uses asterisks for reporting) you include a note at the bottom explicitly stating what each refers to. This is done in some places, but not others.

Page 71: To increase clarity, I might suggest in Appendix B, that the three notes stating: "1. "Participants who filled...were taken into consideration." be phrased as "...are reported here."

Reviewer: 4

Comments to the Author(s)

This Stage 2 Registered Report investigates changes in motivation for isolation, life structure, coping strategies, and mental health variables (depression, loneliness) over time, during and after lockdown in Wuhan, China, in relation to the covid-19 pandemic. The protocol outlined at Stage 1 has been adhered to, the data are able to test the preregistered hypotheses, procedures have been adhered to, and the conclusions are justified by the data (though see one very minor suggestion below), and the authors report and discuss their findings clearly. I have run R code available on OSF, which is easily accessible and clearly commented, and easily managed to reproduce the analysis and figures. I only have a couple of very minor points to make (see below), but otherwise am happy for this to be accepted.

1. The descriptions of internal reliability for some of the self-report measures (page 11 onwards) seem to have been removed (or moved, and I can't find them!) – I think these were useful for the reader to assess the measures, and should be included if possible.

2. I think the conclusion in the very final paragraph regarding mental health (“this set of findings do not yield evidence to support the current media concerns around rise in mental health issues... after imposed lockdown”) is perhaps over-stated. Given that there was no pre-lockdown assessment of the mental health variables, it is hard to conclude whether there was a lockdown-related rise (which has remained steady across the four timepoints) or no change at all. This is discussed well by the authors earlier in the discussion (with reference to the Gao et al., 2020 study), so it is only this final summary sentence that I think should be toned down a bit.

3. I think it may also be worth noting in the discussion further limitations of the conclusions that can be drawn about mental health – for example, we can’t say much about long term effect on mental health, nor the effects on vulnerable populations (e.g., children), as has been speculated upon in the media.

Typos

- P26, line 50 – presumably should read ‘sample of 200’ not ‘sample of 2’.
- P39 – ‘table x’ should be replaced with the correct table number.
- P43, line 38 – should read ‘the current study’, not ‘current study’
- P43, line 48 – should read ‘isolation’ not ‘solation’.

Specialist Associate Editor

Comments to the Author(s):

One thing I noticed is that at p.7 line 50 onward (Measures for descriptive analyses), they mention only t1, t2 and t3, but the measures were also collected at t4. they should correct that. Also for consistency, I would report frequencies and means for all the mentioned measure in each subsection (and not only in the appendix) at each time point (t1 to t4).

Author's Response to Decision Letter for (RSOS-200705.R3)

See Appendix D.

Decision letter (RSOS-200705.R4)

Dear Dr Nguyen:

It is a pleasure to accept your Stage 2 Registered Report entitled "A COVID-19 descriptive study of life after lockdown in Wuhan, China" in its current form for publication in Royal Society Open Science.

Please ensure that you send to the editorial office an editable version of your accepted manuscript, and individual files for each figure and table included in your manuscript. You can send these in a zip folder if more convenient. Failure to provide these files may delay the processing of your proof.

You can expect to receive a proof of your article in the near future. Please contact the editorial office (openscience_proofs@royalsociety.org) and the production office (openscience@royalsociety.org) to let us know if you are likely to be away from e-mail contact -- if

you are going to be away, please nominate a co-author (if available) to manage the proofing process, and ensure they are copied into your email to the journal.

COVID-19 rapid publication process:

We are taking steps to expedite the publication of research relevant to the pandemic. If you wish, you can opt to have your paper published as soon as it is ready, rather than waiting for it to be published the scheduled Wednesday.

This means your paper will not be included in the weekly media round-up which the Society sends to journalists ahead of publication. However, it will still appear in the COVID-19 Publishing Collection which journalists will be directed to each week (<https://royalsocietypublishing.org/topic/special-collections/novel-coronavirus-outbreak>).

If you wish to have your paper considered for immediate publication, or to discuss further, please notify openscience_proofs@royalsociety.org and press@royalsociety.org when you respond to this email.

on behalf of Professor Chris Chambers (Subject Editor)
openscience@royalsociety.org

Appendix A

Dear Prof. Chambers,

We are happy to be submitting a majorly revised version of the Stage 1 manuscript. We have made the following revisions to this manuscript:

1. We have added two additional questions to identify individual differences that predict different slope of change in well-being (basic psychological need satisfaction) and ill-being (loneliness and depression) over time. Specifically, we will investigate coping strategies used during lockdown and stressors experienced during lockdown as covariates to predict average levels of well-being and ill-being as well as change over time of these outcome variables.
2. We have added a Time 4 assessment to our study. So, now there will be 4 assessments administered 2 weeks apart. In total, this means that we will obtain data 4 times within the first 2 months after lockdown was lifted in Wuhan, China.
3. We have made major revisions to our analytical plan. Instead of using paired t-test to compare means of any given 2 time points, we will use latent growth models to investigate overall change across 4 time points. This statistical approach has greater statistical power than paired t-tests and repeated-measures ANOVA and also allows for analyses with missing data.

Changes have been made to the manuscript in “red ink”.

We found the reviewer recommendations to be extremely constructive, and believe the manuscript and study is much improved as a result. Please see our responses to each reviewer’s comments below.

Best wishes,

Thuy-vy Nguyen, PhD.

Associate Editor Comments to Author (Professor Chris Chambers):

Associate Editor: 1

Comments to the Author:

The manuscript has now been assessed by five reviewers and a specialist editor. As you will see, the reviews are very detailed and constructive. All find merit in the proposal while also raising concerns within all areas of the Stage 1 review criteria, from the rationale underpinning the hypotheses, the level of methodological detail (including a wide range of issues from the power analysis and justification of the effect size of interest to the handling of missing data), and clarity/consistency of presentation. Although the criticisms are substantial, they do fall within scope for a Stage 1 RR and therefore a Major Revision is invited. Please be sure to respond point-by-point to every comment of every reviewer and the specialist editor, whose comments you can find at the end of the letter.

Reviewer: 1

Comments to the Author(s)

I thank the authors for their submission. Overall, I think the submission targets a timely and important issue. The following questions, comments, suggestions, etc. are posed in hopes of improving or clarifying aspects of this submission.

1. It is unclear to me if the target population is restricted to adults only. If so, how do the authors ensure that respondents are adults? For example, is there a filtering question provided at the beginning of the survey?

Response: *We have now clarified how we restricted participants' age on page 20. It was indicated in the recruitment advert that we posted on WeChat that participants must be older than 18 to participate. Further, the Wenjuanxing online platform allows us to restrict participants' age and anyone who is younger than 18 will be disqualified.*

2. How is the survey distributed? I understand participants are approached via the WeChat app. Is the survey hosted through another provider (e.g. SoSciSurvey or Qualtrics, or some other alternative)?

Response: *Thank you for requesting for this clarification. This information has now been added to page 19. The survey was hosted through Wenjuanxing online platform, a popular tool in China to design academic surveys.*

3. Similarly, where will the data be hosted? I see that the authors are using the Open Science Framework to post their survey variables. I presume data would be hosted there?

Response: *We have now clarified this on page 21 at the end of question 6. All anonymized data will be shared on the project page of this study on the Open Science Framework.*

4. What measures will be taken to ensure that the data remains anonymized/confidential; while still adhering to the requirement of making data/materials publicly available?

Response: *We have now added the information about how data will be anonymized to our answer to question 6 on page 20. Each participant will be asked to only provide their WeChat aliases that will allow for data to be matched across assessments. All WeChat aliases are unique in this sample. WeChat aliases are not connected to participants' identity because, unlike WeChat IDs, they can be changed after data collection is completed. Further we avoided asking any questions about personal information such as participants' real names, identity numbers or phone numbers. Once data across all four waves is matched, all WeChat aliases will be removed before data is shared on Open Science Framework.*

5. It is unclear to me how participants are tracked from Time 1 to Time 2 to Time 3. The authors appear to be using a form of convenience and/or snowball sampling through the WeChat app. I am unfamiliar with the app, so please excuse my ignorance. Will the participants need to have continued to use the app throughout the study for researchers to have access/connect to them? Or are there alternative ways of connecting once they have enrolled in the study? Further, is there a participant ID linked to the participant? Do they know this ID and re-enter it when completing Time 2 and Time 3? Or do the researchers connect the ID to the participant somehow (e.g. using WeChat profile information)?

Response: *We have added this information to our answer to question 5 on page 22. In the survey administered at Time 1, we asked the participants to provide their WeChat IDs to be*

invited back for later surveys. Additionally, we also gave the participants the option to add our WeChat account (created specifically for this study) to their contact list voluntarily.

6. The Registered Report proposal was submitted before Time 2 data collection was completed. Now that I am reviewing it, I believe data collection has stopped. Is there a way you can update us on what the sample size for Time 2 was? And further what % of participants remained in the study? This may be an important indicator for how many people continue through to Time 3 and has implications for the sensitivity analysis performed as part of the proposal. For example, it may affect how large the smallest effect size of interest is.

Response: *Thank you for this suggestion. We have now added demographic makeup of the sample at Time 2 to table 1. We were able to recruit 293 participants (out of 342 participants that allowed us to contact them again) to come back to the study to complete Time 2 survey. We provided the number of participants who have allowed us to contact, the number of those who have completed Time 2 and Time 3 surveys, in the table on page 21. In Table 2, it is shown that attrition rate appeared to be spread out across all demographic groups.*

7. How will the authors handle missing data? Can participants skip questions or are they forced to answer each question before moving to the next?

Response: *We have added this detail to page 19. All survey items were set as required questions, so the participants would need to answer all the questions before proceeding to the next page. Therefore, there will be no missing data within each submitted survey. Because the participants are allowed to withdraw their participation at any time without giving a reason, there are 20 incomplete surveys at Time 1 and 16 at Time 2. Since the number of incomplete responses are not substantial, we will exclude these incomplete surveys from the study.*

8. Was there missing data for Time 1? What percentage? [This can also apply to Time 2 now that it has been collected].

Response: *On page 19, we have clarified that because all the items in the questionnaires were set as required questions, there is no missing data within each submitted survey. We have now provided number of participants who dropped out of the study (completed Time 1 survey but did not provide contacts to be invited back for later surveys) and those who have missing data at either Time 2 or Time 3 on page 24 and page 25.*

9. Are participants being compensated in any way?

Response: *The participants will receive 6 Chinese Yuan, equivalent to 0.85 USD, for completing each survey. We have added this information to page 19.*

I hope these questions, comments, etc. have helped improve the proposal in some way.

Response: Thank you for requesting for these clarifications. Oftentimes details like these are overlooked, yet we recognize that they are very important for transparency.

Thank you for your consideration of them.

Take care,
Daniel Dunleavy

Reviewer: 2

Comments to the Author(s)

From my understanding, the authors have started data collection for an online study with citizens of Wuhan, where a strict lockdown was imposed and has been lifted. The aim of the research was to understand impact of easing lockdown measures. Specifically, examining changes in motivation to self-isolate, the structure of daily life, as well as the psychosocial impact. The researchers identified a small cohort of general population, and took measurement while the lockdown was in place. These people were followed-up over two time points; specifically, at 2-weeks post-lockdown with an additional follow-up planned at 4-weeks post-lockdown.

1.The scientific validity of the research question(s)

The research addresses some interesting questions around the transitioning out of lockdowns. The study addresses this question in population that is ahead of other countries in terms of COVID-19 management and has the potential to offer a unique insight that feeds into our understanding of the impact national movement restrictions have on individuals and whether these are readily reversed with less stringent confinement measures, despite a lack of vaccine availability.

2.The logic, rationale, and plausibility of the proposed hypotheses

The psychosocial impact of a lockdown during a global pandemic is widely discussed currently and research investigating this is quickly being initiated. However, the slightly longer-term impact remains somewhat unaddressed. However, as countries will eventually transition of lockdowns, understanding how populations adapt is relevant.

It may also be useful to not only examine internal experience but also examine self-reported behaviour more closely (as this data is being collected) in order to provide preliminary insights into how populations behave in response to easing of restrictions. Thus, potentially informing transition strategies. I.e. understanding if people remain careful without external pressures may help understand if a gradual transition is necessary. It would be beneficial for the authors to discuss the impact of their research for extensively, over and above scientific inquiry. Examining behaviour may substantiate the potential impact.

Response: To assess how participants behave during and after lockdown, we have included questions about coping strategies that participants used during lockdown (measured at Time 2) and we are asking them again about the strategies they used after lockdown (measured

at Time 3 and Time 4). Participants were asked to only answer “yes” or “no” to each item related to problem-solving, help-seeking, self-blame, avoidance, rationalization, fantasizing, using items adapted from a measure that has been administered to Chinese samples before. Therefore, we will be able to get a descriptive picture of how prevalent each type of coping strategies is during lockdown and after lockdown.

Additionally, we also assessed how intensely different types of stressors, such as marriage and romantic problems, family problems, interpersonal problems, work-related and health-related problems, have affected people’s life during lockdown and after lockdown. This will allow us to see whether different stressors might affect people’s life differently during time in lockdown compared to time after lockdown is lifted.

The reviewer may find these revisions on pages 9-11.

It would seem more natural to frame this research in exploratory, observational terms rather than strongly hypothesis-driven. There are a lot of assumptions being made about establishing baseline severity which aren’t strictly necessary if the interest is in a trend change over time, as indicated by the research questions. A later more qualitative comparison of baseline severity in Wuhan vs. other countries could be discussed. Given the sparsity of research in this area, any directed hypothesis may be premature. I.e. people may continue to have an intrinsic motivation to self-isolate within reasonable ramifications, which may be interesting in and of itself. Also, the evidence cited seems to somewhat contradict the last hypothesis so the reason for adhering to this hypothesis remains unclear.

Response: We agree with this suggestion. Therefore, we have removed the onset hypothesis about the baseline levels of the dependent outcomes. Instead, we will focus on change over time after lockdown. As such, we have revised our analytic plan to use latent growth models to estimate initial levels of motivation, life structure, well-being and ill-being outcomes as well as their slopes of change over time. The reviewer may find the revised hypotheses on pages 16-18 and the revised analytic plan on pages 25-29.

3.The soundness and feasibility of the methodology and analysis pipeline

Generally, the research is sensible and feasible, as demonstrated by the successful collection of data so far. In places, more detail and greater justification of choices would be helpful.

There could be greater clarity on the design of the research study. From my understanding this is intended to be a small cohort of the general population that was recruited during lockdown and followed-up at two time points post-lockdown to assess a change in outcomes in lockdown vs. after lockdown. The timeframe could be specified more clearly early on describing when the lockdown was imposed, when it was lifted and at which time points measurements were taken. This only becomes somewhat apparent later on, with details of how long the lockdown lasted missing. The introduction conveys a sense that the first measurement was taken prior to lockdown.

Response: Thank you for this comment. We have made the following revisions. First, we have decided to follow up with another assessment after Time 3, allowing us to observe change within the first 2 months after the lockdown lift. We have now specified the

timeframe of when each assessment takes place at the beginning of the report on page 3: one assessment prior to lockdown lift on April 8, 2020, and three assessments every two weeks after that.

The questionnaires are adapted so they may therefore lose some of their inherent properties; however, they appear to have face validity. I was unable to access the revised Loneliness Scale (so I am unable to comment on this) but it may be beneficial to explain why only the subscales were used. It is unclear why not all measures were taken at all time points. However, all self-developed questions are open available to view.

Response: *On page 13, we have now added more details about the loneliness measure that we use in this study, as well as the reason for not including all 40 items from the Loneliness Rating Scale. In consideration of space, we are not including 20 other items that concern lonely feelings related to relational frustration and depression-related states (Scalise et al., 1984), since they are not central to our research question and can be redundant with the measure of depressive symptoms.*

The authors have chosen to opt for a multiple of t-tests, without consideration/explanation of the potential impact. An analysis of changes in raw scores over time rather than dichotomised outcomes based on the midpoints of questionnaires simpler and more intuitive. Furthermore, the analysis could be more parsimonious if the interest was more in the relative change over time, as stated by the research questions.

The authors may want to consider analysing all time points collectively. This could be easily implemented through mixed models, be it linear or additive in nature (depending on the expected trend). Additionally, this could allow for effects examined (i.e. it may be plausible to assume that motivation to self-isolate could vary by prior or present COVID-19 infection). However, if hypothesis driven research is preferred, a justification of statistical analysis would be useful as well as the potential limitations of the approach and why they were chosen nonetheless.

Response: *We thank the reviewer for this suggestion. We have now changed our analytic plan to use latent growth models to estimate change over time; this approach allows for higher statistical power to detect growth than our previous plan to use paired sample t-tests.*

The power calculations appear logical. However, it bears the question of what kind of effect sizes you would expect to find. Framing and justifying the threshold used in terms of effect sizes found in previous literature would be helpful.

Response: *We have now changed our power analysis based on previous literature on statistical power using latent growth models. Since there is not a tool to estimate effect sizes specifically for the sample that we are using ($n = 462$), we take a conservative approach to estimate what a smaller sample of 200 will allow us power of .80 or greater with Type I error rates set at .05. You may see this change on page 22.*

4. Clarity and degree of methodological detail for replication

The measures and analyses are explained in detail and most measures are openly available. The procedural elements of the methodology could be expanded more to allow for greater transparency and reproducibility. For example, are any measures being taken to follow-up non-responders as missing data from non-responders will likely be important to discuss? While the demographics are very useful, it would be beneficial to have a greater elaboration of the representativeness of your sample to the population you would like to extrapolate the results to and to what degree you have achieved this. I.e. the user base of WeChat, the types of “forums” being posted in, comparisons to demographic structure of Wuhan citizens, etc. This will contribute to an understanding of the population that is being recruited, which may ultimately speak to the generalisability of your findings. It would also allow for replication in other countries if i.e. cross-cultural variations are ever explored.

Response: *We have now conducted drop-out analysis of those who were enrolled in the study at Time 1 but did not want to be contacted for the Time 2 survey. Further we also conducted missing data analysis of those who were enrolled in the study at Time 2 and left their WeChat IDs to be contacted for the Time 2 survey but did not complete the Time 2 or Time 3 surveys. We conducted Chi-squared tests to examine whether drop-out rates and missing data concentrate in any particular demographic groups. You may find these changes on page 24 and page 25, and the table of demographic makeups of the sample at Time 1, Time 2, and Time 3 on page 36.*

We have also now emphasized in our manuscript that this data is collected using convenience and snowball sampling techniques, as such this remains a limitation of the study that will be addressed in the Discussion section.

5. Potential identification of deviations from protocol

Similar to above, the report appears to be sufficiently explained to pinpoint deviations from the analysis. The procedural aspect of the methods could be elaborated on.

Response: *We have now provided further procedural details on how we recruited our participants on pages 19-20, including the process of ensuring data anonymity. Further, we also laid out the plan for model building on pages 26-29, including the plan on how to handle missing data on page 29.*

6. Sufficient outcome-neutral conditions (e.g. positive controls) for ensuring that the results obtained are able to test the stated hypotheses

Given the design of the research, there appears to be limited capacity to implement this. However, the additional check of asking people generally if their lives changed over time seems sensible.

Response: *We have also now provided reliability data of the measures being used in this study. We will provide a correlation table in our writeup of the results to ensure that the constructs being measured relate to one another in sensible manners. We will also provide a*

table of means and standard deviations and the response scale on which each variable is assessed so the readers can evaluate the prevalence of observed constructs in this sample.

Reviewer: 3

Comments to the Author(s)

The authors present an interesting study proposal on how life changes in the first month after COVID19 lockdowns have been lifted in Wuhan. The expected results, especially regarding motivation for self-isolation, will be important given the recent (sometimes controversial) public discussions on relaxing lockdown regulations and fear of a repeated spread of the virus. It will furthermore be interesting to see whether and how well-being and mental health changes on average. I want to compliment the authors on preparing this study in the midst of the COVID-19 crisis and for deciding to submit a Registered Report to undergo peer review before analysing the data.

Overall, the study's motivation and proposed methodology seem well-rounded and strong. Anyway, I will make some points that may help to improve the design and analyses in the following. I have ordered my points according to their importance for the current submission as a Stage 1 report:

- I understand the authors rationale for the onset hypotheses in self-isolation and structure of daily life measures. The current justification for the effect sizes of interest seem reasonable, but I still fear that the previous data that inform the criteria are not ideal. Maybe there just isn't enough data to inform a specific value that could be considered "normal" for self-isolation and structure of daily life measures? I would still find it very difficult to draw appropriate inferences on whether these measures are on average elevated during lockdown (T1) in the absence of a pre-lockdown measure or previous data (means and variances) of the population of interest (e.g. residents of Wuhan or China). Thus, I might be inclined to prefer if the authors stick with their descriptive design and drop hypotheses 1.1 and 2.1. The focus on changes after the lockdown lift is anyway more interesting and more informative here, I find. (Please note that I think there is enough data to inform the onset hypotheses of the well-being measures, i.e. hypothesis 4.1., and perhaps 3.1.)

Response: *Thank you for this suggestion. We agree with the reviewer that it is very difficult to predict how participants in this sample would perform at Time 1 (the end of lockdown prior to lockdown lift). Therefore, we have removed our onset hypotheses and will focus on change over time after lockdown is lifted.*

- One important question, at least regarding the mental health outcomes, is whether there was an announcement of the lockdown lift before or during data collection of T1? If so, one might expect that the anticipation of the lift already alleviated some psychological distress. Although I am not sure, it might even be possible that there psychological distress is lowest at T1, while it increases again slightly at T2 and / or T3.

Response: *We agree with the reviewer that it is challenging to make predictions of how psychological well-being and ill-being will change after lockdown is lifted. Therefore, we*

have included two competing hypotheses (Hypothesis 3.1 and 3.2) built on two assessments of the current situation. If lockdown has taken a toll on psychological well-being, we expect an increase in psychological well-being (increase in psychological need satisfaction and decrease in loneliness and depression) after lockdown is lifted. However, if announcement of lockdown lift alleviates some psychological distress but people are hit with the reality of economic turmoil, we might see a decrease in psychological well-being (decrease in psychological need satisfaction and increase in loneliness and depression). The reviewer may find the revisions on page 15 and page 16.

- I really like that the authors propose different models for mean-level changes in their variables during their three time points. The planned analyses of the longitudinal data are what I would call “minimally sufficient” for the proposed research questions. However, conducting a range of t-tests brings about some issues, mainly multiple comparison issues. Have the authors considered alternative analyses that allow to analyse all three time points simultaneously, such as repeated-measures ANOVA or even better mixed-effects models (which allow to estimate the model with all available data)?

Response: We thank the reviewer for this suggestion. We have now changed our analytical plan and will use latent growth models (LGM) to analyze change in dependent outcomes over time. LGM allows us greater statistical power to detect change than a range of multiple t-tests or repeated-measures ANOVA. The reviewer may find the revisions to our analytic plan starting on page 26.

- I assume all of the proposed tests (and according power analyses) are two-sided? Regarding the effect sizes of interest and the power calculations: If I understand correctly, the authors chose relatively criteria for power analyses are fairly conservative (alpha = .001, power = .95) and calculate the smallest Cohen’s d that would be detectable given the (expected) sample size. This would be a value of $d = 0.3$. According to my calculations in R, d would even slightly higher given these criteria, namely $d = 0.33$:

```
> power.t.test(n = 226, sig.level = 0.001, power = 0.95,  
+           type = "paired",  
+           alternative = "two.sided")
```

Paired t test power calculation

```
n = 226  
delta = 0.3323347  
sd = 1  
sig.level = 0.001  
power = 0.95  
alternative = two.sided
```

NOTE: n is number of *pairs*, sd is std.dev. of *differences* within pairs

Anyway, I think that $d = 0.33$ is a fairly high effect size for the effects to be studied. As there are few previous data to inform the determination of the smallest effect of interest, I would also consider less conservative criteria ($\alpha = .05$ and power = .85) appropriate to detect effects of $d < .20$. Alternatively, one could set α to .025 to correct for multiple comparisons (2 tests: T1 vs. T2; T2 vs. T3) for each set of trend analyses.

Response: *We thank the reviewer for this very helpful suggestion. We have now changed our analytical plan to use latent growth models to estimate overall change over time. As such we will no longer focus on paired t-test to compare means between time points. We relied on previous papers using Monte-Carlo simulations to estimate effect sizes that can be achieved with a conservative sample of 200 with power $> .80$ and Type I error rates of .05.*

- Have there been previous studies on motivation during and after lockdown? If not, this aspect might be further highlighted as a unique strength, given its relevance for preventing a spread of COVID-19.

Response: *We thank the reviewer for this suggestion. We have now highlighted the significance of looking at change in motivation during and after lockdown to the Introduction on page 5 and page 6.*

- There are some theories and previous research on psychological well-being and mental health that might be relevant to inform the current study (hypotheses 4.1. and hypotheses 4.2). In the depression literature, I would recommend to look into life events research. This paper might be a good starting point: <https://www.ncbi.nlm.nih.gov/pubmed/9046559>. There are some theories in the well-being literature that propose that well-being doesn't change permanently following life events, although recent studies show that there might be enduring effects. A good overview is probably this paper: <https://www.jstor.org/stable/20183166?seq=1>. Although of course a lockdown and its lift are unique events that haven't been considered before, it might be worth embedding the current study in this literature (at least regarding the well-being and mental health measures). It might also help to discuss a potential absence of effects on mental health.

Response: *We greatly appreciate that the review has suggested these two papers. We have incorporated them into our Introduction to consider different possibilities of how well-being and ill-being will change in the current situation. We have also added to this section other ongoing studies that look at change in well-being outcomes during this lockdown, which allows us to make competing hypotheses on how these outcomes will change after lockdown is lifted. The review may find our changes on page 4 in the "Significance" section of this manuscript.*

- If I understand correctly, several scales that have been designed for this study (life structure) or have been used in this population for the first time (e.g. psychological need satisfaction)? In general, it would be important to provide some basic information on internal consistency (e.g. Cronbach's alpha) and validity for all measures, and these scales in particular. For example, it would be good to know to what extent the motivation for self-isolating measure correlates with actual self-isolating.

Response: We have now added Chronbach's alphas to our descriptions of the study measures starting on page 11. We also provided citations of two other studies that have used the measure of psychological need satisfaction in Chinese samples on page 13. In our writeup of the results, we will include a correlation table to show the extent to which our measures of motivation for self-isolation relates to participants' answering "no", "yes", or "in part" to self-isolating in response to the coronavirus outbreak. As you can see in table 1, the percentage of people answering "yes" have dropped from Time 1 to Time 2 and Time 3, and the percentage of people answering "no" and "in part" have increased.

- Have the authors considered robust alternatives to t-tests if the data are skewed (which is to be expected, at least for the mental health data)?

Response: We have now changed our analytic plan to use latent growth models to estimate change over time; this approach allows for higher statistical power to detect growth than our previous plan to use paired sample t-tests.

- In the Stage 2 report, I would recommend providing a drop-out analysis, if you haven't planned to do so anyway. Drop-out analyses would be important to see whether people with certain characteristics preferentially dropped out of the study, which might bias the results and their interpretation.

Response: We thank the reviewer for this suggestion. We have conducted drop-out analysis of those who were enrolled in the study at Time 1 but did not want to be contacted for the Time 2 survey. Further we also conducted missing data analysis of those who were enrolled in the study at Time 2 and left their WeChat IDs to be contacted for the Time 2 survey but did not complete the Time 2 or Time 3 surveys. We performed Chi-squared tests to examine whether drop-out rates and missing data concentrate in any particular demographic groups. You may find these changes on page 24 and page 25, and the table of demographic makeups of the sample at Time 1, Time 2, and Time 3 on page 36.

- Have the authors considered an exploratory analysis to predict motivation for self-isolation at T2 and T3 (e.g. using multiple regression)? I could imagine that, for example, age, financial difficulties, and self-rated health might be associated with internal motivation to self-isolate.

Response: We will not consider exploratory hypotheses in this Registered Report. The instructions for Royal Society Open Science's Registered Report asked the authors not to include exploratory hypotheses, so we have considered only hypotheses where we can make confirmatory predictions based on previous research or observations of the current events that are unfolding.

- In the abstract (l. 37 or 38) it says "prior to lockdown life"; I assume it should read "prior to lockdown lift"?

Response: Thank you for pointing this out to us. This change has been made.

- If possible, I would generally recommend that the authors share their data. Given that there are many studies being conducted in China and other countries, open data might help to allow cross-country comparisons and meta-analyses.

Response: *We intend to share our anonymized data on the project page of this study on Open Science Framework. This intention has now been stated on page 21.*

Reviewer: 4

Comments to the Author(s)

The proposed study aims to describe levels of motivation for self-isolation, structure of daily life, and quality of life in Wuhan during and after lockdown due to the Covid-19 pandemic. The study proposes collecting questionnaire data at 3 time points (T1 during lockdown, T2 shortly after lockdown was lifted, T3 further after lockdown was lifted). T1 and T2 data have already been collected; hence, I will not focus on evaluating or criticising the specific measures used for this Stage 1 Registered Report, as I acknowledge that these cannot be changed at this stage in the research, and I have no major concerns about them. The research question is an important and valid one, and the hypotheses are plausible and clearly described, but I have important concerns about the proposed analysis strategy, particularly the method by which conclusions are drawn from effect sizes, which I have outlined below:

1. The authors propose basing their conclusions on effect size, specifically Cohen's d . However, their proposed contrasts are all reliant on within-subject contrasts (paired samples t -tests or one sample t -tests). As such, the appropriate effect size would be d_{z} – note that this is calculated differently to d (see Lakens, 2013, *Frontiers in Psychology* for a description). If t -tests are to be computed, will non-parametric alternatives be used if assumptions such as normality are broken, and/or will any data transformations be attempted? If non-parametric tests are used, other effect sizes (e.g. r) may be more appropriate.

Response: *We have now changed our analytical plan and will use latent growth models (LGM) to analyze change in dependent outcomes over time. LGM allows us greater statistical power to detect change than a range of multiple t -tests. To perform LGM with full-information maximum likelihood to handle missing data, we will check for the assumption of multivariate normality prior to performing analyses. If assumption of normality is violated, we will use Expectation Likelihood algorithm to obtain parameter estimates. The reviewer may find the revisions to our analytic plan on pages 26-29.*

2. Furthermore, I am not clear that the proposed method for concluding a difference between time points was reached is appropriate. In section 5, the authors outline the results of a sensitivity power analysis ($\alpha = .001$, power = .95), based on their anticipated sample size of 226. I have a number of concerns about this strategy:

a. The research is described as 'descriptive', yet a number of empirical hypotheses are outlined, many with the criteria of observing $d = 0.3$ for concluding a difference between conditions. Given that the justification for $d = 0.3$ relies upon power analysis using an alpha

level of .001, it would make just as much sense to rely on $p < .001$ to conclude a difference between conditions. I am aware that this is not typical of descriptive research, yet this is essentially how the analysis is framed anyway, as the alpha level of .001 used in the power analysis is used to calculate the 0.3 effect size figure. An alternative perhaps more in line with the ethos of descriptive research would be to report 95% (or 99%) confidence intervals for the observed effect sizes (which may also solve another problem, outlined below).

Response: *We thank the reviewer for this suggestion. Our original plan was to go with a descriptive approach, but after receiving the reviewers' suggestions, we have decided to use latent growth models to analyse the slope of change over time. This approach gives us more statistical power to investigate overall change rather than just focusing on differences between any two time points. As such, the results will be presented in forms of parameter estimates of the intercepts and the slopes, and we will provide 95% confidence intervals for the parameter estimates in the writeup of our results.*

b. The authors outline a strategy for concluding whether any observed effects are linear ($d \geq 0.3$ between T1 and T2, then again $d \geq 0.3$ between T2 and T3), or non-linear ($d \geq 0.3$ between two conditions but not between the others). However, unless I have misunderstood, based on their proposed strategy, a T1-T2 difference of $d = 0.3$, then a T2-T3 difference of $d = 0.29$ would be interpreted as 'non-linear', which does not seem like a valid conclusion. Likewise, a T1-T2 difference of $d = 0.8$, followed by a T2-T3 difference of $d = 0.3$ would be interpreted as linear, which again would not seem valid. Without using more complex modelling, one approach, as outlined above, could be to base conclusions on 95% confidence intervals around the effect size. So, for example, if the T1-T2 difference was $d = 0.4$ [CI = 0.3, 0.5] and T2-T3 difference was $d = 0.2$ [CI = 0.1, 0.3], this would provide some evidence for non-linearity (due to non-overlapping confidence intervals, yet both CIs not overlapping 0), whereas if the two effects were $d = 0.3$ [0.2, 0.4] and $d = 0.2$ [0.1, 0.3] this would not provide evidence for non-linearity. This approach would also solve the issue outlined in point c below re: justifying an effect size of 0.3 – in this approach, one would just rely on CIs not overlapping 0, rather than falling above or below an arbitrary effect size. There may be issues with the sample size leading to quite wide confidence intervals - it could be justified to relax to 90% confidence intervals if fully preregistered.

Response: *We recognized that our previous proposal to anticipate effect sizes of difference between two given time points makes it challenging to determine whether change happens linearly or nonlinearly. As such, latent growth models will be better tests to estimate growth across all time points, and whether growth is linear or nonlinear.*

c. I am not convinced that settling on $d = 0.3$ is justified well in the manuscript. There is no justification that this effect size would be 'meaningful' (or that smaller effect sizes would not be meaningful). The only justification is based on the available sample size, yet this sample size was opportunistically gathered so does not provide justification for what is a 'meaningful' effect size in the real world. Note that my concern here is not with the opportunistic sample, but the justification of the effect size.

Response: *We have now removed our predictions around meaningful effect sizes of difference between two given time points. Instead, latent growth models will allow us to*

estimate change across time points without focusing on change between any two specific time points. With this revised analytic plan, there is no need to decide on meaningful effect sizes because the focus now is how much variance in the data is more or less explained by between-subject differences compared to within-subject differences (aka change across time for each person).

d. The anticipated figure of 226, based on 30% attrition at T2 and 30% attrition at T3 seems reasonable, though can the authors point to previous similar research to justify this figure? In this preregistration, the authors should provide a plan for if attrition is higher than this. For example, if attrition reaches 40% rather than 30%, the final sample size would be substantially lower (N~ 166 compared to 226). At what point would attrition be judged 'too high'?

Response: *We realized that we have been too pessimistic originally about how many people will return to complete Time 3 survey. During the time that we have been revising this RR manuscript, we have run the Time 3 survey and have been able to invite 301 participants who have responded to Time 1 and Time 2 surveys to return to the study. In this case, even with the attrition rate of 30%, we hope to still be able to invite at least 200 people back to complete the last survey (Time 4). Further, because we no longer use pair t-tests to analyse change between two given time points and instead use a full-information maximum likelihood algorithm for latent growth models, this will allow us to take advantage of the whole sample of 462 with missing data. We have added information about how many participants returned to complete surveys at later times on page 21 and our plan to handle missing data on page 29.*

3. There is also an issue of bias due to missing data. The authors plan to conduct some analysis on T1 data (N = 462) and some analysis on T1-2-3 data (anticipated N = 226), yet the two analyses may not be comparable if attrition is linked to any of the outcome measures. It is not inconceivable, for example, that participants suffering lower quality of life (e.g., the depression or loneliness measures) are more likely to drop out of the study. I suppose one method to deal with this would be to recalculate effects for the T1 data using only participants that did not later drop-out, though this of course leaves issues of generalizability. There are numerous other techniques for dealing with missing data (though I am not an expert in that area so can provide limited advice).

Response: *We have now added our plan to handle missing data in latent growth models on page 29. We have also conducted drop-out and missing data analyses using participants' demographic characteristics to determine whether drop-out rates and missing data concentrate in any particular demographic groups; this is reported on page 24. In the writeup of our results, we will report drop-out and missing data analyses on the dependent variables, including motivation, life structure, and well-being outcomes, to determine whether those who drop out or have missing data at each time of assessments might have higher or lower scores on those variables compared to those who respond to the surveys.*

4. It should be acknowledged that the sample gained so far at T1 does not seem particularly representative of the wider Wuhan population, being mainly female. Presumably, there are also issues regarding who is likely to use the WeChat social media platform? With an

unrepresentative demographic, at the very least the authors should clearly outline that we should be *very* cautious in generalizing this to the wider Chinese population (and even moreso to the whole world).

Response: *We have now revised the manuscript to reflect the limitation of the present sample obtained through convenience and snowball sampling techniques (pages 19 and 24). We have also provided more information on our sampling method using WeChat app, which is an immensely popular app in China (page 19). While using WeChat to recruit participants for this study has its limitations, this allows us to reach those outside of the typical samples of university undergraduates. The mean age of our sample is around 35.*

5. In a number of places, the manuscript proposes comparing to a midpoint of a Likert scale using a one-sample t-test, though there are a number of aspects of this that I find unclear.
- Firstly, the authors repeatedly classify 3.5 as the midpoint of a 7-point scale and 3 as the midpoint of a 6-point scale, when presumably this should be 4 (1-2-3 and 5-6-7 on each side) and 3.5 (1-2-3 and 4-5-6 on each side) respectively? (Please correct me if I have misunderstood something here!).
 - Secondly, I assume that the authors plan to sum the scores from questionnaires and use a one-sample t-test to compare to the score a participant would receive if they responded at the midpoint for all items – if this is the case, this should be outlined and the specific scores to be compared against mentioned.
 - Thirdly, I think the justification for comparing against a midpoint is quite weak – for example, for hypothesis 3.1, the justification is that in the UK the ‘mean happiness’ was 6.8 out of 10, which the authors interpret as ‘the middle range’, though 6.8 does not seem to be in the middle! Furthermore, the extent to which the mean happiness of UK citizens (using a different scale) is comparable to Wuhan citizens is debatable. Is a mean score from previous research with each scale (in this case the Psychological Need Satisfaction scale) available, and if so could that be used as a benchmark instead?

Response: We agree with the reviewer’s suggestions and other reviewers have raised similar concerns about our onset hypotheses. Therefore, we have removed those hypotheses and focus on predictions about change over time.

Reviewer: 5

Comments to the Author(s)

Nguyen and colleagues propose an interesting and timely descriptive study of the psychological impacts of the government enforced lockdown in Wuhan, China, following the outbreak of COVID-19. Important considerations are below.

1. The scientific validity of the research question(s)

The authors have asked a number of important and timely research questions, which appear scientifically valid.

2. The logic, rationale, and plausibility of the proposed hypotheses

The proposed hypotheses are logical based on existing data and have been well justified by the authors.

3. The soundness and feasibility of the methodology and analysis pipeline (including statistical power analysis where applicable).

The power calculations are sound and appear to reflect estimated effect sizes reported in the literature.

4. Whether the clarity and degree of methodological detail would be sufficient to replicate exactly the proposed experimental procedures and analysis pipeline

The authors have done well to explain the variables used in the study. The nature of the circumstances of the study (a lockdown in a specific region precipitated by a viral outbreak) will make these results inherently difficult to replicate. However, the authors have done well to capture these data in a timely fashion and replication of their findings should be considered in light of these circumstances. The authors should include in their forthcoming manuscript exactly how all variables were handled/coded/transformed in analyses as well as some comments regarding possible clustering of participants within families/households (as this may impact independence of observations).

Response: *Upon releasing the data and making it available on OSF, we will provide a codebook with information regarding how each item is coded and the response scale used to measure each observed variable. However, we are not able to collect information about whether any participants belong to the same families or households as obtaining this information necessarily breaks confidentiality. In other words, we did not ask the participants if another person in their household is also filling out the survey.*

5. Whether the authors provide a sufficiently clear and detailed description of the methods to prevent undisclosed flexibility in the experimental procedures or analysis pipeline
It will be important for the authors to explicitly describe how variables are dealt with in this study and include a STROBE checklist for their reports of missing data, etc.

Response: *We thank the reviewer for this suggestion. In the writeup of the results, we will provide a table to provide numbers of participants with available data at each time point. Because all the items in the questionnaires were set as required questions and participants need to answer all the questions before moving to the next page (otherwise, they can withdraw at any time they want), if participants fill out the survey, they will have all the data for all observed variables.*

Additionally, I have some concern of some of the adapted items on the 'Self-Regulation Questionnaire', with respect to redundancy in both the "Identified motivation for self-isolation" and "External motivation for self-isolating". For example: "because self-isolation was beneficial to me" and "because it was beneficial to me" are almost identical, and "because I was afraid there would be harsh consequences if I didn't self-isolate" and "because I would get into trouble with others if I didn't" are also similar. The authors should explain how they will deal with these items.

Response: We thank the reviewer for this comment; however, we have opted to keep all the items as they are the same items used in another Registered Report that has recently been accepted at Royal Society Open Science (Weinstein & Nguyen, 2020). That paper also looked at people's psychological reactions during lockdown, so keeping the same items will allow for comparisons across the two papers.

Additionally, the authors should describe any possible relatedness between study participants (e.g. spouses, family members, shared household).

Response: Like we have mentioned above, we have not figured out a way to obtain this information without breaking participants' confidentiality. WeChat does not have a function that allows us to do this. Further, we are concerned that asking the participants if any family members or people in the same household also participate in the study might compromise the authenticity of our data. For example, people that are in the same household might start discussing with one another about the survey or sharing their responses with one another.

6. Whether the authors have considered sufficient outcome-neutral conditions (e.g. positive controls) for ensuring that the results obtained are able to test the stated hypotheses

Given the circumstances, I think it will be difficult for authors to consider outcome-neutral conditions (as all study participants will have undergone lock-down). However it would be helpful to provide comparisons with other reports (in Wuhan/China if these exist) regarding scores on the included measures to provide some context.

Response: We agree with the reviewer that ideally, we can compare our data with previous data collected from the same population prior to lockdown. Unfortunately, there is not any data from Wuhan, China, that can be used as comparison for our current study. We will discuss this limitation in the Discussion section.

Reviewer: Specialist Editor

The registered report of Nguyen and colleagues addresses the question of how life changes after the COVID-19 related lockdown. In particular, over three time points the following aspects will be investigated: changes in motivation to self-isolate, changes in structure of daily life and changes in psychological well-being. The study is surely relevant and timely, aiming at observing the post-lockdown effects in the first city publicly affected by the COVID crisis and one with the stringent measures.

I have several suggestions that the authors should consider in the compilation of their report.

1) While the time points of measurement are stated in the abstract, it is only later in the main text that it become clear what t1, t2 and t3 refer to. I would suggest to make them explicit (with dates) already in the introduction

Response: We have now included in the abstract the dates on which surveys were administered at Time 1, Time 2, and Time 3 and which dates Time 4 survey will take place. We also included a diagram at the beginning of the Introduction to describe the time frame of when each assessment is in relation to the lockdown lift.

2) Related to this, the use of the verb tense is changing from past to future (we collected/we will collect) for the same time points, making it difficult to understand if they refer to the same thing. Please use the same tense throughout the manuscript

Response: *The manuscript now has been rewritten in future tense. The final version of this manuscript with the results will be revised to be in past tense.*

3) Descriptive measures: the division of descriptive measures vs. dependent variables needs to be better specified. While I agree that sociodemographic data can be considered descriptive measure, I do not understand why variables such as 4.6.9. and 10. are not analyzed as dependent variables. Especially changes tracked by item 9. And 10. seem to be particularly relevant for later dependent variables (such as psychological well-being)

4) If any relationship between descriptive variables and dependent variables is expected, should be (at least briefly) addressed in the report

Response to 3 & 4: *Due to the unprecedented nature of the current pandemic, we have found it difficult to make confirmatory predictions around change in variables that the editors proposed, including subjective health, self-isolation rates, coping strategies, and stressors. We can provide descriptive data for these variables so readers can observe the means across 4 time points. Further, we have used coping strategies and stressors during lockdown (measured at Time 2) as time-invariant covariates to predict levels of psychological well-being and ill-being and their change over time. Therefore, we have added two new set of hypotheses, Hypotheses 4.1, 4.2, 4.3, and 4.4 on page 17, and Hypotheses 5.1 and 5.2 on page 18.*

5) In the main text, a change in psychological well-being toward negative outcomes, as possible effect of an economic crisis, is mentioned but not elaborated anymore in the hypotheses section. I think this point is very relevant and I would recommend to consider it further.

Response: *We have now elaborated on the effect of economic crisis and turned this into a competing hypothesis (Hypothesis 3.2 on page 16) to be tested simultaneously with the hypothesis that psychological well-being will increase after lockdown is lifted.*

6) A rationale for the distance between t2 and t3 measure should be provided.

Response: *We have now changed our analytic plan, using latent growth models to investigate overall change across 4 time points instead of focusing on the difference between any two given time points. As such, the discussion around meaningful effect size of interest has been removed. Instead, we plan to look at both linear trend nonlinear trend, using model fitness to determine which trend fits our data better.*

7) What is the minimum age for taking part to the survey? Or this is not an inclusion criteria

Response: *We only recruited participants that are 18 years old or above. The process of screening participants based on age has been added to the manuscript on page 20.*

Journal Name: Royal Society Open Science

Journal Code: RSOS

Online ISSN: 2054-5703

Journal Admin Email: openscience@royalsociety.org

Journal Editor: Andrew Dunn

Journal Editor Email: openscience@royalsociety.org

MS Reference Number: RSOS-200705

Article Status: SUBMITTED

MS Dryad ID: RSOS-200705

MS Title: A COVID-19 descriptive study of life after lockdown in Wuhan, China

MS Authors: Nguyen, Thuy-vy; Zhou, Tong; Zhong, Jiayi; Liu, Junsheng

Contact Author: Thuy-vy Nguyen

Contact Author Email: thuy-vy.nguyen@durham.ac.uk

Contact Author Address 1:

Contact Author Address 2:

Contact Author Address 3:

Contact Author City: Durham

Contact Author State: Durham

Contact Author Country: United Kingdom of Great Britain and Northern Ireland

Contact Author ZIP/Postal Code:

Keywords: loneliness, depression, well-being, covid-19

Abstract: This study will examine the psychological experiences of those residing in Wuhan, China, at the end of the 76-day lockdown in response to the coronavirus outbreak. Since mid-February of 2020, being the epicentre of the pandemic, Wuhan with a population of over 11 million people underwent the most stringent lockdown restrictions in the world: no one in the city was allowed to leave their homes without permission and all movements were surveillanced to monitor purchases of cold medicines. On April 8th, 2020, after the Chinese government lifted the lockdown and opened up public transportation, Wuhan residents were allowed to travel outside of the city and go back to work. Yet, given that there is still no vaccine for the virus, this leaves many doubting whether life will indeed go back to normal. Therefore, this research will investigate 3 questions: 1) will people continue to feel motivated to practice self-isolation after the government has eased movement restrictions?, 2) will people experience their life as more planned and structured as they are allowed to go back to work and their normal routines?, and 3) with the promise of return to normality after lockdown is lifted, should we expect that quality of life will become “better” after lockdown is lifted compared to during lockdown? We have collected data from 462 participants in Wuhan, China, prior to lockdown life between the 3rd and 7th of April, 2020 (Time 1), and have followed up with another wave of data collection between 18th and 22nd of April, 2020 (Time 2). We will continue to collect another wave of data between 6th and 10th of May, 2020 (Time 3). This 3-wave study will allow us to see how Wuhan residents’ psychological experiences change (if at all) within the 1st month after lockdown is lifted. We have not performed any confirmatory tests on Time 1 and Time 2 data, but we have looked at demographic information of the current sample to ensure that the sample fits with the purpose of this research.

EndDryadContent

Date Sent: 30-Apr-2020

Appendix B

Dear Prof. Chambers,

We are happy to be submitting revised manuscript of the Stage 1 manuscript with the following minor revisions as requested by the reviewers:

1. We have set criteria to exclude any new participants that have joined in at later time points as they do not have the same baseline (measures before lockdown lift) as those who were recruited at Time 1.
2. We have changed the reports of drop-out and missing data analyses to only include those who have data that can be matched with Time 1. In other words, new participants that have joined in at later time points are not included in those analyses.
3. We have added further details to our modelling plan to clarify which model allows us to test which hypothesis.

Changes have now been made to the manuscript in “green ink”.

We again thank the reviewers for their feedback on this manuscript. Please see our responses to each reviewer’s comments below.

Reviewer: 1

Comments to the Author(s)

I thank the authors for their timely response and thoughtful consideration of my feedback. Below are a few more questions and considerations.

1. In the abstract (p. 3) it states:

"We have collected data from 462 participants in Wuhan, China, prior to lockdown lift between the 3rd and 7th of April, 2020 (Time 1), and have followed up with another wave of data collection between 18th and 22nd of April, 2020 (Time 2), and between 6th and 10th of May, 2020 (Time 3). We will continue to collect one more wave of data between 25th and 29th of May, 2020 (Time 4)."

I would suggest amending this to reflect the number of participants collected for Time 2 (i.e. $n = 293$) and Time 3 ($n = 342$).

Response: *We have now changed this sentence in the abstract to “We have recruited 462 participants in Wuhan, China, prior to lockdown lift between the 3rd and 7th of April, 2020 (Time 1), and have followed up with 292 returning participants between 18th and 22nd of April, 2020 (Time 2), and 284 between 6th and 10th of May, 2020 (Time 3).” to reflect how many participants were recruited and how many have returned to complete later surveys.*

2. Missing data.

On page 25, I would simply recommend being just a bit more explicit in your reporting of what you are comparing. In other words, just being a bit more clear in what timepoint and which groups you are comparing. By all means, it could just be me, but I found it to be a bit tricky to keep track of the total sample for each timepoint, the number of new participants, the number included in the missing data comparison, and how it might relate to previous

timepoints. The table on page 21-22 did help in this regard, but if there is anyway to make that section a bit more clear, I think it would be helpful.

***Response:** We have now made changes to the drop-out and missing data analyses (page 25 & page 26) to include only participants whose data can be matched with Time 1. In other words, we don't count new participants that have joined in at later time points in these analyses. Further, we also made changes to table 1 on page 39 to also include only participants whose data can be matched with Time 1. Further, we also added clarifications to page 23 in green ink to further explain why new participants were able to join the study at later time points. We have also added that we will not include participants who join in at later time points, because those who do not have data prior to lockdown lift do not have the same baseline level as others who were recruited at Time 1.*

Thank you again and take care,
Daniel Dunleavy

Reviewer: 2

Comments to the Author(s)

The authors have made substantial changes to the protocol addressing the previous limitations. The rationale is still very relevant and the procedural detail is much improved statistical methodology appears more fitting now.

However, it is unclear as to why new participants were recruited at each stage. I believe it is more common to have a sample at baseline and then only follow these participants up over time. Or, alternatively sampling a cross-sectional population at each time point as previous surveys have done. It would be good to have a rationale for combining the two and also provide some information on how many people from the original cohort were retained at each time point. This also leads to some uncertainty around how much missing data is present for each participant. It can be problematic to not have adequate baseline to primary follow-up time data, even when using methods to account for missingness. I.e. It would be problematic to have a participant who only contributed data at T3. Also, some consideration to the amount of missing data might be given when using methods to address this problem, especially when missingness is expected to be high (from what I understand ~43% , n= 200 at T4, of the 462 participants who completed the survey at baseline are expected to respond across all time points).

***Response:** We have added clarifications to page 23 in green ink to further explain why new participants were able to join the study at later time points. We have added that we will not include participants who join in at later time points, because we agree that those who do not have data prior to lockdown lift do not have the same baseline level as others who were recruited at Time 1. Thank you for this suggestion.*

We have now made changes to the drop-out and missing data analyses (page 25 & page 26) to include only participants whose data can be matched with Time 1. In other words, we don't count new participants that have joined in at later time points in these analyses.

Further, we also made changes to table 1 on page 37 to also include only participants whose data can be matched with Time 1.

The model building process could be elaborated on slightly further, including planned model evaluation and slightly more clarity on idiosyncrasies of the models for each research question. I.e. clarifying which models are being built for which hypothesis - is model 7 be relevant to hypothesis 1-3? Given that your data will be open access, it would be good to state software and make your annotated code openly available too to make the analysis pipeline transparent and reproducible.

***Response:** We have now clarified which models are being built for which hypothesis (Models 4 - 6 for Hypotheses 1.1, 1.2, 2.1, 3.1, 3.2, Model 7a for Hypotheses 4.1-4.4, and Model 7b for Hypotheses 5.1 & 5.2. We also added detail that models will be built in R using the 'Lavaan' package and all reproducible codes will be shared on Open Science Framework (see page 30).*

Reviewer: 3

Comments to the Author(s)

I think that the revised stage 1 report has considerably improved by the revision. I especially think that dropping the baseline hypotheses and switching to growth models to model the observed changes will make the study stronger. I also thank the authors for addressing all my comments point.

I only have two small comments: First, Cronbach's alpha, as now provided, is better described as "internal consistency" (rather than internal reliability).

***Response:** Thank you for this suggestion. We have made this change to the manuscript.*

Second, it would be good to be more explicitly how the authors want to discern no linear change, linear increase, and linear decrease. The first (no linear change vs. linear change) should be reflects in model comparison 3 vs. 4. it's good to note that an increase would be reflected in a positive coefficient (slope intercept), and a decrease in a negative one.

***Response:** Thank you for this suggestion. We have now clarified on page 29 to further explain what it means if Model 4 significantly yields better fit than Model 3, and what it means to have positive vs. negative coefficient of the slope intercept.*

Reviewer: 4

Comments to the Author(s)

I think this revised manuscript is much improved on the first submission. My main concerns from the first submission regarded the use of effect sizes and t-tests, so the inclusion of latent growth models has thus improved the analysis strategy. Although I have never used LGMs (and so cannot comment in detail on the proposed specifics of the modelling), this usage seems appropriate and well detailed. A number of the hypotheses which I had queries about have been removed or replaced, which I also think improves this manuscript, and it is

good to see that so far there has been a relatively low attrition rate across time points, removing my concerns about possible high attrition. The addition of a fourth time point also seems worthwhile.

I would like to acknowledge that the authors have done a lot of work to improve this manuscript in a relatively short amount of time, and have responded in detail to all reviewers. I believe this will be an important and interesting study and will look forward to reviewing the eventual Stage 2 manuscript.

Response: Thank you for your comments to this manuscript; they indeed have helped us greatly improve it.

Reviewer: 5

Comments to the Author(s)

The authors have done a nice job responding to my comments. I have no further critiques and wish them well.

Response: Thank you!

Journal Name: Royal Society Open Science

Journal Code: RSOS

Online ISSN: 2054-5703

Journal Admin Email: openscience@royalsociety.org

Journal Editor: Andrew Dunn

Journal Editor Email: openscience@royalsociety.org

MS Reference Number: RSOS-200705.R1

Article Status: SUBMITTED

MS Dryad ID: RSOS-200705.R1

MS Title: A COVID-19 descriptive study of life after lockdown in Wuhan, China

MS Authors: Nguyen, Thuy-vy; Zhou, Tong; Zhong, Jiayi; Liu, Junsheng

Contact Author: Thuy-vy Nguyen

Contact Author Email: thuy-vy.nguyen@durham.ac.uk

Contact Author Address 1:

Contact Author Address 2:

Contact Author Address 3:

Contact Author City: Durham

Contact Author State: Durham

Contact Author Country: United Kingdom of Great Britain and Northern Ireland

Contact Author ZIP/Postal Code:

Keywords: loneliness, depression, well-being, covid-19

Abstract: On April 8th, 2020, the Chinese government lifted the lockdown and opened up public transportation in Wuhan, China, the epicentre of the COVID-19 pandemic. After 76 days in lockdown, Wuhan residents were allowed to travel outside of the city and go back to work. Yet, given that there is still no vaccine for the virus, this leaves many doubting whether life will indeed go back to normal. Therefore, this research will investigate 5 questions: 1) will people continue to feel motivated to practice self-isolation after the

government has eased movement restrictions?, 2) will people experience their life as more planned and structured as they are allowed to go back to work and their normal routines?, 3) with the promise of return to normalcy after lockdown is lifted, should we expect that quality of life will become “better” after lockdown is lifted compared to during lockdown?, 4) Which coping strategies used during lockdown and after lockdown is lifted will contribute to psychological well-being and alleviate ill-being?, and 5) Which stressors experienced during lockdown and after lockdown is lifted will undermine psychological well-being and contribute to ill-being? We have collected data from 462 participants in Wuhan, China, prior to lockdown lift between the 3rd and 7th of April, 2020 (Time 1), and have followed up with another wave of data collection between 18th and 22nd of April, 2020 (Time 2), and between 6th and 10th of May, 2020 (Time 3). We will continue to collect one more wave of data between 25th and 29th of May, 2020 (Time 4). This 4-wave study will use latent growth models to examine how Wuhan residents’ psychological experiences change (if at all) within the first two months after lockdown is lifted. We have not performed any confirmatory tests on Time 1, Time 2, and Time 3 data, but we have looked at demographic information of the current sample to ensure that the sample fits with the purpose of this research.

EndDryadContent

Chris Chambers BSc PhD CPsychol FBPsS
Professor of Cognitive Neuroscience
School of Psychology
Cardiff University
CF10 3AT
United Kingdom

Tel: +44 (0)29-208-70331

Email: chambersc1@cardiff.ac.uk

Official site: <https://www.cardiff.ac.uk/people/view/133632-chambers-chris>

Blog: <http://www.theguardian.com/science/head-quarters>

Twitter: @chrisdc77

Appendix C

Dear Prof. Chambers,

We are happy to be submitting Stage 2 manuscript titled “A COVID-19 descriptive study of life after lockdown in Wuhan, China” (RSOS-200705.R2) with results reported. Please note that the following changes have been made:

1. We changed the authorship order as follows: Tong Zhou, Thuy-vy Nguyen, Jiayi Zhong, Junsheng Liu
2. We changed all sentences in the manuscript to past tense
3. We have made all measures, data and code available on Open Science Framework at https://osf.io/5rxz6/?view_only=b3a7c3653bd043d189eea2513e66bfde

We again thank the editor and reviewers for their feedback on this manuscript and assistance throughout this process. We look forward to further comments to improve this manuscript.

Thuy-vy Nguyen
University of Durham
Tong Zhou
East China Normal University

Appendix D

Dear Prof. Chamber and the reviewers of this Registered Report,

We have gone over each of your comments and addressed all of them in the submitted revision. We have found this process incredibly helpful for us from the beginning stage of designing the study and planning out our analyses until the final stage of interpreting the data we have collected. We believe this process has greatly improved the manuscript and we thank all of you for your dedicated efforts and a constructive review process.

Best wishes,
Zhou Tong and Thuy-vy Nguyen

Please see our specific responses below:

Editor's revision request: In addition to addressing the reviewers' comments, please also update the Abstract to summarise the results and conclusions. Also, please move the study design table from Appendix A to either the main text of the Methods, or (better yet in my opinion), update the table to include an extra column on the right that states the *outcome* of each test and place the table at or near the start of the Results section (changing the table from portrait to landscape mode, and reducing the font size, would help fit in this extra information while keeping the table readable). I think an augmented design table with results would be useful addition to the Stage 2 manuscript that will help readers quickly digest the questions posed and answers obtained, but I will leave the authors to decide which approach works best for them (either including the current design table as-is in the Method, or an enhanced version in the Results). Either way, please move it from the Appendix to the main text.

Response: We have now moved the study design table from Appendix A and turn it into a table embedded on pages 32-36. We also added in the summary of the findings for each research question in this table.

Reviewer 1's revision request: My only major feedback to the discussion section, would be to add more detailed content describing how these concepts (e.g. isolation, coping, resilience) may be further investigated in future research or follow-up studies, by use of qualitative or other interviewing methods....methods that may complement this study and help triangulate the results found here.

Response: We thank the reviewer for this suggestion. We believe further qualitative data will clarify and contextualise the current quantitative results in meaningful ways. We have added this discussion to the Discussion on pages 50-51.

Reviewer 1's revision request: Pages 51-66: While I'm not a huge proponent of using asterisks in tables, I'd suggest that (for each table that uses asterisks for reporting) you include a note at the bottom explicitly stating what each refers to. This is done in some places, but not others.

Response: Thank you for catching this. We have added this information in all the tables where asterisks are presented.

Reviewer 1's revision request: Page 71: To increase clarity, I might suggest in Appendix B, that the three notes stating: "1. "Participants who filled...were taken into consideration." be phrased as "...are reported here."

Response: We have made these changes as the reviewer suggested in Appendix A on page 74.

Reviewer 4's revision request: The descriptions of internal reliability for some of the self-report measures (page 11 onwards) seem to have been removed (or moved, and I can't find them!) – I think these were useful for the reader to assess the measures, and should be included if possible.

Response: We have removed the descriptions of internal reliability from the self-reported measures and moved them to the correlation table (Table 1).

Reviewer 4's revision request: I think the conclusion in the very final paragraph regarding mental health ("this set of findings do not yield evidence to support the current media concerns around rise in mental health issues... after imposed lockdown") is perhaps overstated. Given that there was no pre-lockdown assessment of the mental health variables, it is hard to conclude whether there was a lockdown-related rise (which has remained steady across the four timepoints) or no change at all. This is discussed well by the authors earlier in the discussion (with reference to the Gao et al., 2020 study), so it is only this final summary sentence that I think should be toned down a bit.

Response: We have now changed our interpretation as the reviewer suggested. We agree that a hint on no change would be an over-stated conclusion so we have revised the conclusion to say that our findings did not yield evidence of dramatic changes. Please see our highlighted revision on page 56.

Reviewer 4's revision request: I think it may also be worth noting in the discussion further limitations of the conclusions that can be drawn about mental health – for example, we can't say much about long term effect on mental health, nor the effects on vulnerable populations (e.g., children), as has been speculated upon in the media.

Response: We have added a paragraph on page 51 discussing the limitation of not being able to make conclusion about the long-term effect of the pandemic on mental health in the general population.

Reviewer 4's revision request: Typos

Response: All those typos have been corrected as noted below. Thank you for spotting these errors.

- P26, line 50 – presumably should read 'sample of 200' not 'sample of 2'.

This has been changed on page 26

- P39 – 'table x' should be replaced with the correct table number.

This has been changed on page 44

- P43, line 38 – should read 'the current study', not 'current study'

The sentence “First, we stick to the data analysis plan where we have registered at Stage 1 to add more transparency in current study” has been changed to “First, we followed the data analysis plans that we have registered at Stage 1.”

- P43, line 48 – should read ‘isolation’ not ‘solation’.

This has been changed on page 48.

Specialist Associate Editor’s revision request: One thing I noticed is that at p.7 line 50 onward (Measures for descriptive analyses), they mention only t1, t2 and t3, but the measures were also collected at t4. they should correct that.

Also for consistency, I would report frequencies and means for all the mentioned measure in each subsection (and not only in the appendix) at each time point (t1 to t4).

Response: We apologise for missing these details in our previous submission. We have now added in the frequencies and means for all mentioned measure in each subsection at each time point from Time 1 to Time 4. You can see the highlighted changes between pages 8 to 10.